# SHANG++: ROBUST STOCHASTIC ACCELERATION UNDER MULTIPLICATIVE NOISE

## ABSTRACT

Training with multiplicative noise scaling (MNS) is often destabilized by momentum methods such as Nesterov's acceleration, as gradient noise can overwhelm the signal. A new method, SHANG++, is introduced to achieve fast convergence while remaining robust under MNS. With only one-shot hyperparameter tuning, SHANG++ consistently reaches accuracy within $1\%$ of the noise-free setting across convex problems and deep networks. In experiments, it outperforms existing accelerated methods in both robustness and efficiency, demonstrating strong performance with minimal parameter sensitivity.

## 1 INTRODUCTION

Empirical Risk Minimization (ERM) is central to modern large-scale machine learning, including deep neural networks and reinforcement learning (Hastie et al., 2009). Given a large dataset $\{(X_i, Y_i)\}_{i=1}^{N}$, where $Y_i$ denotes the label of data $X_i$ and $N \gg 1$, the training objective is

$$\min_x f(x), \qquad f(x) = \frac{1}{N} \sum_{i=1}^{N} f_i(x), \tag{1.1}$$

where $x$ denotes the network parameters and $f_i(x)$ is the loss associated with sample $(X_i, Y_i)$. We use $x$ instead of $\theta$ for consistency with the optimization formulation. Efficiently computing the minimizer $x^\star = \arg\min_x f(x)$ is critical for training neural network with large data.

Exact gradient evaluation is expensive, so Stochastic Gradient Descent (SGD) uses mini-batches:

$$g(x) = \frac{1}{M} \sum_{i \in B} \nabla f_i(x), \tag{1.2}$$

where $B \subset \{1, \ldots, N\}$ is a random batch of size $M$. SGD slows down when the condition number of $f$ is large. Momentum methods such as Heavy Ball (HB) (Polyak, 1964) and Nesterov accelerated gradient (NAG) (Nesterov, 1983) are widely used to accelerate convergence. In training deep neural networks, Adam (Adaptive Moment Estimation) (Kingma & Ba, 2015) is a widely used optimization algorithm that combines momentum and adaptive step sizes for fast and stable convergence.

The mini-batch estimator $g(x)$ reduces the cost of computing $\nabla f(x)$ but introduces noise. In regimes such as small-batch training or highly over-parameterized models, the variance can scale with and even dominate the signal $\|\nabla f(x)\|^2$. This effect is modeled by the multiplicative-noise scaling (MNS) condition (Wu et al., 2019; 2022b; Gupta et al., 2024). Hodgkinson & Mahoney (2021) further shows that multiplicative noise induces geometric distortions in the loss landscape, beyond the smoothing effects of additive noise.

**Definition 1.1** (Multiplicative Noise Scaling (MNS)). The stochastic gradient estimator $g(x)$ satisfies the MNS condition if there exists $\sigma \geq 0$ such that

$$\mathbb{E}\left[\|g(x) - \nabla f(x)\|^2\right] \leq \sigma^2 \|\nabla f(x)\|^2. \tag{1.3}$$

**Related work.** Accelerated variants of SGD have been extensively studied. However, momentum methods are highly sensitive to stochastic noise (Devolder et al., 2014; Aujol & Dossal, 2015; Liu et al., 2018), and stability depends critically on parameter choices (Kidambi et al.,

2018), (Liu & Belkin, 2020; Assran & Rabbat, 2020; Ganesh et al., 2023). Gupta et al. (2024) further showed that under MNS with $\sigma \geq 1$, NAG fails to converge even in convex and strongly convex settings. In practice, the apparent benefits of momentum largely arise from large mini-batches, which reduce gradient variance and make the dynamics closer to the deterministic regime.

To address these issues, a series of corrections have been developed. Following Jain et al. (2018), many accelerated stochastic algorithms have been proposed (Liu & Belkin, 2020; Vaswani et al., 2019; Even et al., 2021; Bollapragada et al., 2022; Laborde & Oberman, 2020; Gupta et al., 2024; Hermant et al., 2025), aiming to retain acceleration while improving robustness to noise. Vaswani et al. (2019) introduced a four-parameter NAG variant with optimal accelerated rates; Liu & Belkin (2020) proposed the Mass method with a three-parameter correction, though acceleration was proved only for over-parameterized linear models; Gupta et al. (2024) developed AGNES with guarantees matching Vaswani et al. (2019); and Hermant et al. (2025) analyzed SNAG, a four-parameter variant in Nesterov's framework (Nesterov, 2012), showing similar rates under mild tuning. A more detailed discussion appears in Appendix F.

From the viewpoint of convex theory, these algorithms are competitive. However, our deep-learning experiments show that they often lose acceleration under high noise and can perform worse than SGD even with recommended hyperparameters (see Section 3). For example, on CIFAR-100 with ResNet-50 and batch size 50, SGD attains $58.326\%$ test accuracy whereas AGNES reaches only $42.82\%$. With smaller batches, both AGNES and SNAG exhibit strong oscillations and require additional hyperparameter tuning.

**Contribution.** Motivated by this gap, our goal is not only to design another accelerated method, but to develop a complementary approach that (i) retains optimal theoretical guarantees, (ii) reduces tuning effort, and (iii) improves stability. Our contributions emphasize simplicity (fewer parameters), provable acceleration with explicit noise dependence, and robust empirical behavior.

1. We begin with SHANG, a stochastic extension of HNAG (Chen & Luo, 2021). Unlike the classical Heavy-Ball method, HNAG includes the Hessian term $\nabla^2 f(x) x'$, yielding a more accurate continuous-time model of NAG. SHANG inherits this structure and already demonstrates noise-suppression behavior.

2. We then refine SHANG into SHANG++ using the $\mu$-**shift principle**: replacing $f$ with $f_{-\mu}(x) = f(x) - \frac{\mu}{2}\|x - x^\star\|^2$ reduces the effective Lipschitz constant and introduces a correction term $-\beta\mu(x_{k+1} - x_k)$. SHANG++ generalizes this to a flexible correction $-m(x_{k+1} - x_k)$ that does not require strong convexity, and helps mitigate the multiplicative-noise–induced rescaling of the key constants $\mu$ and $L$. The $\mu$-shift mechanism and its noise-suppression effect are new and absent from HNAG. SHANG++ achieves optimal accelerated rates in both convex and strongly convex settings with multiplicative noise.

3. We evaluate SHANG++ on convex optimization, image classification, and generative modeling tasks (MNIST, CIFAR-10, CIFAR-100). SHANG++ matches or outperforms NAG, SNAG, AGNES, and Adam, with clear advantages under high multiplicative noise in Section 3.

4. Section 3 further examines robustness to multiplicative noise. For realistic noise levels ($\sigma \leq 0.5$), SHANG++ retains near noise-free accuracy (within $1\%$ degradation), demonstrating that stability can be achieved with fewer parameters and a simpler design than earlier corrections such as AGNES and SNAG.

**Limitation.** Current convergence guarantees cover only convex objectives under multiplicative-noise scaling and do not yet extend to general nonconvex landscapes. Empirically, the method typically enters locally convex basins after leaving unstable saddle regions, suggesting that similar stability mechanisms operate in deep networks. We are exploring extensions under the Polyak–Łojasiewicz condition and weak-convexity assumptions, where our Lyapunov framework naturally applies.

Although SHANG++ reduces tuning complexity through one-shot, non-adaptive hyperparameters, its performance may still depend on accurate estimates of smoothness constants (e.g., $L, \mu$). In highly non-convex settings or under very high noise, the one-shot strategy may require refinement.

**Notation.** Let $f : \mathbb{R}^d \to \mathbb{R}$ be differentiable. The Bregman divergence of $f$ between $x, y \in \mathbb{R}^d$ is

$$D_f(y, x) := f(y) - f(x) - \langle \nabla f(x), y - x \rangle.$$

The function $f$ is $\mu$-strongly convex if for some $\mu > 0$, $D_f(y, x) \geq \frac{\mu}{2}\|y - x\|^2$, $\quad \forall x, y \in \mathbb{R}^d$.

It is $L$-smooth, for some $L > 0$, if its gradient is $L$-Lipschitz:

$$\|\nabla f(y) - \nabla f(x)\| \leq L\|y - x\|, \quad \forall x, y \in \mathbb{R}^d.$$

Let $\mathcal{S}_{\mu,L}$ be the class of all differentiable functions that are both $\mu$-strongly convex and $L$-smooth. For $f \in \mathcal{S}_{\mu,L}$, the Bregman divergence satisfies

$$\frac{\mu}{2}\|x - y\|^2 \leq D_f(x, y) \leq \frac{L}{2}\|x - y\|^2, \quad \forall x, y \in \mathbb{R}^d, \tag{1.4}$$

Bregman divergence here is used purely as an analytical tool in the Lyapunov analysis. Parameters $\mu$ and $L$ are treated as known hyperparameters for the given problem. Their adaptivity is beyond the scope of this work.

## 2 STOCHASTIC HESSIAN-DRIVEN ACCELERATED NESTEROV GRADIENT

**Flow.** To accelerate gradient descent, Polyak introduced a momentum term, which incorporates information from previous iterates, inspired by the "heavy-ball" ODE model (Polyak, 1964):

$$x'' + \theta x' + \eta \nabla f(x) = 0. \tag{2.1}$$

However, the discrete heavy-ball method $x_{k+1} = x_k - \gamma \nabla f(x_k) + \beta(x_k - x_{k-1})$ can diverge; see Lessard et al. (2016); Goujaud et al. (2025) for non-convergent examples.

We will use the second-order dynamical system introduced in Chen & Luo (2019; 2021), known as the Hessian-driven Nesterov Accelerated Gradient (HNAG) flow:

$$\gamma x'' + (\gamma + \mu)x' + \beta\gamma\nabla^2 f(x)x' + (1 + \mu\beta)\nabla f(x) = 0, \tag{2.2}$$

where $\beta > 0$ is a parameter and $\gamma$ is a time-scaling function. Compared with the classical HB flow (2.1), the additional Hessian-driven term $\nabla^2 f(x)x'$ captures how the local curvature of $f$ affects the damping strength of the dynamics. As shown in Chen & Luo (2019), this curvature aware mechanism provides a more accurate continuous-time description of NAG. The second-order ODE (2.2) can be equivalently reformulated as the first-order system:

$$x' = v - x - \beta\nabla f(x), \qquad v' = \frac{\mu}{\gamma}(x - v) - \frac{1}{\gamma}\nabla f(x), \qquad \gamma' = \mu - \gamma, \tag{2.3}$$

which removes the explicit dependence on $\nabla^2 f(x)$.

**Methods.** Discretizing (2.3) via a Gauss–Seidel–type scheme, adding an extra term $-m(x_{k+1} - x_k)$ to the $x$-update, and replacing $\nabla f(x_k)$ with an unbiased estimator $g(x_k)$ yield the Stochastic Hessian-driven Nesterov Accelerated Gradient (SHANG++) method:

$$\begin{cases} \dfrac{x_{k+1} - x_k}{\alpha_k} = v_k - x_{k+1} - m(x_{k+1} - x_k) - \beta_k g(x_k), \\[2mm] \dfrac{v_{k+1} - v_k}{\alpha_k} = \dfrac{\mu}{\gamma_k}(x_{k+1} - v_{k+1}) - \dfrac{1}{\gamma_k}g(x_{k+1}), \\[2mm] \dfrac{\gamma_{k+1} - \gamma_k}{\alpha_k} = \mu - \gamma_{k+1}, \end{cases} \tag{2.4}$$

where $\alpha_k > 0$ is the step size, $m \geq 0$ controls the extra noise-damping term $-m(x_{k+1} - x_k)$, and $\beta_k > 0$ depends on $\alpha_k$ and $\gamma_k$, typically scaling as $\frac{\alpha_k}{\gamma_k/(1+\sigma^2)}$.

If the damping term is absorbed into the left-hand side, the $x$-update becomes

$$\frac{x_{k+1} - x_k}{\tilde{\alpha}_k} = v_k - x_{k+1} - \beta_k g(x_k), \tag{2.5}$$

where $\tilde{\alpha}_k = \frac{\alpha_k}{1+m\alpha_k} \leq \alpha_k$. SHANG++ can thus be interpreted as a modified discretization of the HNAG flow with a reduced step size $\tilde{\alpha}_k$. The case $m = 0$ recovers SHANG, a direct stochastic extension of HNAG. The "++" indicates two improvements: faster theoretical convergence and greater robustness to noise. With the parameter choices specified in Theorem 2.1 for the strongly convex case $f \in \mathcal{S}_{\mu,L}$, and in Theorem 2.2 for $\mu = 0$, accelerated convergence rate can be established.

**SHANG++ for Strongly Convex Minimization.** Setting $\gamma = \mu$ and $m = 1$ when $f \in \mathcal{S}_{\mu,L}$ with $0 < \mu < L < \infty$. Define the auxiliary variable $x_k^+ := x_k - \tilde{\alpha}\beta g(x_k)$. Then SHANG++ can be rewritten in the following form:

$$\frac{x_{k+1} - x_k^+}{\tilde{\alpha}} = v_k - x_{k+1},$$
$$\frac{v_{k+1} - v_k}{\alpha} = x_{k+1} - v_{k+1} - \frac{1}{\mu}g(x_{k+1}). \tag{2.6}$$

where $\tilde{\alpha} = \frac{\alpha}{1+\alpha}$. Schemes (2.6) and (2.4) generate the same sequences $(x_k, v_k)_0^\infty$; the explicit appearance of $x_k^+$ is only for analysis and does not affect the algorithm itself.

**Theorem 2.1.** *Let $f \in \mathcal{S}_{\mu,L}$. Given $x_0^+ = v_0 = x_0$, suppose $(x_k, x_k^+, v_k)$ are generated by (2.6) with $g(x_k)$ defined in (1.2) and MNS (1.3) holds. If the step size satisfies $\alpha = \frac{\tilde{\alpha}}{1-\tilde{\alpha}}$ with $0 < \tilde{\alpha} \le \frac{1}{1+\sigma^2}\sqrt{\frac{\mu}{L}}$, and $\beta = \frac{\tilde{\alpha}}{\mu/(1+\sigma^2)}$, then*

$$\mathbb{E}\left[f(x_k^+) - f(x^\star) + \frac{\mu}{2}\|v_k - x^\star\|^2\right] \le (1+\alpha)^{-k}\left(f(x_0) - f(x^\star) + \frac{\mu}{2}\|v_0 - x^\star\|^2\right).$$

We give a proof sketch of Theorem 2.1 and refer to Appendix C.1 for full details, which cover the range $0 \le m \le 1$; Theorem 2.1 treats the optimal special case $m = 1$ and shows that $\mathbb{E}[f(x_k) - f(x^\star)]$ contracts linearly at rate $\mathcal{O}\left((1 - \frac{1}{1+\sigma^2}\sqrt{\mu/L})^k\right)$. Note that $m = \beta\mu \le \sqrt{\mu/L}$ is also a particular instance with $0 \le m \le 1$.

*Proof.* Let $z_k^+ = (x_k^+, v_k)$ and define the Lyapunov function

$$\mathcal{E}(z_k^+) = f(x_k^+) - f(x^\star) + \frac{\mu}{2}\|v_k - x^\star\|^2. \tag{2.7}$$

Given $(x_k, v_k)$ and $g(x_k)$, the quantities $x_k^+$ and $x_{k+1}$ are deterministic, while randomness is introduced through $g(x_{k+1})$ and consequently affects $(x_{k+1}^+, v_{k+1})$. The expectation $\mathbb{E}$ is with respect to the randomness in $g(x_{k+1})$.

First of all, we have the sufficient decay of SGD for $x_{k+1}^+ := x_{k+1} - \tilde{\alpha}\beta(x_{k+1})$: if $\tilde{\alpha}\beta = \frac{\tilde{\alpha}^2}{\mu/(1+\sigma^2)} \le \frac{1}{(1+\sigma^2)L}$, which is equivalent to $\tilde{\alpha} \le \frac{1}{(1+\sigma^2)}\sqrt{\mu/L}$, then

$$\mathbb{E}\left[f(x_{k+1}^+) - f(x_{k+1})\right] \le -\tilde{\alpha}\beta/2 \cdot \|\nabla f(x_{k+1})\|^2 = -(1+\sigma^2)\tilde{\alpha}^2/2\mu \cdot \|\nabla f(x_{k+1})\|^2. \tag{2.8}$$

By the definition of Bregman divergence, $\mathcal{E}(z_{k+1}) - \mathcal{E}(z_k^+) = \langle \nabla\mathcal{E}(z_{k+1}), z_{k+1} - z_k^+ \rangle - D_\mathcal{E}(z_k^+, z_{k+1})$. Expanding the term $\langle \nabla\mathcal{E}(z_{k+1}), z_{k+1} - z_k^+ \rangle$ and using the update in (2.6) gives

$$-\tilde{\alpha}\langle \nabla f(x_{k+1}) - \nabla f(x^\star), x_{k+1} - x^\star \rangle - \frac{\alpha\mu}{2}\|v_{k+1} - x^\star\|^2 - \frac{\alpha\mu}{2}\|v_{k+1} - x_{k+1}\|^2$$
$$+ \frac{\alpha\mu}{2}\|x_{k+1} - x^\star\|^2 + \alpha\langle g(x_{k+1}), v_k - v_{k+1} \rangle - (\alpha - \tilde{\alpha})\langle g(x_{k+1}), v_k - x^\star \rangle \tag{2.9}$$
$$+ \tilde{\alpha}\langle \nabla f(x_{k+1}) - g(x_{k+1}), v_k - x^\star \rangle$$

After taking the expectation $\mathbb{E}(\langle \nabla f(x_{k+1}) - g(x_{k+1}), v_k - x^\star \rangle) = 0$. We use $v_k - x^\star = (1 + \alpha)(v_{k+1} - x_{k+1}) + (x_{k+1} - x^\star) + \frac{\alpha}{\mu}g(x_{k+1})$ and the identity $2\langle a, b \rangle = \|a\|^2 + \|b\|^2 - \|a - b\|^2$ to bound the cross term $-(\alpha - \tilde{\alpha})\langle g(x_{k+1}), v_k - x^\star \rangle = -\alpha\tilde{\alpha}\langle g(x_{k+1}), v_k - x^\star \rangle$:

$$-\tilde{\alpha}\mu\langle\frac{\alpha}{\mu}g(x_{k+1}), (1+\alpha)(v_{k+1} - x_{k+1})\rangle - \alpha\tilde{\alpha}\langle g(x_{k+1}), x_{k+1} - x^\star \rangle - \frac{\alpha^2\tilde{\alpha}}{\mu}\|g(x_{k+1})\|^2$$
$$= -\frac{\tilde{\alpha}\mu}{2}\|v_k - x_{k+1}\|^2 - \frac{\alpha^2\tilde{\alpha}}{2\mu}\|g(x_{k+1})\|^2 + \frac{\alpha(1+\alpha)\mu}{2}\|v_{k+1} - x_{k+1}\|^2 - \alpha\tilde{\alpha}\langle g(x_{k+1}), x_{k+1} - x^\star \rangle \tag{2.10}$$

The last term can be combined with the first term of (2.9) after taking expectations, and using strong convexity we obtain:

$$-\alpha\langle \nabla f(x_{k+1}) - \nabla f(x^\star), x_{k+1} - x^\star \rangle \le -\alpha(f(x_{k+1}) - f(x^\star) + \frac{\mu}{2}\|x_{k+1} - x^\star\|^2) \tag{2.11}$$

This negative contribution cancels the corresponding positive term in (2.9). The most difficult term is the expectation of the cross term $\mathbb{E}\left[\langle g(x_{k+1}), v_k - v_{k+1}\rangle\right]$, as both $g(x_{k+1})$ and $v_{k+1}$ are random variables. Using the identity $2\langle a, b\rangle = \|a\|^2 + \|b\|^2 - \|a - b\|^2$ again to obtain

$$\alpha\langle g(x_{k+1}), v_k - v_{k+1}\rangle = \frac{\alpha^2}{2\mu}\|g(x_{k+1})\|^2 + \frac{\mu}{2}\|v_k - v_{k+1}\|^2 - \frac{\alpha^2\mu}{2}\|v_{k+1} - x_{k+1}\|^2, \quad (2.12)$$

where the term involving $v_{k+1} - x_{k+1}$ follows from $\frac{v_k - v_{k+1}}{\alpha} - \frac{1}{\mu}g(x_{k+1}) = v_{k+1} - x_{k+1}$ by the update of $v_{k+1}$. The positive $\frac{\mu}{2}\|v_k - v_{k+1}\|^2$ is canceled by $-\frac{\mu}{2}\|v_k - v_{k+1}\|^2$ contained in $-D_{\mathcal{E}}(z_k^+, z_{k+1})$. The stochastic gradient term splits into two parts: one part is directly canceled by the corresponding negative term in (2.10). For the remaining part, taking expectations termwise and applying the MNS condition yields the positive gradient contribution $\frac{\tilde{\alpha}\alpha(1+\sigma^2)}{2\mu}\|\nabla f(x_{k+1})\|^2$, which is then canceled by the negative term in the sufficient decay condition (2.8), together with the additional negative term generated by applying the same sufficient decay condition to $f(x_{k+1}) - f(x^\star)$. This cancellation motivates our choice of $(\alpha, \beta, m)$.

Combining all the above estimates, we obtain,

$$\mathbb{E}\left[\mathcal{E}(z_{k+1}^+)\right] - \mathcal{E}(z_k^+) \leq \mathbb{E}\left[-\alpha\mathcal{E}(z_{k+1}^+)\right].$$

Moving $\mathcal{E}(z_{k+1}^+)$ to the left-hand side yields the desired result. □

When $\sigma = 0$, SHANG++ reduces to the deterministic HNAG++ method of Chen & Xu (2025). As $\sigma$ grows, convergence slows but acceleration is preserved. While Gupta et al. (2024) interpret noise as inflating smoothness to $(1 + \sigma^2)L$, our analysis shows it perturbs both smoothness and curvature, giving $L_\sigma = (1 + \sigma^2)L$ and $\mu_\sigma = \mu/(1 + \sigma^2)$. We compare the parameters

$$\text{(SHANG)} \quad 0 < \alpha \leq \sqrt{\frac{\mu_\sigma}{L_\sigma}} \qquad \text{(SHANG++)} \quad 0 < \alpha \leq \frac{1}{1 - \tilde{\alpha}}\sqrt{\frac{\mu_\sigma}{L_\sigma}},$$

The noise-damping term in SHANG++ further reduces the effective Lipschitz constant from $L_\sigma$ to $(1 - \tilde{\alpha})L_\sigma$ and increase the effective strongly convex constant from $\mu_\sigma$ to $\mu_\sigma/(1 - \tilde{\alpha})$, explaining its stronger stability.

**SHANG++ Method for Convex Minimization** Recall the modified step size $\tilde{\alpha}_k = \frac{\alpha_k}{1+m\alpha_k}$. To facilitate analysis, we define an auxiliary time-scaling variable $\tilde{\gamma}_k = \frac{\gamma_k}{1+m\alpha_k}$. Setting $\alpha_k = \frac{2}{k+1}$ and $\gamma_k/(1 + \sigma^2) = \alpha_k\tilde{\alpha}_k L_\sigma$, for any fixed $m \geq 0$, we obtain:

$$\frac{\tilde{\gamma}_{k+1} - \tilde{\gamma}_k}{\tilde{\alpha}_k} = -(1 + \frac{1}{2(k+1+2m)})\tilde{\gamma}_{k+1} \leq -\tilde{\gamma}_{k+1} \quad (2.13)$$

Replacing the $x$-update in (2.4) with the equivalent modified discretization (2.5) and combining it with (2.13) yields the following convergence result. The full proof appears in Appendix C.2.

**Theorem 2.2.** *Let $f \in \mathcal{S}_{0,L}$. Suppose that $(x_k, v_k)$ are generated by the time-stepping scheme (2.4). $g(x_k)$ defined in (1.2) and MNS holds. Given $x_0^+ = v_0 = x_0, m \geq 0$, choose the step size $\alpha_k = \frac{2}{k+1}$, $\gamma_k/(1 + \sigma^2) = \alpha_k\tilde{\alpha}_k L_\sigma$ and $\beta_k = \frac{\alpha_k}{\gamma_k/(1+\sigma^2)}$, we have*

$$\mathbb{E}\left[f(x_{k+1}^+) - f(x^\star) + \frac{\tilde{\gamma}_{k+1}}{2}\|v_{k+1} - x^\star\|^2\right] \leq \frac{(1+2m)(2+2m)}{(k+2+2m)(k+3+2m)}\mathcal{E}(z_0; \tilde{\gamma}_0) = \mathcal{O}(\frac{L_\sigma}{k^2})$$

We compare the parameters

$$\text{(SHANG)} \quad \frac{\gamma_k}{1+\sigma^2} = \alpha_k^2 L_\sigma, \qquad \text{(SHANG++)} \quad \frac{\gamma_k}{1+\sigma^2} = \alpha_k\tilde{\alpha}_k L_\sigma = \alpha_k^2 \cdot \frac{L_\sigma}{1+m\alpha_k},$$

which reduces the effective Lipschitz constant from $L_\sigma$ to $\frac{L_\sigma}{1+m\alpha_k}$. The noise-damping term offsets part of the $\sigma^2$–induced amplification, improving stability by slowing down the effective rate. Our experiments suggest that choosing $m$ in the range $[0, 1.5]$ provides a good trade-off.

**Other Convergence Results.** *Quadratic Loss.* Consider a special case of problem (1.1): the quadratic loss with Tikhonov regularization (also known as weight decay), which is widely used in regression tasks. The objective takes the form

$$f(x) = \frac{1}{N}\sum_{i=1}^{N}(x^\top X_i - Y_i)^2 + \frac{\lambda}{2}\|x\|_2^2 = \frac{1}{N}\|X^\top x - Y\|_2^2 + \frac{\lambda}{2}\|x\|_2^2, \tag{2.14}$$

where $\frac{1}{N}\sum_{i=1}^{N}(x^\top X_i - Y_i)^2$ is the empirical quadratic loss and $\frac{\lambda}{2}\|x\|_2^2$ is the regularizer with $\lambda > 0$. The Tikhonov regularizer ensures that the objective is $\lambda$–strongly convex with smoothness constant $(L + \lambda)$. Under multiplicative noise scaling, setting $\alpha = \frac{1}{1 - \tilde{\alpha}}\sqrt{\mu_\sigma/L_\sigma}$ yields the accelerated convergence rate $(1 - \frac{1}{1+\sigma^2}\sqrt{\lambda/(L + \lambda)})$ in the leading term.

*Batching.* Gradient noise can be reduced by increasing the mini-batch size $M$ in (1.2). If $\sigma_1^2$ is the MNS constant for $M = 1$, then $\sigma_M^2 = \sigma_1^2/M$. Another approach is to average $K$ independent gradient estimators, $g^K = \frac{1}{K}\sum_{i=1}^{K}g_i$, which gives an effective MNS constant of $\sigma^2/K$. Both strategies reduce noise at the cost of higher computation, and a straightforward analysis shows that averaging multiple estimates can accelerate convergence to some extent.

*Variance decay under MNS.* Beyond the expectation bound, we show geometric variance decay of the Lyapunov energy. Specifically, by Theorem D.1,

$$\mathrm{Var}\left(f_{-\mu}(x_k^+) - f_{-\mu}(x^\star) + \frac{\mu}{2}\|v_k - x^\star\|^2\right) \leq (f(x_0) - f(x^\star))^2(r^2 + K_2)^k.$$

A sufficient (practically verifiable) condition is $K_2 < 1 - r^2$, where $r = (1 + \alpha)^{-1}$ is the decay rate in Theorem 2.1 and $K_2$ collects the fluctuation constants. This holds, for example, in low-condition regime, with a damped stepsize $\alpha \leftarrow \delta\alpha$ ($0 < \delta \leq 1$) or with a minibatch of larger $M$ (or $K$ independent multiple estimates). Complete proofs and the explicit expressions of related constants are provided in Appendix D.

## 3 NUMERICAL EXPERIMENTS

We design our experiments to validate the theoretical alignment, scalability, and robustness of SHANG++ and SHANG ($m = 0$).

For deep learning tasks, we adopt SHANG++ with three explicit hyperparameters $(\alpha, \gamma, m)$, with $\mu = 0$ and $\beta = \alpha/\gamma$, summarized in Algorithm 1, where $v$ is updated first by index shifting. Here we fix $\beta = \alpha/\gamma$ to simplify tuning. Although theory suggests $\beta = (1 + \sigma^2)\alpha/\gamma$, estimating $\sigma$ is unreliable, and the fixed ratio provides stable performance with implicit noise scaling. Adaptive choices of $\sigma$ offered little practical improvement.

SHANG++ incurs no extra per-iteration cost compared with standard momentum methods: each update requires one gradient evaluation and a constant number of vector operations.

---
**Algorithm 1:** SHANG++ for Deep Learning

**Input:** Objective function $f$, initial point $x_0$, step size $\alpha$, time scaling factor $\gamma$, noise-damping $m$, , iteration horizon $T$.

$k \leftarrow 1$, $v_0 \leftarrow x_0$, $x_1 \leftarrow x_0$, $\tilde{\alpha} \leftarrow \frac{\alpha}{1+m\alpha}$

**while** $k \leq T$ **do**

$\quad g_k \leftarrow \frac{1}{M}\sum_{i\in B}\nabla f_i(x_k)$       `// stochastic gradient estimate`

$\quad v_k \leftarrow v_{k-1} - \frac{\alpha}{\gamma}g_k$

$\quad x_{k+1} \leftarrow \frac{1}{1+\tilde{\alpha}}x_k + \frac{\tilde{\alpha}}{1+\tilde{\alpha}}v_k - \frac{\tilde{\alpha}}{1+\tilde{\alpha}}\frac{\alpha}{\gamma}g_k$

$\quad k \leftarrow k + 1$

**end**

**return** $x_T$

---

Throughout this section, NAG refers to the stochastic version of Nesterov's accelerated gradient (Nesterov, 1983) by replacing $\nabla f(x)$ by $g(x)$. While SNAG refers to the method in (Hermant et al.,

2025), which can be treat as an alternative discretization of the HNAG flow (Appendix E). The stability of SNAG can be also explained with our theoretical analysis. Similarly SHB is the stochastic version of Heavy-Ball method (SGD with momentum).

**Convex optimization**   We first consider the family of objective functions from Gupta et al. (2024):

$$f_d : \mathbb{R} \to \mathbb{R}, \qquad f_d(x) = \begin{cases} |x|^d, & |x| < 1, \\ 1 + d(|x| - 1), & \text{else,} \end{cases}$$

for $d \geq 2$, with gradient estimators $g(x) = (1 + \sigma Z)\nabla f(x)$, where $Z \sim \mathcal{N}(0, I_d)$ is a standard normal random variable. The functions $f_d$ belong to $\mathcal{S}_{0,L}$ with $L = d(d-1)$.

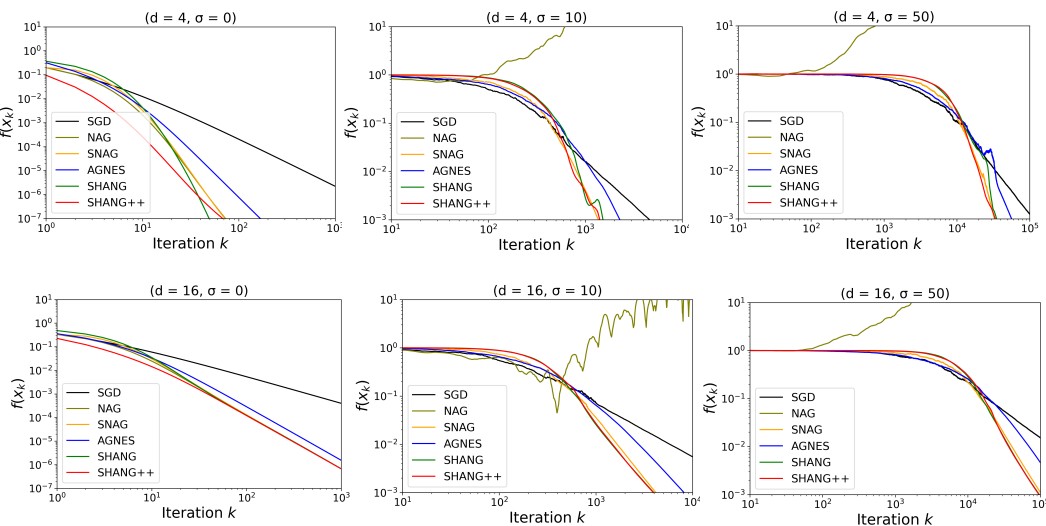

Figure 3.1: Performance of different algorithms under varying noise levels.

We compare SHANG and SHANG++ with SGD, NAG, AGNES (Gupta et al., 2024), and SNAG (Hermant et al., 2025) under $\sigma \in \{0, 10, 50\}$ and $d \in \{4, 16\}$. The parameters used follow their optimal choices for the convex case. All simulations are initialized at $x_0 = 1$, and expectations are averaged over 200 independent runs. See Appendix A.1 for the full experimental setup, hyperparameter choices, and results.

In Figure 3.1, both SHANG and SHANG++ remain stable as the noise level $\sigma$ increases, whereas NAG diverges under large noise. SHANG is generally very competitive, with SHANG++ showing consistently slightly better behavior than the other accelerated stochastic schemes. These results suggest that the proposed methods are reasonably robust to noisy gradients with modest tuning, while maintaining accelerated-like behavior in the high-noise regime.

**Classification Tasks on MNIST, CIFAR-10 and CIFAR-100**   We benchmark on three training tasks: LeNet-5 on MNIST (LeCun et al., 1998), ResNet-34 (He et al., 2016) on CIFAR-10 (Krizhevsky, 2009), and ResNet-50 on CIFAR-100. Each model is trained for 50 epochs, and results are reported as mean ± s.d. over five random seeds.

For hyperparameter selection, SHANG and SHANG++ used $\alpha = 0.5$ with $\gamma$ chosen from grids: $\{1, 1.5, 2\}$ for LeNet-5, $\{5, 10\}$ for ResNet-34, and $\{10, 15\}$ for ResNet-50. SHANG++ fixed $m = 1.5$. AGNES followed defaults $(\eta, \alpha, m) = (0.01, 0.001, 0.99)$; SNAG used $(\eta, \beta)$ with $\eta \in \{0.5, \dots, 0.001\}$, $\beta \in \{0.7, 0.8, 0.9, 0.99\}$, where $(0.05, 0.9)$ performed best, consistent with prior CIFAR work. Other baselines used $\eta = 0.001$ and momentum 0.99 when applicable. After 25 epochs, all baseline learning rates (including AGNES's correction) were decayed by 0.1, while $\gamma$ was doubled for our methods. Full details are in Appendix A.2.

Figure 3.2 reports SHANG with $(\alpha, \gamma) = (0.5, 10)$ and SHANG++ with $(\alpha, \gamma, m) = (0.5, 10, 1.5)$, while Figure 3.3 reports the corresponding results for $(\alpha, \gamma) = (0.5, 15)$ and $(\alpha, \gamma, m) = (0.5, 15, 1.5)$. Figure 3.2 shows the training and test losses of ResNet-34 on CIFAR-10 under

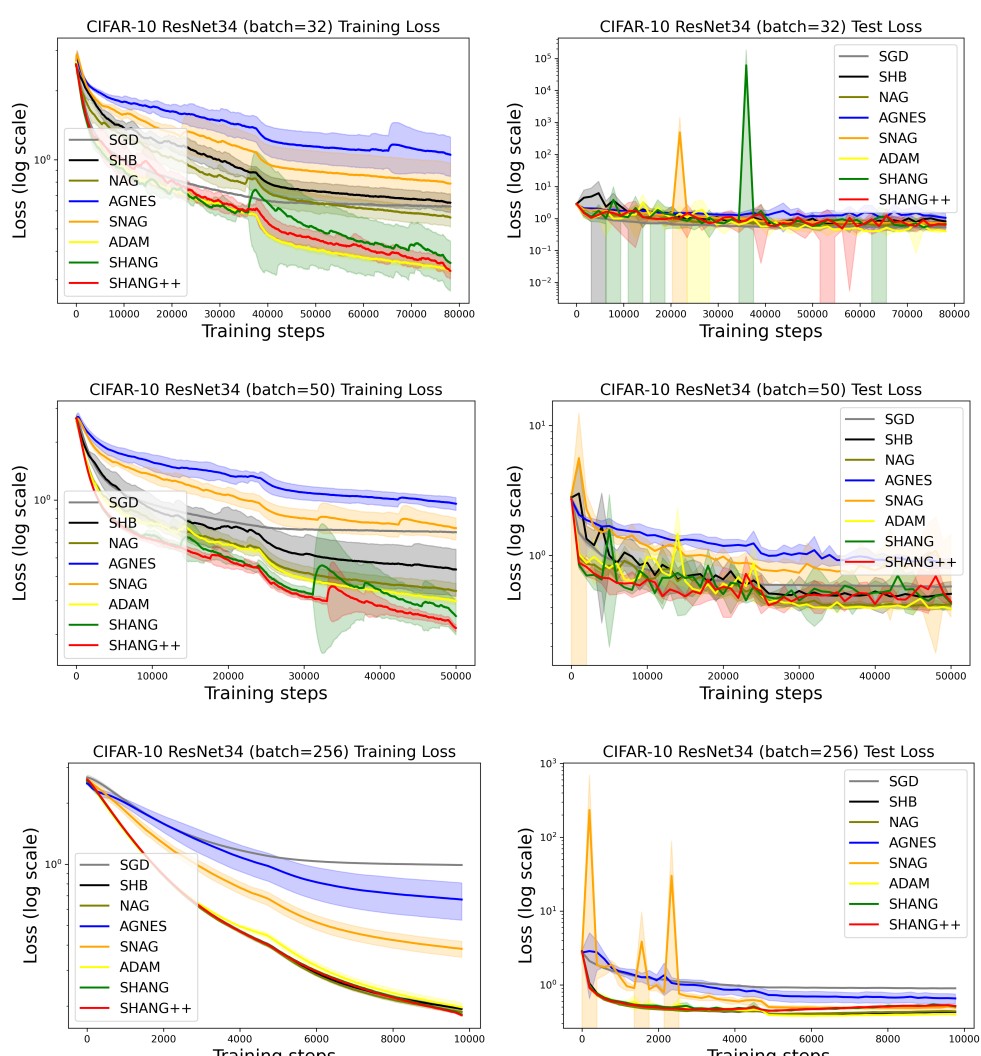

Figure 3.2: Training loss (left) and test loss (right) in log scale (running average with decay 0.99) on CIFAR-10 with ResNet-34, for batch sizes 32 (top row), 50 (middle row), and 256 (bottom row).

different batch sizes, with all algorithmic hyperparameters kept fixed across batch sizes. Batch size strongly affects gradient variance: smaller batches increase noise, larger batches reduce it. At 256, all methods are stable and gaps narrow; at 50, NAG, SNAG, and AGNES oscillate with wider bands (AGNES also plateaus higher). At batch size 32, differences among methods become more pronounced.

Even under extreme noise, SHANG and SHANG++ consistently outperform other first-order stochastic momentum methods. Notably, when the batch size falls below 50, AGNES and SNAG lose their acceleration advantage over SGD, whereas SHANG, SHANG++, and Adam still offer clear improvements (though Adam is not directly comparable). As also observed by Hermant et al. (2025), non–variance-reduced accelerated methods often lose acceleration at very small batch sizes; however, SHANG and SHANG++ appear to remain robust down to relatively smaller thresholds.

Figure 3.3 shows ResNet-50 training and test losses on CIFAR-100. SHANG and SHANG++ deliver competitive or superior performance to non-adaptive baselines. An interesting observation is that SGD attains the lowest test loss, yet this does not correspond to the best classification accuracy (see Figure A.3). This mismatch is aligned with prior findings: SGD on cross-entropy with hard labels is a likely cause of "confidently wrong" predictions (Thulasidasan et al., 2020). Table 3.1

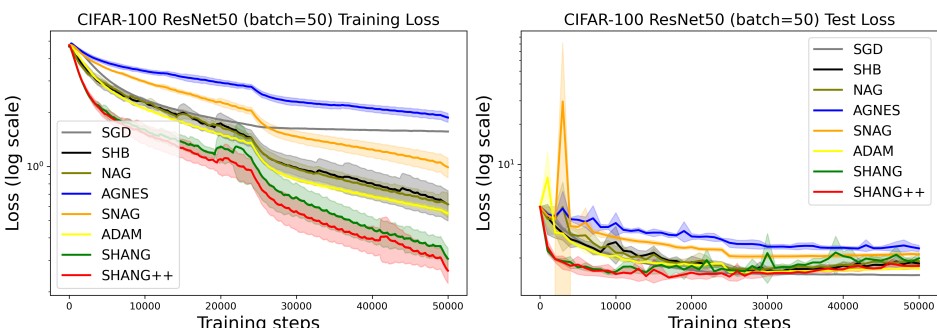

Figure 3.3: Training, test loss (log scale, running average with decay 0.99) on CIFAR-100 with ResNet-50 (batch size 50).

further summarizes the mean final test accuracy over five independent runs: SHANG and SHANG++ are comparable to Adam, often surpass AGNES and SNAG, and clearly improve over SGD and NAG. The slightly lower absolute accuracies arise because we use intentionally small batch sizes and only 50 training epochs to stress-test optimizer stability rather than to reach full convergence; with standard, longer training schedules, baselines attain their usual performance and the relative ranking of the methods remains essentially unchanged.

Table 3.1: Test accuracy of SGD, SHB, NAG, Adam, AGNES, SHANG, and SHANG++ on MNIST (LeNet-5), CIFAR-10 (ResNet-34), and CIFAR-100 (ResNet-50). Here $b$ is batch size.

|  | SGD | SHB | NAG | Adam | AGNES | SNAG | SHANG | SHANG++ |
|---|---|---|---|---|---|---|---|---|
| LeNet-5 | 91.07 | 98.98 | 98.9 | 99.07 | 98.88 | 99.07 | 99.06 | **99.11** |
| ($b = 50$) | ±0.11 | ±0.05 | ±0.08 | ±0.07 | ±0.09 | ±0.08 | ±0.02 | ±0.03 |
| ResNet-34 | 81.74 | 78.9 | 81.28 | **86.99** | 67.45 | 75.58 | 84.5 | 85.36 |
| ($b = 32$) | ±0.38 | ±1.67 | ±1.58 | ±0.14 | ±7.7 | ±6.02 | ±2 | ±1.42 |
| ResNet-34 | 79.91 | 84.59 | 86.43 | 87.38 | 70.49 | 77.65 | 87.15 | **87.4** |
| ($b = 50$) | ±0.11 | ±2.62 | ±0.81 | ±0.26 | ±2.51 | ±2.7 | ±0.82 | ±0.5 |
| ResNet-34 | 68.49 | 87.6 | 87.61 | **88.23** | 77.84 | 84.5 | 86.67 | 86.57 |
| ($b = 256$) | ±0.19 | ±0.27 | ±0.29 | ±0.11 | ±3.7 | ±0.92 | ±0.13 | ±0.17 |
| ResNet-50 | 58.31 | 58.17 | 57.66 | 59.87 | 42.82 | 49.51 | 63.31 | **65.02** |
| ($b = 50$) | ±0.51 | ±1.99 | ±1.44 | ±0.61 | ±1.24 | ±1.56 | ±0.93 | ±1.25 |

**Robustness to Multiplicative Gradient Noise**   Our theory predicts that time-scale coupling $(\alpha, \gamma)$ in SHANG and $(\alpha, \gamma, m)$ in SHANG++ mitigates multiplicative gradient noise. To test this, we fix one hyperparameter configuration per optimizer and evaluate across $\sigma \in \{0, 0.05, 0.1, 0.2, 0.5\}$. The effective noise is higher than nominal $\sigma$, since minibatch SGD adds sampling noise. This one-shot protocol isolates each optimizer's robustness without re-tuning. All experiments use CIFAR-10 with ResNet-34, batch size 50, the same settings as subsection 3, trained for 100 epochs and averaged over three seeds. Final validation error at epoch 100 is reported; full setup and hyperparameters are in Appendix A.4.

Figure 3.4 shows the mean final classification error rate under varying noise levels, and Table 3.2 reports the relative degradation $\Delta(\sigma) = (\mathbb{E}(\sigma) - \mathbb{E}(0))/\mathbb{E}(0)$, where $\mathbb{E}(\sigma)$ denotes the mean classification error rate (averaged over three seeds) at noise level $\sigma$.

1. At $\sigma = 0$, SHANG and SHANG++ reach 15.9%, outperforming SNAG (17.5%) and AGNES (20.5%).

2. At $\sigma = 0.1$, SHANG slightly improves to 15.6 %, SHANG++ remains stable at 15.9%, SNAG marginally improves to 17.1%, while AGNES degrades to 23.8%.

3. At $\sigma = 0.5$, SHANG and SHANG++ remain near 16%, while SNAG rises to 17.6% and AGNES drifts to 23.2% ($\approx$13.5% relative increase).

These results align with our Lyapunov analysis: time-scale coupling $(\alpha, \gamma, m)$ suppresses $\sigma^2$ amplification, ensuring stable performance without re-tuning. SNAG is stable but less accurate, while AGNES is most sensitive to noise.

Table 3.2: Relative change in final classification error compared with $\sigma = 0$ (lower is better; negative values indicate improvement). Values are averaged over three seeds.

| Method | Relative degradation $\Delta(\%)$ at $\sigma$ | | | |
| --- | --- | --- | --- | --- |
| | 0.05 | 0.1 | 0.2 | 0.5 |
| SHANG | $-2.5$ | $-2.1$ | $-1.0$ | $-0.2$ |
| SHANG++ | $+3.4$ | $-0.6$ | $-2.1$ | $-0.9$ |
| AGNES | $-14.4$ | $+16.0$ | $+14.6$ | $+13.5$ |
| SNAG | $-2.0$ | $-2.1$ | $-5.0$ | $0.7$ |

Figure 3.4: Validation error under varying multiplicative noise level $\sigma$. Lower is better.

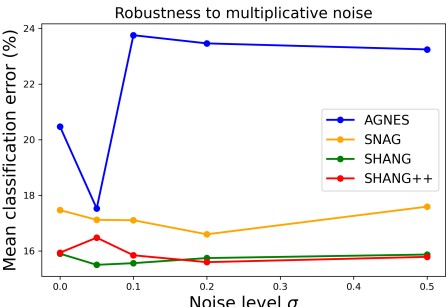

**Image Reconstruction with Small Batch Size** We further evaluate our algorithms on a generative task of image reconstruction with small-batch training, using a lightweight U-Net (Ronneberger et al., 2015) on CIFAR-10 with batch size 5. SHANG and SHANG++ are compared against SNAG, AGNES, NAG, SGD, SHB, and Adam, with full experimental details provided in the appendix A.6. Figure 3.5 shows training and test losses. Adam achieves the lowest loss due to its adaptive learning

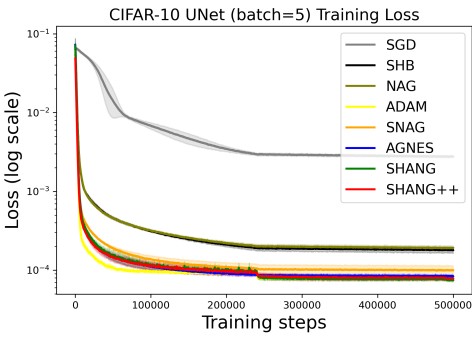
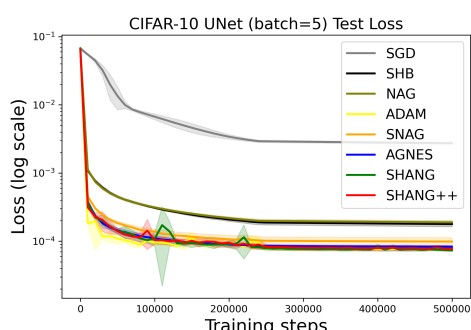

Figure 3.5: Training and test loss (log scale, running average with decay 0.99) on CIFAR-10 using U-Net with batch size 5.

rate, but both SHANG and SHANG++ outperform all other non-adaptive methods. In particular, SHANG++ shows stable and efficient training even in this high-noise regime, highlighting its practical robustness. We additionally include a sanity-check experiment on ImageNet-100 with ResNet-34 in Appendix A.5, which shows that SHANG and SHANG++ remain competitive with classical momentum methods on this larger-scale task, and we also conduct a comparative hyperparameter study, with full settings and results given in Appendix A.7.

## 4 CONCLUSION

We presented SHANG++, an accelerated first-order stochastic optimizer for robust and simple training under multiplicative noise. Theoretically, it retains the optimal accelerated rate in both convex and strongly convex settings under the MNS condition. Empirically, across convex tasks, image classification, and generative reconstruction, one-shot hyperparameter choices sustain near noise-free accuracy (within $1\%$ for $\sigma \leq 0.5$). Compared with other stochastic momentum methods, SHANG++ demonstrates enhanced stability under small-batch or high-noise conditions, with accuracy exceeding baselines and comparable to Adam. These properties make SHANG++ a practical, scalable optimizer for large-scale, noise-intensive training. Its empirical success on nonconvex problems further suggests that extending the theory beyond convexity is a natural next step.

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

## LLM USAGE

In preparing this manuscript, large language models (LLMs) were employed exclusively to assist with language-related tasks, such as improving readability, grammar, and style. The models were not used for research ideation, development of methods, data analysis, or interpretation of results. All scientific content, including problem formulation, theoretical analysis, and experimental validation, was conceived, executed, and verified entirely by the authors. The authors bear full responsibility for the accuracy and integrity of the manuscript.

## ETHICS STATEMENT

This work is purely theoretical and algorithmic, focusing on convex optimization methods. It does not involve human subjects, sensitive data, or applications that raise ethical concerns related to privacy, security, fairness, or potential harm. All experiments are based on publicly available datasets or synthetic data generated by standard procedures. The authors believe that this work fully adheres to the ICLR Code of Ethics.

## REPRODUCIBILITY STATEMENT

We have taken several measures to ensure the reproducibility of our results. All theoretical assumptions are explicitly stated, and complete proofs are provided in the appendix. For the experimental evaluation, we describe the setup, parameter choices, and baselines in detail in the main text. The source code for our algorithms and experiments are available as supplementary materials. Together, these resources should allow others to reproduce and verify our theoretical and empirical findings.

# A SUPPLEMENT OF EXPERIMENTS

Here are some experimental setup and results that are not presented in the main text.

## A.1 SUPPLEMENT OF THE CONVEX EXPERIMENT

For the convex example in Section 3, we compare SHANG and SHANG++ with SGD, NAG, AGNES, and SNAG under $\sigma \in \{0, 10, 50\}$ and $d \in \{4, 16\}$. The parameters used follow their optimal choices for the convex case. For SHANG, $\alpha_k = \frac{2}{k+1}$, $\gamma_k = \alpha_k^2 L(1+\sigma^2)^2$ and $\beta_k = \frac{(1+\sigma^2)\alpha_k}{\gamma_k}$; For SHANG++, $\alpha_k = \frac{2}{k+1}$, $m = 1.5$, $\gamma_k = \frac{\alpha_k^2}{1+m\alpha_k}(1+\sigma^2)^2 L$ and $\beta_k = \frac{(1+\sigma^2)\alpha_k}{\gamma_k}$; For AGNES, we adopted the best-performing parameters reported by the authors for this problem: learning rate $\eta = \frac{1}{L(1+2\sigma^2)}$, correction step size $\alpha = \frac{\eta}{1+\sigma^2}$, and momentum $m_k = \frac{k}{k+5}$. For SNAG, we use $s = \frac{1}{L(1+\sigma^2)}$, $\eta_k = \frac{1}{L(1+\sigma^2)^2}\frac{k+1}{2}$, $\beta = 1$, $\alpha_k = \frac{k^2/(k+1)}{2+(k^2/(k+1))}$. For NAG, we used a learning rate of $\frac{1}{L(1+\sigma^2)}$ and momentum parameter of $\frac{k}{k+3}$. SGD was also run with a learning rate of $\frac{1}{L(1+\sigma^2)}$. All hyperparameter notations match those used in the original publications; note, however, that symbol meanings may vary across algorithms (e.g., $\alpha$ denotes the discretization step size in SHANG, while in AGNES it refers to the correction step size). All simulations are initialized at $x_0 = 1$, and expectations are averaged over 200 independent runs.

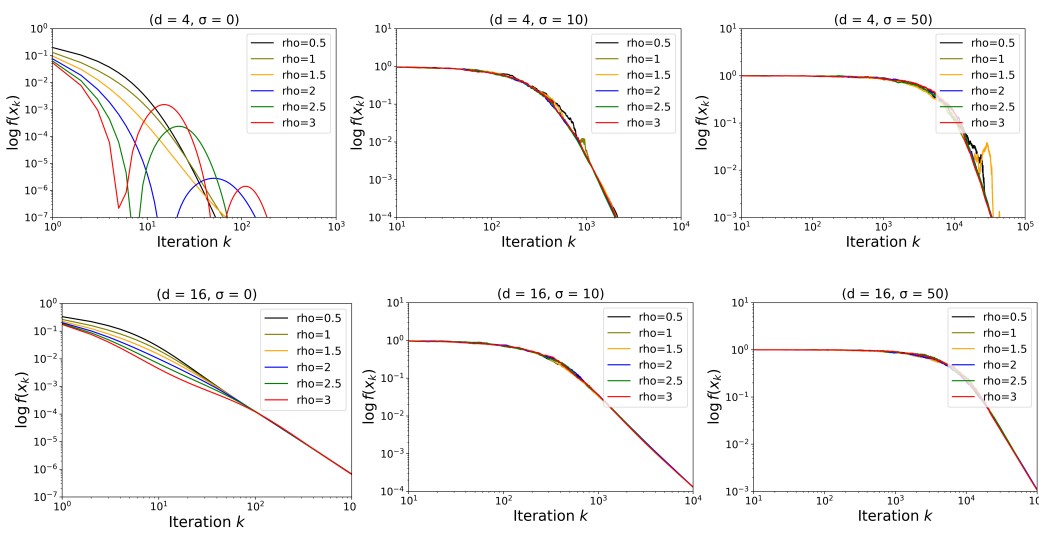

Figure A.1: Log-log plots of $\mathbb{E}[f_d(x_k)]$ for SHANG++ using $m = 0.5$ (black), $m = 1$ (olive), $m = 1.5$ (orange), $m = 2$ (blue), $m = 2.5$ (green), $m = 3$ (red) with $d = 4$ (Top Row) and $d = 16$ (Bottom Row), under noise levels $\sigma = 0$ (Left Column), $\sigma = 10$ (Middle Column) and $\sigma = 50$ (Right Column). From the figures, it can be observed that $m \leq 1.5$ provides a good choice.

Figure A.1 highlights SHANG++'s stability across $m$: values $m \leq 1.5$ consistently yield strong performance. Our theoretical variance-decay predictions directly manifest in practice.

## A.2 SUPPLEMENT OF CLASSIFICATION TASKS

**Setup.** We benchmark SHANG, SHANG++, Adam, SNAG, AGNES, NAG, SHB (or SGD with momentum) and SGD on the following tasks: training LeNet-5 on the MNIST dataset, training ResNet-34 on the CIFAR-10 image dataset and training ResNet-50 on the CIFAR-100 dataset with standard data augmentation (normalization, random crop, and random flip). All models have pretrain set to True. For each dataset, we run all algorithms for 50 epochs with batch size 50 and report averages over five trials. After 25 epochs, the learning rates for all baseline methods (excluding SHANG and SHANG++) are decayed by a factor of 0.1; AGNES's correction step size is similarly reduced. For our methods, the time-scaling factor $\gamma$ is doubled after 25 epochs. This learning-rate

schedule follows Gupta et al. (2024) and helps the baselines achieve better performance on deep learning tasks. SHANG and SHANG++ do not use an explicit learning rate; their effective learning rate is controlled by the time-scaling parameter $\gamma$, with effective learning rate $1/\gamma$ (see Algorithm 1). To implement an analogous decay, we increase $\gamma$ after 25 epochs (thereby reducing the effective step size $1/\gamma$), so that all methods undergo a comparable mid-training learning-rate reduction.

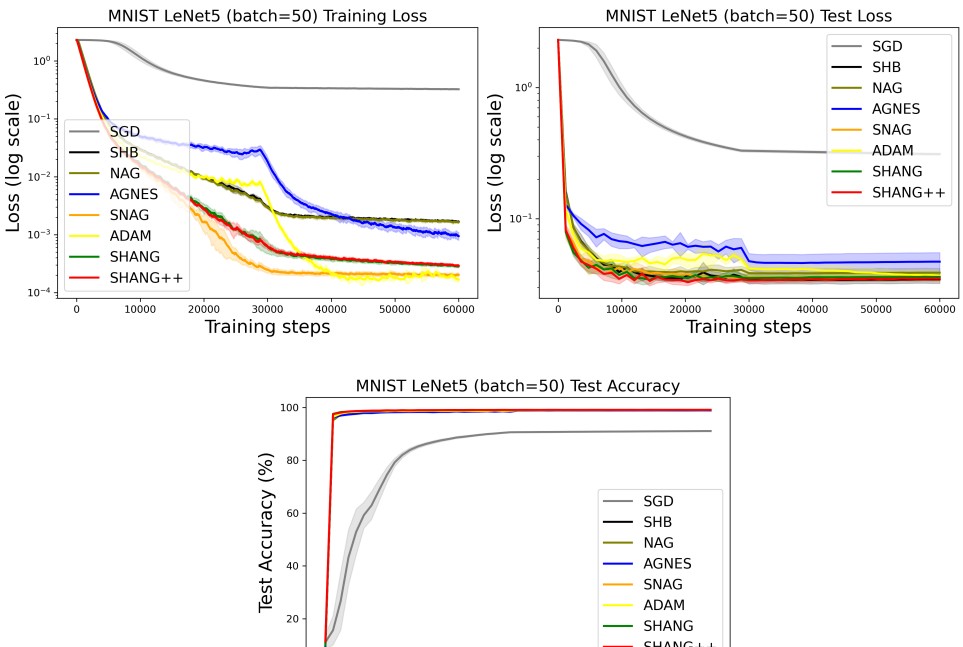

Figure A.2: Training loss (log scale) (left), test loss (log scale) (middle) as a running average with decay rate 0.99, and test accuracy (right) on the MNIST dataset using LeNet-5 trained with batch size 50. The compared methods include SGD (gray), SHB (black), NAG (olive), AGNES (blue), SNAG (orange), Adam (yellow), SHANG (green) and SHANG++ (red). In SHANG, $(\alpha, \gamma) = (0.5, 2)$ and in SHANG++, $(\alpha, \gamma, m) = (0.5, 2, 1.5)$.

For hyperparameter selection, our two methods were evaluated under three settings: $\alpha = 0.5$ with $\gamma \in \{1, 1.5, 2\}$ for LeNet-5, $\gamma \in \{5, 10\}$ for ResNet-34 and $\gamma \in \{10, 15\}$ for ResNet-50. For SHANG++, we fixed $m = 1.5$. AGNES employed the default parameter configuration recommended by its authors, $(\eta, \alpha, m) = (0.01, 0.001, 0.99)$, which has demonstrated strong performance across various tasks. For SNAG, we adopt the two-parameter variant $(\eta, \beta)$ proposed by the original authors for machine-learning tasks. Hyperparameters are selected via a grid search, learning rate $\eta \in \{0.5, 0.1, 0.05, 0.01, 0.005, 0.001\}$ and momentum $\beta \in \{0.7, 0.8, 0.9, 0.99\}$. Among these, $(\eta, \beta) = (0.05, 0.9)$ yields the best performance, which coincides with the parameter choice recommended by the original authors for training CNNs on the CIFAR dataset. All other baseline algorithms used a fixed learning rate of $\eta = 0.001$; for those involving momentum, the momentum coefficient was set to 0.99.

**Results.** Figures A.2, A.3, A.4 and A.5 depict the evolution of training/test loss and test accuracy across datasets. Overall, SHANG and SHANG++ achieve competitive or superior performance compared with non-adaptive baselines.

### A.3 BATCH-SIZE SCALING ON CIFAR-10 (RESNET-34)

To further assess the robustness of our algorithms to stochastic gradient noise, we evaluate all methods on CIFAR-10 with ResNet-34 under three batch-size settings: 32, 50 and 256. Smaller batches introduce higher gradient variance, whereas larger batches reduce the noise level. Importantly,

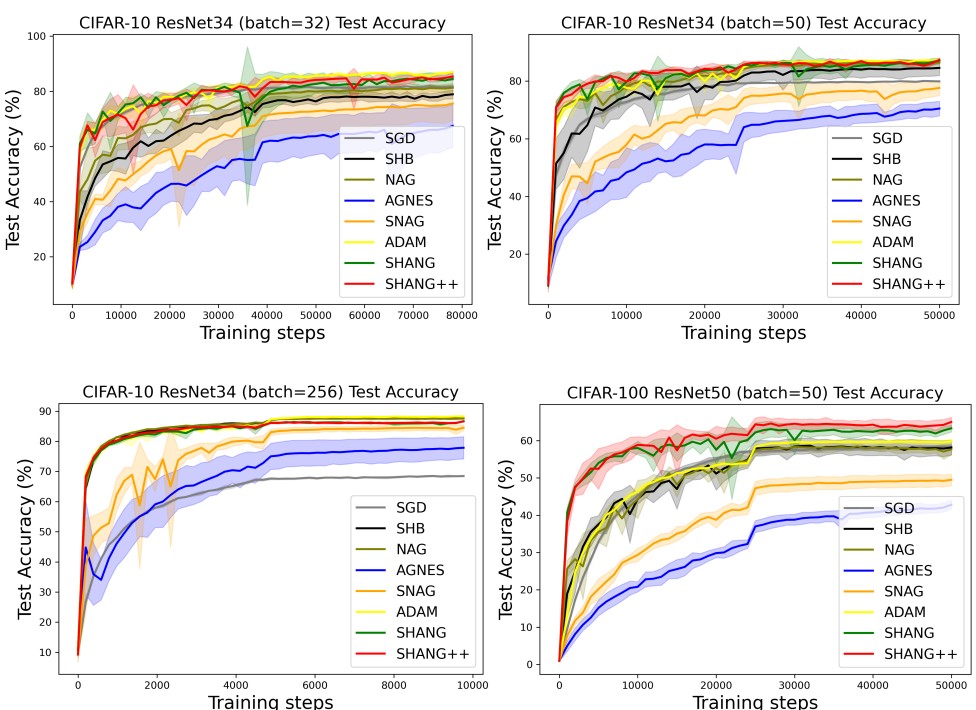

Figure A.3: Test accuracy on CIFAR10 with ResNet-34 and CIFAR-100 with ResNet-50.

all hyperparameters are kept fixed across batch sizes to isolate the effect of noise on algorithmic performance.

**Setup.** All data augmentation and experiments setting follows Appendix A.2. Hyperparameters are held fixed across batch sizes: for SHANG/SHANG++ we use $(\alpha, \gamma) = (0.5, 10)/(\alpha, \gamma, m) = (0.5, 10, 1.5)$, and all baselines reuse their best settings from Appendix 3. No re-tuning is performed when switching the batch size.

**Results.** Figure 3.2 shows the training/test dynamics.

- *Small batch (32).* Under the smallest batch size, classical momentum variants SHB, SNAG and AGNES exhibit a clear loss of acceleration relative to SGD, while SHANG and SHANG++ consistently retain accelerated convergence.

- *Small batch (50).* NAG, SNAG and AGNES exhibit larger oscillations and wider variance bands; AGNES also shows a higher error plateau. In contrast, SHANG/SHANG++ produce the lowest losses among non-adaptive methods and maintain narrow shaded regions, indicating markedly improved stability across seeds. Adam remains competitive in accuracy but with higher variance in test loss.

- *Large batch (256).* The gap between methods narrows: all optimizers become more stable and the curves cluster. SHANG/SHANG++ continue to match the best-performing baselines while preserving smooth convergence.

Robustness to multiplicative noise translates into tangible benefits in the small-batch regime: with a single, fixed hyperparameterization ($\alpha = 0.5, \gamma = 10, m = 1.5$), SHANG/SHANG++ achieve stable training and strong test accuracy without re-tuning, whereas competing momentum methods are more sensitive (larger variance, higher plateaus). As batch size increases, all methods stabilize and the performance gap diminishes, consistent with the noise-abatement expected from larger batches.

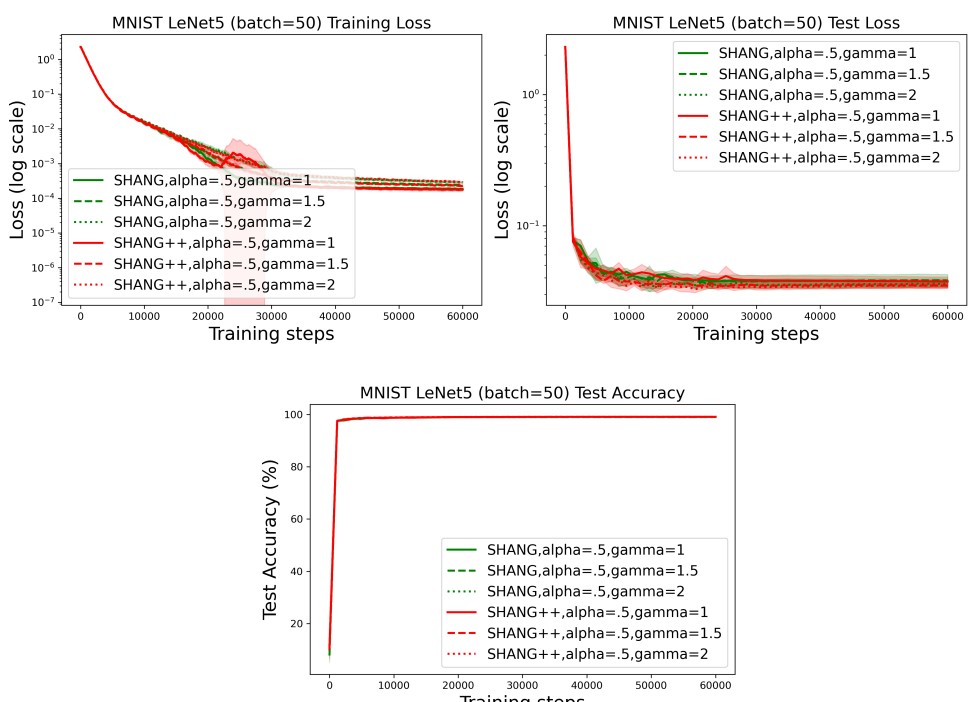

Figure A.4: Training loss (log scale), test loss (log scale) as a running average with decay rate $0.99$, and test accuracy on the MNIST dataset using LeNet-5 trained with batch size $50$. The compared methods include SHANG (green) and SHANG++ (red) under different parameter choices.

## A.4 SUPPLEMENT OF ROBUSTNESS TO MULTIPLICATIVE GRADIENT NOISE

All runs use an identical experimental setup: CIFAR-10 dataset, ResNet-34, batch size $50$, trained for 100 epochs, and averaged over three random seeds. Once initialized, no hyperparameters were adjusted or re-tuned during the experiments. This fixed-parameter setup allows us to isolate the effect of increasing multiplicative noise and directly observe each optimizer's inherent stability. Specifically, SHANG with $(\alpha = 0.5, \gamma = 10)$, SHANG++ with $(\alpha = 0.5, \gamma = 10, m = 1.5)$, AGNES with $(\eta = 0.01, \alpha = 0.001, m = 0.99)$ and SNAG with $(\eta = 0.05, m = 0.9)$. Note that the actual gradient noise level experienced by the optimizer is higher than the nominal $\sigma$, because minibatch stochastic gradient descent inherently introduces sampling noise. The multiplicative noise we introduce,

$$g(x_k) = (1 + \sigma\mathcal{N}(0, I_d))\nabla f(x_k),$$

is therefore imposed on top of this intrinsic minibatch stochasticity. We record the final validation error at epoch $100$.

**Discussion.** The empirical trends align with our Lyapunov analysis: coupling the time scales $(\alpha, \gamma, m)$ suppresses the $\sigma^2$ amplification and yields stable behavior across noise levels without retuning. SNAG—while reasonably stable—does not match the consistently low error of SHANG/SHANG++, and AGNES is the most sensitive to increased multiplicative noise.

## A.5 ADDITIONAL CLASSIFICATION TASK ON IMAGENET-100 WITH RESNET-34

We further evaluate all methods on the ImageNet-100 (Deng et al., 2009) subset using ResNet-34 with input size $224 \times 224$ and batch size $64$. We adopt the standard ImageNet data augmentation: random resized crops to $224 \times 224$ with scale in $[0.08, 1.0]$, random horizontal flips, and normalization with the ImageNet mean and standard deviation. The model is trained for $40$ epochs. For hyperparameter selection, SHANG uses $(\alpha = 0.5, \gamma = 3)$ and SHANG++ uses $(\alpha = 0.5, \gamma = 3, m = 1)$. AGNES follows the default $(\eta = 0.01, \alpha = 0.001, m = 0.99)$; SNAG uses $(\eta = 0.05, \beta = 0.9)$. Other

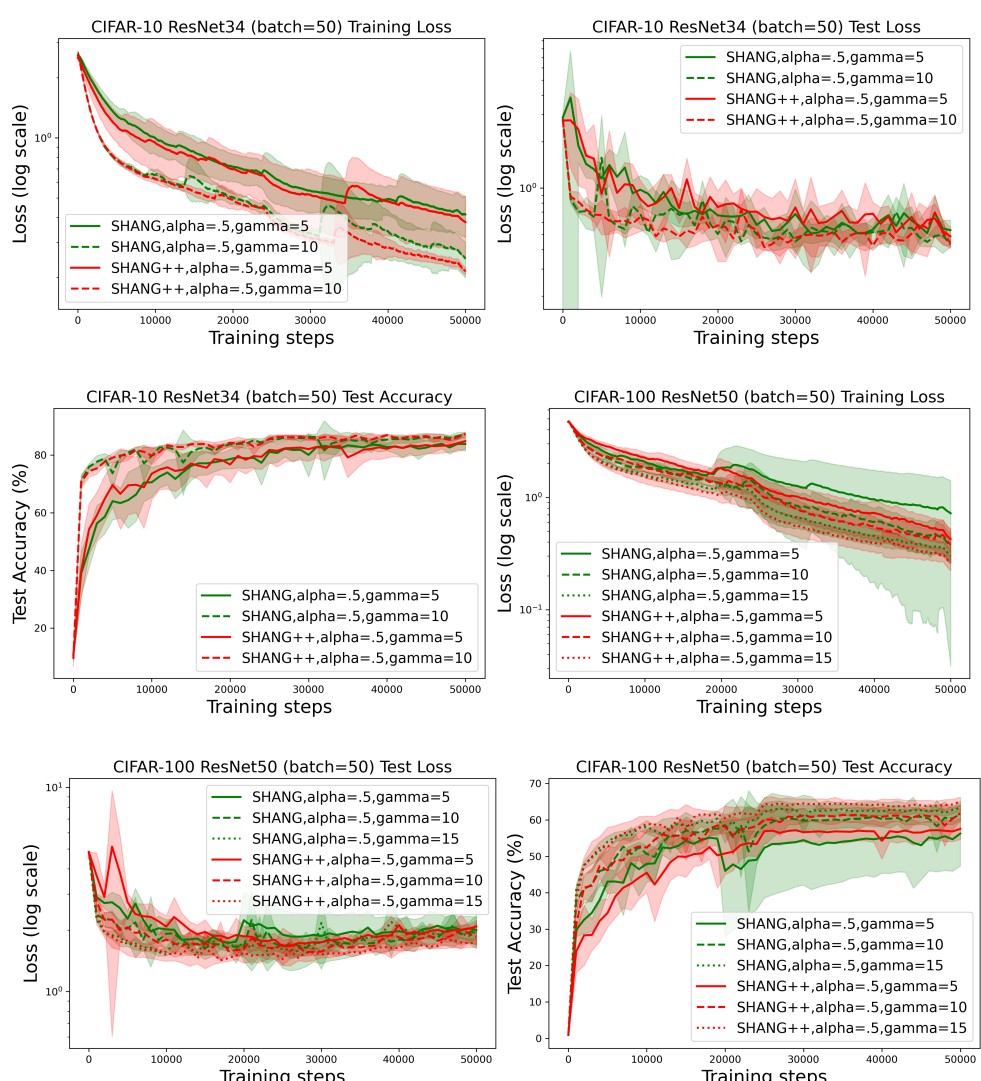

Figure A.5: Training loss (log scale), test loss (log scale) as a running average with decay rate $0.99$, and test accuracy on the MNIST dataset usingCIFAR-10 dataset using ResNet-34 and CIFAR-100 dataset using ResNet-50 trained with batch size $50$. The compared methods include SHANG (green) and SHANG++ (red) under different parameter choices.

baselines use $\eta = 0.001$ and momentum $0.99$ when applicable. After $25$ epochs, all baseline learning rates (including AGNES's correction) are decayed by a factor of $0.1$, while $\gamma$ is doubled for our methods. Due to computational constraints, we report a single representative run.

Figure A.6 shows the training/test loss and test accuracy. SHANG and SHANG++ achieve test losses comparable to the best classical momentum baselines (SHB and NAG), while clearly outperforming AGNES, SNAG, and Adam. In terms of final test accuracy, SHANG and SHANG++ reach about $98.1\%$, within roughly $0.4$ percentage points of SHB ($98.49\%$) and NAG ($98.53\%$). This is unsurprising: on this relatively benign ImageNet-100 setup with a moderate batch size, all well-tuned momentum methods behave very similarly, and classical SHB and NAG are known to be extremely strong baselines. Our goal here is not to dominate them in this regime, but to demonstrate that SHANG and SHANG++ remain fully competitive on a larger-scale task. Their main advantages appear in the noisier, small-batch regimes (e.g., CIFAR-10 and U-Net) highlighted in the main text, where classical momentum becomes less stable.

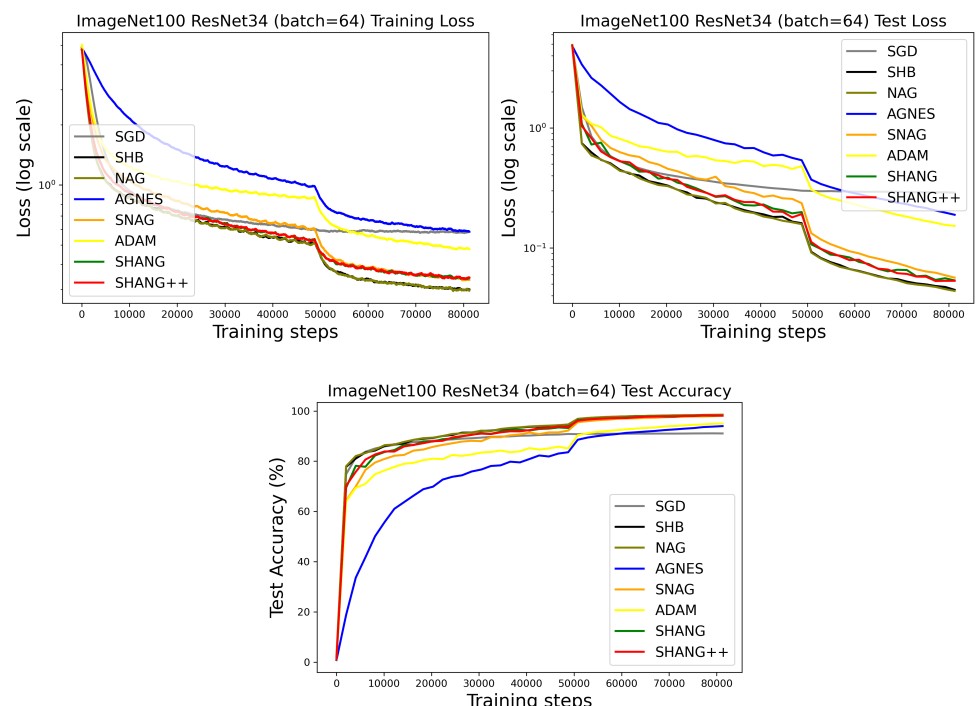

Figure A.6: Training loss (log scale), test loss (log scale) as a running average with decay rate $0.99$, and test accuracy on the ImageNet-100 dataset using ResNet-34 trained with batch size $64$.

### A.6 SUPPLEMENT OF IMAGE RECONSTRUCTION

We further evaluate our algorithms on a generative task—image reconstruction with small-batch training, which introduces substantial gradient noise. Specifically, we train a lightweight U-Net (Ronneberger et al., 2015) (base channels $32 \rightarrow 64 \rightarrow 128$, with bilinear up-sampling and feature concatenation) on CIFAR-10 using batch size $5$. We compare SHANG ($\alpha = 0.5, \gamma = 0.5$) and SHANG++ ($\alpha = 0.5, \gamma = 0.5, m = 1$) against SNAG, AGNES, NAG, SGD, SHB, and Adam. All other experimental settings follow those in earlier sections.

### A.7 HYPERPARAMETER COMPARISON

To identify optimal hyperparameter configurations for our stochastic algorithms, we perform grid searches over $\alpha \in (0.005, 0.1)$ and $\gamma \in (0.5, 30)$ on MNIST and CIFAR-10 (Figures A.7). For SHANG++, we additionally vary $m \in (0.5, 3)$ while keeping $\alpha = 0.5$ fixed. Results indicate that: (1) $\alpha = 0.5$ and $m = 1.5$ are generally effective across tasks; (2) Smaller $\gamma$ values work well for LeNet-5, while larger $\gamma$ are preferred for deeper networks like ResNet-34; (3) SHANG++ exhibits low sensitivity to $m$ in practice, with performance remaining stable across tested values. These findings confirm the practical usability and tuning simplicity of our methods.

## B SHANG

### B.1 MODEL

Applying a Gauss-Seidel-type scheme to discretize HNAG flow (2.3) and replace the deterministic gradient $\nabla f(x_k)$ with its unbiased stochastic estimate $g(x_k)$, we can obtain the Stochastic Hessian-

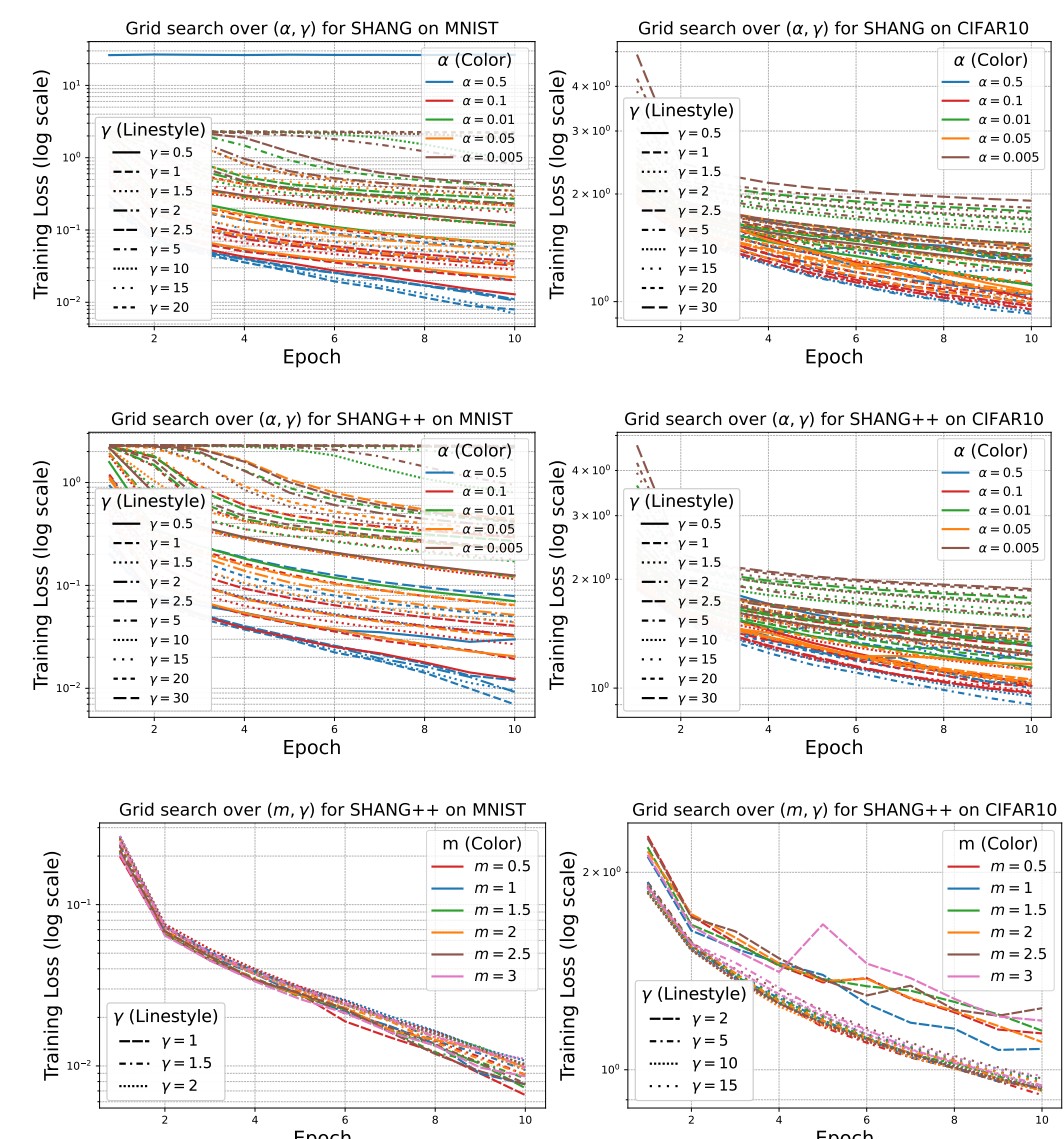

Figure A.7: Training loss (log scale) on the MNIST dataset using LeNet-5 (Left column) and CIFAR-10 dataset using ResNet-34 (Right column) trained with batch size 50. The plots show results for SHANG (top row) and SHANG++ (middle row) under different combinations of hyperparameters $\alpha \in \{0.1, 0.5, 0.01, 0.05, 0.005\}$ (different color) and $\gamma \in \{0.5, 1, 1.5, 2, 2.5, 5, 10, 15, 20\}$ (different line style). The left two figures show that $\alpha = 0.5$ and $\gamma \in \{1, 1.5, 2\}$ are relatively good parameter choices. The plots in bottom row illustrate the performance of the SHANG++ method under different combinations of $\gamma \in \{1, 1.5, 2\}$ (on MNIST dataset), $\gamma \in \{2, 5, 10, 15\}$ (on CIFAR-10 dataset) and $m \in \{0.5, 1, 1.5, 2, 2.5, 3\}$ with $\alpha$ fixed at $0.5$. The differences among various $m$ values are minor for this task. In practice, we typically choose $m = 1.5$. When using a very small batch size, $m$ can be appropriately reduced.

driven Nesterov Accelerated Gradient (SHANG) method:

$$\frac{x_{k+1} - x_k}{\alpha_k} = v_k - x_{k+1} - \beta_k g(x_k)$$

$$\frac{v_{k+1} - v_k}{\alpha_k} = \frac{\mu}{\gamma_k}(x_{k+1} - v_{k+1}) - \frac{1}{\gamma_k}g(x_{k+1}) \tag{B.1}$$

$$\frac{\gamma_{k+1} - \gamma_k}{\alpha_k} = \mu - \gamma_{k+1}$$

In the strongly convex case, we fix $\gamma = \mu$ and use a constant step size $\alpha$; in general case, we set $\mu = 0$ and allow both $\alpha_k$ and $\gamma_k$ to vary. The coupling $\beta_k > 0$ depends on $(\alpha_k, \gamma_k)$ and typically scales as $(1 + \sigma^2)\alpha_k/\gamma_k$. Consequently, SHANG reduces to a two-parameter scheme $(\alpha, \beta)$ in the strongly convex regime and a three-parameter scheme $(\alpha, \gamma, \beta)$ otherwise. For practical tuning, tying $\beta$ to $\alpha$ and $\gamma$ via $\beta = \alpha/\gamma$ yields an effective two-parameter $(\alpha, \gamma)$ algorithm. The SHANG method for deep learning tasks is described in Algorithm 2.

---

**Algorithm 2:** SHANG for Deep Learning

**Input:** Objective function $f$, initial point $x_0$, stepsize $\alpha$, time scaling factor $\gamma$, iteration horizon $T$.

$n \leftarrow 0, \quad v_0 \leftarrow x_0, \quad x_1 \leftarrow x_0$

**while** $k < T$ **do**

$\quad g_k \leftarrow \nabla f(x_k)$                                    `// gradient estimate`

$\quad v_k = v_{k-1} - \frac{\alpha}{\gamma} g_k$

$\quad x_{k+1} = \frac{1}{1+\alpha} x_k + \frac{\alpha}{1+\alpha} v_k - \frac{\alpha}{1+\alpha} \frac{\alpha}{\gamma} g_k$

$\quad k \leftarrow k + 1$

**end**

**return** $x_T$

---

Observe that SHANG is the $m = 0$ special case of SHANG++. Table B.1 summarizes the theoretical convergence complexities and the number of tunable parameters required by leading stochastic optimization methods under multiplicative noise. As shown, SHANG and SHANG++ achieve optimal theoretical guarantees while significantly reducing hyperparameter complexity.

Table B.1: Assume $f$ is $L$-smooth and $g(x)$ satisfies the multiplicative noise scaling (MNS) condition (see Definition 1.1) with constant $\sigma \geq 0$. This table summarizes the iteration complexity of leading first-order stochastic optimization algorithms under optimal parameter settings to reach $\varepsilon$-precision.

| Algorithm | Convex | Strongly Convex |
|---|---|---|
| SGD (Hermant et al., 2025) | $(1 + \sigma^2)\frac{L}{\varepsilon}$ | $(1 + \sigma^2)\frac{L}{\mu} \log(\frac{1}{\varepsilon})$ |
| NAG (Gupta et al., 2024) | $\sqrt{\frac{1+\sigma^2}{1-\sigma^2}}\sqrt{\frac{L}{\varepsilon}}$ | $\sqrt{\frac{1+\sigma^2}{1-\sigma^2}}\sqrt{\frac{L}{\mu}} \log(\frac{1}{\varepsilon})$ |
| AGNES (Gupta et al., 2024) | $\sqrt{\frac{L(1+2\sigma^2)(1+\sigma^2)}{\varepsilon}}$ | $(1 + \sigma^2)\sqrt{\frac{L}{\mu}} \log(\frac{1}{\varepsilon})$ |
| SNAG (Hermant et al., 2025) | $(1 + \sigma^2)\sqrt{\frac{L}{\varepsilon}}$ | $(1 + \sigma^2)\sqrt{\frac{L}{\mu}} \log(\frac{1}{\varepsilon})$ |
| SHANG | $(1 + \sigma^2)\sqrt{\frac{L}{\varepsilon}}$ | $\mathbf{(1 + \sigma^2)}\sqrt{\frac{\mathbf{L}}{\mu}} \log(\frac{\mathbf{1}}{\varepsilon})$ |
| SHANG++ | $(1 + \sigma^2)\sqrt{\frac{L}{\varepsilon}}$ | $\textcolor{red}{\mathbf{(1 + \sigma^2)}\sqrt{\frac{\mathbf{L}}{\mu}} \log(\frac{1}{\varepsilon})}$ |

## B.2 Convergence Analysis for SHANG

Define the discrete Lyapunov function

$$\mathcal{E}(z_k^+; \gamma_k) = f(x_k^+) - f(x^\star) + \frac{\gamma_k}{2} \|v_k - x^\star\|^2 \tag{B.2}$$

where $z_k^+ = (x_k^+, v_k)$, $z_k = (x_k, v_k)$ and $z^* = (x^\star, x^\star)$. The following theorem establishes a decay bound for $\mathbb{E}\left[\mathcal{E}(z_k^+; \gamma_k)\right]$.

**Theorem B.1.** *Let $f \in \mathcal{S}_{\mu,L}$, $(x_k, v_k)$ be generated by SHANG (B.1). $x_k^+ = x_k - \alpha_k \beta_k g(x_k)$ is an auxiliary variable. Assume $g(x)$ (defined in (1.2)) satisfies the MNS condition with constant $\sigma$. Given $x_0^+ = v_0 = x_0$,*

*(1) When $0 < \mu < L < \infty$, choose step size $0 < \alpha \leq \frac{1}{1+\sigma^2}\sqrt{\frac{\mu}{L}}$ and $\beta = \frac{(1+\sigma^2)\alpha}{\mu}$, we have*

$$\mathbb{E}\left[f(x_{k+1}^+) - f(x^\star) + \frac{\mu}{2}\|v_{k+1} - x^\star\|^2\right] \leq (1+\alpha)^{-(k+1)}\mathcal{E}_0^\mu$$

*(2) When $\mu = 0$, choose $\alpha_k = \frac{2}{k+1}$, $\gamma_k = \alpha_k^2(1+\sigma^2)^2 L$ and $\beta_k = \frac{(1+\sigma^2)\alpha_k}{\gamma_k}$, we have*

$$\mathbb{E}\left[f(x_{k+1}^+) - f(x^\star) + \frac{\gamma_{k+1}}{2}\|v_{k+1} - x^\star\|^2\right] \leq \frac{2}{(k+2)(k+3)}\mathcal{E}_0^{\gamma_0} = \mathcal{O}(\frac{1}{k^2})$$

*where $\mathcal{E}_0^\mu = f(x_0) - f(x^\star) + \frac{\mu}{2}\|x_0 - x^\star\|^2$ and $\mathcal{E}_0^{\gamma_0} = f(x_0) - f(x^\star) + \frac{\gamma_0}{2}\|x_0 - x^\star\|^2$.*

When $\sigma = 0$, SHANG reduces to the deterministic HNAG method analyzed in Chen & Luo (2021).

Before presenting the proof of Theorem B.1, we first establish several auxiliary lemmas, beginning with one that relies on conditional expectations under the MNS assumption.

**Lemma B.1.** *Let $(\Omega, \mathcal{F}, \{\mathcal{F}_k\}_{k\geq 0}, \mathbb{P})$ be a complete probability space with filtration $\{\mathcal{F}_k\}_{k\geq 0}$. Suppose $x_k$ is generated by SHANG/SHANG++, $g(x_k)$ denotes the stochastic estimator of $\nabla f(x_k)$, then the following statements hold*

*1. $\mathbb{E}\left[g(x_k) \mid \mathcal{F}_k\right] = \nabla f(x_k)$.*

*2. $\mathbb{E}\left[\|g(x_k) - \nabla f(x_k)\|^2\right] \leq \sigma^2\|\nabla f(x_k)\|^2$.*

*3. $\mathbb{E}\left[\langle g(x_k), \nabla f(x_k)\rangle\right] = \|\nabla f(x_k)\|^2$*

*4. $\mathbb{E}\left[\|g(x_k)\|^2\right] \leq (1+\sigma^2)\|\nabla f(x_k)\|^2$*

*Proof of Lemma B.1.* **First and second claim.** This follows from Fubini's theorem.

**Third claim.** For the third result, we observe that since $f$ is a deterministic function, $\nabla f(x_k)$ is $\mathcal{F}_k$-measurable, then, by the Theorem 8.14 in Klenke (2013), we have

$$\mathbb{E}\left[\langle g(x_k), \nabla f(x_k)\rangle\right] = \mathbb{E}\left[\mathbb{E}\left[\langle g(x_k), \nabla f(x_k)\rangle \mid \mathcal{F}_k\right]\right] = \mathbb{E}\left[\langle \mathbb{E}\left[g(x_k) \mid \mathcal{F}_k\right], \nabla f(x_k)\rangle\right] = \mathbb{E}\left[\|\nabla f(x_k)\|^2\right]$$

**Fourth claim.** For the fourth result, using the previous results, we have

$$\begin{aligned}
\mathbb{E}\left[\|g(x_k)\|^2\right] &= \mathbb{E}\left[\|g(x_k) - \nabla f(x_k)\|^2 + 2\langle g(x_k), \nabla f(x_k)\rangle - \|\nabla f(x_k)\|^2\right] \\
&= \mathbb{E}\left[\|g(x_k) - \nabla f(x_k)\|^2\right] + \mathbb{E}\left[2\langle g(x_k), \nabla f(x_k)\rangle\right] - \|\nabla f(x_k)\|^2 \\
&\leq \sigma^2\|\nabla f(x_k)\|^2 + 2\|\nabla f(x_k)\|^2 - \|\nabla f(x_k)\|^2 \\
&= (1+\sigma^2)\|\nabla f(x_k)\|^2
\end{aligned}$$

$\square$

Under the MNS assumption, this setup of auxiliary variable $x^+$ yields the following descent lemma for smooth objectives.

**Lemma B.2.** *Suppose that $x_k^+ = x_k - \eta g(x_k)$, $f \in \mathcal{C}_L^{1,1}$. Given $0 < \eta \leq \frac{1}{L(1+\sigma^2)}$, we have*

$$\mathbb{E}\left[f(x_k^+) - f(x^\star)\right] \leq f(x_k) - f(x^\star) - \frac{\eta}{2}\|\nabla f(x_k)\|^2$$

*Proof of Lemma B.2.* Using the $L$-smoothness of the function $f$:

$$f(y) - f(x) - \langle \nabla f(x), y - x\rangle \leq \frac{L}{2}\|y - x\|^2 \quad \forall x, y \in \mathbb{R}^d \tag{B.3}$$

and Lemma B.1, under the assumption of $0 < \eta \leq \frac{1}{L(1+\sigma^2)}$, we can obtain the desired result

$$
\mathbb{E}\left[f(x_k^+)\right] \leq \mathbb{E}\left[f(x_k) - \langle \eta g(x_k), \nabla f(x_k)\rangle + \frac{L}{2}\|\eta g(x_k)\|^2\right]
$$

$$
= f(x_k) - \mathbb{E}\left[\langle \eta g(x_k), \nabla f(x_k)\rangle\right] + \mathbb{E}\left[\frac{L}{2}\|\eta g(x_k)\|^2\right]
$$

$$
\leq f(x_k) - \eta\|\nabla f(x_k)\|^2 + \frac{L\eta^2(1+\sigma^2)}{2}\|\nabla f(x_k)\|^2
$$

$$
= f(x_k) - \eta(1 - \frac{L(1+\sigma^2)\eta}{2})\|\nabla f(x_k)\|^2
$$

$$
\leq f(x_k) - \frac{\eta}{2}\|\nabla f(x_k)\|^2
$$

$\square$

Define an auxiliary variable $x_k^+ = x_k - \alpha_k\beta_k g(x_k)$, substitue it into (Eq.B.1) yield:

$$
\frac{x_{k+1} - x_k^+}{\alpha_k} = v_k - x_{k+1}
$$

$$
\frac{v_{k+1} - v_k}{\alpha_k} = \frac{\mu}{\gamma_k}(x_{k+1} - v_{k+1}) - \frac{1}{\gamma_k}g(x_{k+1}) \tag{B.4}
$$

$$
\frac{\gamma_{k+1} - \gamma_k}{\alpha_k} = \mu - \gamma_{k+1}
$$

The next lemma controls the decay of $\mathbb{E}\left[\mathcal{E}(z_{k+1}^+; \gamma_{k+1})\right]$.

**Lemma B.3.** *Let $f \in \mathcal{S}_{\mu,L}$ with $0 \leq \mu < L < \infty$, Lyapunov function $\mathcal{E}$ is defined by (B.2). Given $(v_k, x_k^+)$, $(x_{k+1}, v_{k+1})$ are generated by (B.4) and $x_{k+1}^+ = x_{k+1} - \alpha_{k+1}\beta_{k+1}g(x_{k+1})$. Assume $0 < \alpha_{k+1}\beta_{k+1} = \alpha_k\beta_k \leq \frac{1}{L(1+\sigma^2)}$, we have*

$$
(1 + \alpha_k)\mathbb{E}\left[\mathcal{E}(z_{k+1}^+; \gamma_{k+1})\right]
$$
$$
\leq \mathcal{E}(z_k^+; \gamma_k) + \mathbb{E}\left[\frac{1}{2}(\frac{\alpha_k^2(1+\sigma^2)}{\gamma_k} - (1+\alpha_k)\alpha_k\beta_k)\|\nabla f(x_{k+1})\|^2 - \frac{\alpha_k\mu}{2}\|x_{k+1} - v_{k+1}\|^2 - D_f(x_k^+, x_{k+1})\right]
$$

*proof of Lemma B.3.* By Lemma B.2, if $0 < \alpha_k\beta_k = \alpha_{k+1}\beta_{k+1} \leq \frac{1}{L(1+\sigma^2)}$, we obtain the one-step decrease

$$
\mathbb{E}\left[\mathcal{E}(z_{k+1}^+; \gamma_{k+1})\right] - \mathcal{E}(z_k^+; \gamma_k)
$$
$$
\leq \mathbb{E}\left[\mathcal{E}(z_{k+1}; \gamma_{k+1}) - \mathcal{E}(z_k^+; \gamma_k) - \frac{\alpha_k\beta_k}{2}\|\nabla f(x_{k+1})\|^2\right] \tag{B.5}
$$
$$
= \mathbb{E}\left[\mathcal{E}(z_{k+1}; \gamma_k) - \mathcal{E}(z_k^+; \gamma_k) + \frac{\gamma_{k+1} - \gamma_k}{2}\|v_{k+1} - x^\star\|^2 - \frac{\alpha_k\beta_k}{2}\|\nabla f(x_{k+1})\|^2\right]
$$

Applying the Bregman divergence identity Chen & Teboulle (1993):

$$
\langle \nabla f(y) - \nabla f(x), y - z\rangle = D_f(z, y) + D_f(y, x) - D_f(z, x) \quad \forall, x, y, z \in \mathbb{R}^d \tag{B.6}
$$

together with the representation $\mathcal{E}(z; \gamma) = D_{\mathcal{E}}(z, z^*; \gamma)$ and the update rules into (B.5), we obtain

$$\mathbb{E}\left[\mathcal{E}(z_{k+1}^+; \gamma_{k+1})\right] - \mathcal{E}(z_k^+; \gamma_k)$$

$$\leq \mathbb{E}\left[\langle\nabla\mathcal{E}(z_{k+1}; \gamma_k), z_{k+1} - z_k^+\rangle - D_{\mathcal{E}}(z_k^+, z_{k+1}; \gamma_k) + \frac{\gamma_{k+1} - \gamma_k}{2}\|v_{k+1} - x^\star\|^2 - \frac{\alpha_k\beta_k}{2}\|\nabla f(x_{k+1})\|^2\right]$$

$$\leq \mathbb{E}\left[\langle\nabla f(x_{k+1}) - \nabla f(x^\star), x_{k+1} - x_k^+\rangle + \gamma_k\langle v_{k+1} - x^\star, v_{k+1} - v_k\rangle - D_{\mathcal{E}}(z_k^+, z_{k+1}; \gamma_k)\right.$$

$$\left. + \frac{\alpha_k(\mu - \gamma_{k+1})}{2}\|v_{k+1} - x^\star\|^2 - \frac{\alpha_k\beta_k}{2}\|\nabla f(x_{k+1})\|^2\right]$$

$$= \mathbb{E}\left[-\alpha_k\langle\nabla f(x_{k+1}) - \nabla f(x^\star), x_{k+1} - x^\star\rangle + \alpha_k\langle\nabla f(x_{k+1}), v_k - x^\star\rangle - \alpha_k\langle g(x_{k+1}), v_{k+1} - x^\star\rangle\right.$$

$$+ \alpha_k\mu\langle v_{k+1} - x^\star, x_{k+1} - v_{k+1}\rangle + \frac{\alpha_k(\mu - \gamma_{k+1})}{2}\|v_{k+1} - x^\star\|^2 - D_{\mathcal{E}}(z_k^+, z_{k+1}; \gamma_k)$$

$$\left. - \frac{\alpha_k\beta_k}{2}\|\nabla f(x_{k+1})\|^2\right]$$

(B.7)

By the definition of the Bregman divergence and the $\mu$-strong convexity of $f$, we have

$$\langle\nabla f(x_{k+1}) - \nabla f(x^\star), x_{k+1} - x^\star\rangle = D_f(x_{k+1}, x^\star) + D_f(x^\star, x_{k+1})$$
$$\geq D_f(x_{k+1}, x^\star) + \frac{\mu}{2}\|x_{k+1} - x^\star\|^2$$

(B.8)

and

$$\alpha_k\mu\langle v_{k+1} - x^\star, x_{k+1} - v_{k+1}\rangle = \frac{\alpha_k\mu}{2}(\|x_{k+1} - x^\star\|^2 - \|x_{k+1} - v_{k+1}\|^2 - \|v_{k+1} - x^\star\|^2) \quad \text{(B.9)}$$

We denote $\mathcal{F}_{k+1} = \sigma(x_0, \cdots, x_{k+1})$ the $\sigma$-algebra generated by the $k + 1$ first interates $\{x_i\}_{i=1}^{k+1}$ generated by SHANG. Since $f$ is a deterministic function, $v_k - x^\star$ is $\mathcal{F}_{k+1}$-measurable, then

$$\mathbb{E}\left[\langle g(x_{k+1}), v_k - x^\star\rangle\right] = \mathbb{E}\left[\mathbb{E}\left[\langle g(x_{k+1}), v_k - x^\star\rangle \mid \mathcal{F}_{k+1}\right]\right]$$
$$= \mathbb{E}\left[\langle\mathbb{E}\left[g(x_{k+1}) \mid \mathcal{F}_{k+1}\right], v_k - x^\star\rangle\right]$$
$$= \mathbb{E}\left[\langle\nabla f(x_{k+1}), v_k - x^\star\rangle\right]$$

Now, we apply this result in reverse, and using Young Inequality, Cauchy-Schwarz Inequality to obtain

$$\mathbb{E}\left[\alpha_k\langle\nabla f(x_{k+1}), v_k - x^\star\rangle - \alpha_k\langle g(x_{k+1}), v_{k+1} - x^\star\rangle\right]$$
$$= \mathbb{E}\left[\alpha_k\langle g(x_{k+1}), v_k - v_{k+1}\rangle\right]$$
$$\leq \mathbb{E}\left[\frac{\alpha_k^2}{2\gamma_k}\|g(x_{k+1})\|^2 + \frac{\gamma_k}{2}\|v_k - v_{k+1}\|^2\right] \quad \text{(B.10)}$$
$$\leq \mathbb{E}\left[\frac{\alpha_k^2(1 + \sigma^2)}{2\gamma_k}\|\nabla f(x_{k+1})\|^2 + \frac{\gamma_k}{2}\|v_k - v_{k+1}\|^2\right]$$

In addition, using the identity of squares (for $v$) and Bregman divergence indentity (B.6) (for $x^+$), we have the component form of

$$D_{\mathcal{E}}(z_k^+, z_{k+1}; \gamma_k) = D_f(x_k^+, x_{k+1}) + \frac{\gamma_k}{2}\|v_k - v_{k+1}\|^2 \quad \text{(B.11)}$$

Substituting (B.8-B.11) back into (B.7), we can obtain

$$\mathbb{E}\left[\mathcal{E}(z_{k+1}^+; \gamma_{k+1})\right] - \mathcal{E}(z_k^+; \gamma_k)$$

$$\leq \mathbb{E}\left[-\alpha_k D_f(x_{k+1}, x^\star) + \frac{1}{2}(\frac{\alpha_k^2(1 + \sigma^2)}{\gamma_k} - \alpha_k\beta_k)\|\nabla f(x_{k+1})\|^2\right.$$

$$\left. - \frac{\alpha_k\gamma_{k+1}}{2}\|v_{k+1} - x^\star\|^2 - \frac{\alpha_k\mu}{2}\|x_{k+1} - v_{k+1}\|^2 - D_f(x_k^+, x_{k+1})\right]$$

(B.12)

$$\leq \mathbb{E}\left[-\alpha_k D_f(x_{k+1}^+, x^\star) + \frac{1}{2}(\frac{\alpha_k^2(1 + \sigma^2)}{\gamma_k} - (1 + \alpha_k)\alpha_k\beta_k)\|\nabla f(x_{k+1})\|^2\right.$$

$$\left. - \frac{\alpha_k\gamma_{k+1}}{2}\|v_{k+1} - x^\star\|^2 - \frac{\alpha_k\mu}{2}\|x_{k+1} - v_{k+1}\|^2 - D_f(x_k^+, x_{k+1})\right]$$

By moving $\mathbb{E}\left[\mathcal{E}(z_{k+1}^+; \gamma_{k+1}) = D_f(x_{k+1}^+, x^\star) + \frac{\gamma_{k+1}}{2}\|v_{k+1} - x^\star\|^2\right]$ to the left side of the inequality to obtain the desired result. $\square$

Now we begin to prove Theorem B.1.

*Proof.* **(1).** When $0 < \mu < L < \infty$, set $\gamma = \mu$. By Lemma B.3, if $\alpha\beta \leq \frac{1}{(1+\sigma^2)L}$, we have

$$(1+\alpha)\mathbb{E}\left[\mathcal{E}(z_{k+1}^+;\mu)\right]$$

$$\leq \mathcal{E}(z_k^+;\mu) + \mathbb{E}\left[\frac{1}{2}(\frac{\alpha^2(1+\sigma^2)}{\mu} - (1+\alpha)\alpha\beta)\|\nabla f(x_{k+1})\|^2 - \frac{\alpha\mu}{2}\|x_{k+1} - v_{k+1}\|^2 - D_f(x_k^+, x_{k+1})\right]$$
(B.13)

Assume $\alpha\beta = \frac{(1+\sigma^2)\alpha^2}{\mu} \leq \frac{1}{(1+\sigma^2)L}$, i.e., the step size satisfies $0 < \alpha \leq \frac{1}{1+\sigma^2}\sqrt{\frac{\mu}{L}}$ to ensure that all the coefficients of the terms on the right side of the inequality, except for $\mathcal{E}(z_k^+;\mu)$, are non-positive. Thus,

$$\mathbb{E}\left[\mathcal{E}(z_{k+1}^+;\mu)\right] \leq (1+\alpha)^{-1}\mathcal{E}(z_k^+;\mu) \leq (1+\alpha)^{-(k+1)}\mathcal{E}(z_0;\mu) \tag{B.14}$$

**(2).** When $\mu = 0$. Assume $\alpha_k = \frac{2}{k+1}$, $\gamma_k = \alpha_k^2(1+\sigma^2)^2 L$ and $\beta_k = \frac{(1+\sigma^2)\alpha_k}{\gamma_k}$. Using Lemma B.3 to obtain

$$\mathbb{E}\left[\mathcal{E}(z_{k+1}^+;\gamma_{k+1})\right] \leq \frac{k+1}{k+3}\mathcal{E}(z_k^+;\gamma_k) \leq \frac{2}{(k+2)(k+3)}\mathcal{E}(z_0;\gamma_0) \tag{B.15}$$

$\square$

**Corollary B.1.** *Under the setting of Theorem B.1, SHANG achieves an $\varepsilon$-precision solution within the following number of iterations:*

*(1) When $\mu = 0$, with $\alpha_k = \frac{2}{k+1}$, $\gamma_k = \alpha_k^2(1+\sigma^2)^2 L$ and $\beta_k = \frac{(1+\sigma^2)\alpha_k}{\gamma_k}$,*

$$k \geq \sqrt{\frac{2(f(x_0) - f(x^\star) + 2(1+\sigma^2)^2 L\|x_0 - x^\star\|^2)}{\varepsilon}}$$

*(2) When $0 < \mu < L < \infty$, with $\alpha = \frac{1}{1+\sigma^2}\sqrt{\frac{\mu}{L}}$ and $\beta = \frac{(1+\sigma^2)\alpha}{\mu}$,*

$$k \geq (1+\sigma^2)\sqrt{\frac{L}{\mu}}\log\left(\frac{f(x_0) - f(x^\star) + \frac{\mu}{2}\|x_0 - x^\star\|^2}{\varepsilon}\right).$$

**Corollary B.2.** *In the setting of Theorem B.1, $f(x_k^+) \overset{a.s.}{\to} f(x^\star)$.*

*proof of Corollary B.2.* We assume that all the conditions of Theorem B.1 have been met, we have

$$\mathbb{E}\left[|f(x_k^+) - f(x^\star)|\right] = \mathbb{E}\left[f(x_k^+) - f(x^\star)\right] \leq Cq^k$$

holds for some positive constant $C$. Here $0 < q < 1$ is the decay factor. In fact, $q = (1 + \frac{1}{1+\sigma^2}\sqrt{\frac{\mu}{L}})^{-1}$ in strongly convex cases and $q = \frac{2}{(k+2)(k+3)}$ in convex cases. Since

$$\mathbb{P}\left(\lim_{k\to\infty} f(x_k^+) \neq f(x^\star)\right) = \mathbb{P}\left(\limsup_{k\to\infty} |f(x_k^+) - f(x^\star)| > 0\right)$$

$$= \mathbb{P}\left(\bigcup_{n=1}^{\infty} \limsup_{k\to\infty} |f(x_k^+) - f(x^\star)| > \frac{1}{n}\right)$$

$$\leq \sum_{n=1}^{\infty} \mathbb{P}\left(\limsup_{k\to\infty} |f(x_k^+) - f(x^\star)| > \frac{1}{n}\right)$$

For any fixed $\varepsilon = \frac{1}{n} > 0$ and for any $N \in \mathbb{N}$, we have

$$\mathbb{P}\left(\limsup_{k\to\infty} \mid f(x_k^+) - f(x^\star) \mid > \varepsilon\right) \leq \mathbb{P}\left(\exists k \geq N \quad \text{s.t.} \mid f(x_k^+) - f(x^\star) \mid > \varepsilon\right)$$

$$= \mathbb{P}\left(\bigcup_{k\geq N}\{\mid f(x_k^+) - f(x^\star) \mid > \varepsilon\}\right)$$

$$\leq \sum_{k\geq N} \mathbb{P}\left(\mid f(x_k^+) - f(x^\star) \mid > \varepsilon\right)$$

$$\leq \sum_{k\geq N} \frac{\mathbb{E}\left[\mid f(x_k^+) - f(x^\star) \mid\right]}{\varepsilon}$$

$$\leq \frac{C}{\varepsilon} \sum_{k\geq N} q^k$$

$$\leq \frac{C}{\varepsilon}\frac{q^N}{1-q}$$

where in the penultimate step we use Markov's inequality. For a fixed $\varepsilon = 1/n$, the inequality above holds for any $N$. Letting $N \to \infty$, the right-hand side converges to $0$, hence the probability on the left-hand side is zero. Since

$$\left\{\lim_{k\to\infty} f(x_k^+) \neq f(x^\star)\right\} = \bigcup_{n=1}^{\infty}\left\{\limsup_{k\to\infty} |f(x_k^+) - f(x^\star)| \geq 1/n\right\},$$

taking a countable union over all $\varepsilon = 1/n$ yields $\mathbb{P}(\lim_{k\to\infty} f(x_k^+) \neq f(x^\star)) = 0$. $\qquad\square$

## C   SHANG++

### C.1   PROOF OF THEOREM 2.1

Setting $\gamma = \mu$, SHANG++ (2.4) can be rewritten in the following equivalent form:

$$\frac{x_{k+1} - x_k^+}{\tilde{\alpha}} = v_k - x_{k+1}$$

$$\frac{v_{k+1} - v_k}{\alpha} = x_{k+1} - v_{k+1} - \frac{1}{\mu}g(x_{k+1}) \tag{C.1}$$

$$x_{k+1}^+ = x_{k+1} - \tilde{\alpha}\beta g(x_{k+1})$$

where $\tilde{\alpha} = \frac{\alpha}{1+m\alpha}$ or $\alpha = \frac{\tilde{\alpha}}{1-m\tilde{\alpha}}$.

For this equivalent form of SHANG++, we obtain the following convergence result.

**Theorem C.1.** *Let $f \in \mathcal{S}_{\mu,L}$. Given $x_0^+ = v_0 = x_0$, suppose $(x_k, v_k)$ are generated by (C.1) with $g(x_k)$ defined in (1.2) and MNS (1.3) holds. Given $0 \leq m \leq 1$, if the step size satisfies $\alpha = \frac{\tilde{\alpha}}{1-m\tilde{\alpha}}$ with $0 < \tilde{\alpha} \leq \frac{1}{1+\sigma^2}\sqrt{\frac{\mu}{L}}$ and $\beta = \frac{\tilde{\alpha}}{\mu/(1+\sigma^2)}$, then*

$$\mathbb{E}\left[f(x_k^+) - f(x^\star) + \frac{\mu(1+\alpha)}{2(1+m\alpha)}\|v_k - x^\star\|^2\right] \leq (1+\tilde{\alpha})^{-k}\left(f(x_0)-f(x^\star)+\frac{\mu(1+\alpha)}{2(1+m\alpha)}\|v_0-x^\star\|^2\right).$$

Define the discrete Lyapunov function

$$\mathcal{E}(z_k^+; \mu) = f(x_k^+) - f(x^\star) + \frac{\mu(1+\alpha)}{2(1+m\alpha)}\|v_k - x^\star\|^2 \tag{C.2}$$

The next lemma controls the decay of $\mathbb{E}\left[\mathcal{E}(z_{k+1}^+; \mu)\right]$.

**Lemma C.1.** *Let $f \in \mathcal{S}_{\mu,L}$ with $0 < \mu < L < \infty$. Lyapunov function $\mathcal{E}$ is defined by (C.2). Given $(x_k^+, x_k, v_k)$, $(x_{k+1}^+, x_{k+1}, v_{k+1})$ are generated by (C.1). Assume $0 < \tilde{\alpha}\beta \leq \frac{1}{L(1+\sigma^2)}$, we have*

$$(1 + (1 + m\alpha)\tilde{\alpha})\mathbb{E}\left[\mathcal{E}(z_{k+1}^+; \mu)\right] - \mathcal{E}(z_k^+; \mu)$$

$$\leq \mathbb{E}\left[-(1-m)\alpha\tilde{\alpha}(f(x_{k+1}^+) - f(x^\star)) - \frac{\tilde{\alpha}\mu}{2}\|v_k - x_{k+1}\|^2\right.$$

$$\left. - D_f(x_k^+, x_{k+1}; \mu) + \left(\frac{\tilde{\alpha}\alpha(1+\sigma^2)}{2\mu} - \frac{\tilde{\alpha}\beta}{2}(1 + (1+\alpha)\tilde{\alpha})\right)\|\nabla f(x_{k+1})\|^2\right]$$

*proof of Lemma C.1.* By Lemma B.2, if $0 < \tilde{\alpha}\beta \leq \frac{1}{L(1+\sigma^2)}$, we obtain the one-step decrease

$$\mathbb{E}\left[\mathcal{E}(z_{k+1}^+; \mu)\right] - \mathcal{E}(z_k^+; \mu) \leq \mathbb{E}\left[\mathcal{E}(z_{k+1}; \mu) - \mathcal{E}(z_k^+; \mu) - \frac{\tilde{\alpha}\beta}{2}\|\nabla f(x_{k+1})\|^2\right] \tag{C.3}$$

Expand it yields

$$\mathbb{E}\left[\mathcal{E}(z_{k+1}^+; \mu)\right] - \mathcal{E}(z_k^+; \mu)$$

$$\leq \mathbb{E}\left[\langle\nabla\mathcal{E}(z_{k+1}; \mu), z_{k+1} - z_k^+\rangle - D_\mathcal{E}(z_k^+, z_{k+1}; \mu) - \frac{\tilde{\alpha}\beta}{2}\|\nabla f(x_{k+1})\|^2\right]$$

$$= \mathbb{E}\left[\langle\nabla f(x_{k+1}) - \nabla f(x^\star), x_{k+1} - x_k^+\rangle + \frac{\mu(1+\alpha)}{(1+m\alpha)}\langle v_{k+1} - x^\star, v_{k+1} - v_k\rangle - D_\mathcal{E}(z_k^+, z_{k+1}; \mu)\right.$$

$$\left. - \frac{\tilde{\alpha}\beta}{2}\|\nabla f(x_{k+1})\|^2\right]$$

$$= \mathbb{E}\left[\tilde{\alpha}\langle\nabla f(x_{k+1}) - \nabla f(x^\star), v_k - x_{k+1}\rangle + \alpha\frac{\mu(1+\alpha)}{(1+m\alpha)}\langle v_{k+1} - x^\star, x_{k+1} - v_{k+1} - \frac{1}{\mu}g(x_{k+1})\rangle\right.$$

$$\left. - D_\mathcal{E}(z_k^+, z_{k+1}; \mu) - \frac{\tilde{\alpha}\beta}{2}\|\nabla f(x_{k+1})\|^2\right]$$

$$= \mathbb{E}\left[-\tilde{\alpha}\langle\nabla f(x_{k+1}) - \nabla f(x^\star), x_{k+1} - x^\star\rangle + (1+\alpha)\tilde{\alpha}\langle\nabla g(x_{k+1}), v_k - v_{k+1}\rangle - \alpha\tilde{\alpha}\langle g(x_{k+1}), v_k - x^\star\rangle\right.$$

$$\left. + \mu(1+\alpha)\tilde{\alpha}\langle v_{k+1} - x^\star, x_{k+1} - v_{k+1}\rangle - D_\mathcal{E}(z_k^+, z_{k+1}; \mu) - \frac{\tilde{\alpha}\beta}{2}\|\nabla f(x_{k+1})\|^2\right]$$

$$\tag{C.4}$$

where in the last step, we rewrote the coefficient as $\alpha\frac{\mu(1+\alpha)}{(1+m\alpha)} = \mu(1 + \alpha)\tilde{\alpha}$, and use $\mathbb{E}\left[\langle\nabla f(x_{k+1}), v_k - x^\star\rangle\right] = \mathbb{E}\left[\langle g(x_{k+1}), v_k - x^\star\rangle\right]$.

Using (B.8, B.9) and B.11), we further bound (C.4) as

$$\mathbb{E}\left[\mathcal{E}(z_{k+1}^+; \mu)\right] - \mathcal{E}(z_k^+; \mu)$$

$$= \mathbb{E}\left[-\tilde{\alpha}D_f(x_{k+1}, x^\star) - \frac{\mu(1+\alpha)\tilde{\alpha}}{2}\|v_{k+1} - x^\star\|^2 - \frac{\mu(1+\alpha)\tilde{\alpha}}{2}\|v_{k+1} - x_{k+1}\|^2 + \frac{\mu\alpha\tilde{\alpha}}{2}\|x_{k+1} - x^\star\|^2\right.$$

$$\left. - D_f(x_k^+, x_{k+1}; \mu) - \frac{\mu(1+\alpha)}{2(1+m\alpha)}\|v_k - v_{k+1}\|^2 - \frac{\tilde{\alpha}\beta}{2}\|\nabla f(x_{k+1})\|^2\right.$$

$$\left. + (1+\alpha)\tilde{\alpha}\langle\nabla g(x_{k+1}), v_k - v_{k+1}\rangle - \alpha\tilde{\alpha}\langle g(x_{k+1}), v_k - x^\star\rangle\right]$$

$$\tag{C.5}$$

For the terms in last line, using the update for $v_{k+1}$ in (C.1) yields the following bound.

$$\mathbb{E}\left[(1+\alpha)\tilde{\alpha}\langle g(x_{k+1}), v_k - v_{k+1}\rangle\right]$$

$$= \mathbb{E}\left[(1+\alpha)\tilde{\alpha}\alpha\mu\langle\frac{1}{\mu}g(x_{k+1}), \frac{v_k - v_{k+1}}{\alpha}\rangle\right]$$

$$= \mathbb{E}\left[\frac{(1+\alpha)\tilde{\alpha}\alpha\mu}{2}(\|\frac{1}{\mu}g(x_{k+1})\|^2 + \|\frac{v_k - v_{k+1}}{\alpha}\|^2 - \|\frac{v_k - v_{k+1}}{\alpha} - \frac{1}{\mu}g(x_{k+1})\|^2)\right]$$

$$= \mathbb{E}\left[\frac{(1+\alpha)\tilde{\alpha}\alpha}{2\mu}\|g(x_{k+1})\|^2 + \frac{(1+\alpha)\tilde{\alpha}\mu}{2\alpha}\|v_k - v_{k+1}\|^2 - \frac{(1+\alpha)\tilde{\alpha}\alpha\mu}{2}\|x_{k+1} - v_{k+1}\|^2\right]$$

$$\leq \mathbb{E}\left[\frac{\tilde{\alpha}\alpha(1+\sigma^2)}{2\mu}\|\nabla f(x_{k+1})\|^2 + \frac{\alpha^2\tilde{\alpha}}{2\mu}\|g(x_{k+1})\|^2 + \frac{(1+\alpha)\mu}{2(1+m\alpha)}\|v_k - v_{k+1}\|^2\right.$$

$$\left. - \frac{(1+\alpha)\tilde{\alpha}\alpha\mu}{2}\|x_{k+1} - v_{k+1}\|^2\right]$$

$$\text{(C.6)}$$

where in the last step, we split the coefficient of $\|g(x_{k+1})\|^2$ into $\frac{\tilde{\alpha}\alpha}{2\mu}$ and $\frac{\alpha^2\tilde{\alpha}}{2\mu}$, and then use Lemma B.1 to control the first term.

By the update for $v_{k+1}$, we have $v_k - v^\star = (1+\alpha)(v_{k+1} - x_{k+1}) + (x_{k+1} - x^\star) + \frac{\alpha}{\mu}g(x_{k+1})$ and $v_k - x_{k+1} = (1+\alpha)(v_{k+1} - x_{k+1}) + \frac{\alpha}{\mu}g(x_{k+1})$, then

$$\mathbb{E}\left[-\alpha\tilde{\alpha}\langle g(x_{k+1}), v_k - v^\star\rangle\right]$$

$$= \mathbb{E}\left[-\alpha\tilde{\alpha}\langle g(x_{k+1}), (1+\alpha)(v_{k+1} - x_{k+1}) + (x_{k+1} - x^\star) + \frac{\alpha}{\mu}g(x_{k+1})\rangle\right]$$

$$= \mathbb{E}\left[-\tilde{\alpha}\mu\langle\frac{\alpha}{\mu}g(x_{k+1}), (1+\alpha)(v_{k+1} - x_{k+1})\rangle - \alpha\tilde{\alpha}\langle g(x_{k+1}), x_{k+1} - x^\star\rangle - \frac{\alpha^2\tilde{\alpha}}{\mu}\|g(x_{k+1})\|^2\right]$$

$$= \mathbb{E}\left[\frac{\tilde{\alpha}\mu}{2}\left(-\|\frac{\alpha}{\mu}g(x_{k+1}) + (1+\alpha)(v_{k+1} - x_{k+1})\|^2 + \|\frac{\alpha}{\mu}g(x_{k+1})\|^2 + \|(1+\alpha)(v_{k+1} - x_{k+1})\|^2\right)\right.$$

$$\left. - \alpha\tilde{\alpha}\langle\nabla f(x_{k+1}), x_{k+1} - x^\star\rangle - \frac{\alpha^2\tilde{\alpha}}{\mu}\|g(x_{k+1})\|^2\right]$$

$$= \mathbb{E}\left[-\frac{\tilde{\alpha}\mu}{2}\|v_k - x_{k+1}\|^2 + \frac{\alpha^2\tilde{\alpha}}{2\mu}\|g(x_{k+1})\|^2 + \frac{\tilde{\alpha}(1+\alpha)^2\mu}{2}\|v_{k+1} - x_{k+1}\|^2\right.$$

$$\left. - \alpha\tilde{\alpha}(D_f(x_{k+1}, x^\star) + D_f(x^\star, x_{k+1})) - \frac{\alpha^2\tilde{\alpha}}{\mu}\|g(x_{k+1})\|^2\right]$$

$$\leq \mathbb{E}\left[-\frac{\tilde{\alpha}\mu}{2}\|v_k - x_{k+1}\|^2 - \frac{\alpha^2\tilde{\alpha}}{2\mu}\|g(x_{k+1})\|^2 + \frac{\tilde{\alpha}(1+\alpha)^2\mu}{2}\|v_{k+1} - x_{k+1}\|^2\right.$$

$$\left. - \alpha\tilde{\alpha}D_f(x_{k+1}, x^\star) - \frac{\alpha\tilde{\alpha}\mu}{2}\|x_{k+1} - x^\star\|^2\right]$$

$$\text{(C.7)}$$

Substituating (C.6) and (C.7) back into (C.5), we have

$$\mathbb{E}\left[\mathcal{E}(z_{k+1}^+; \mu)\right] - \mathcal{E}(z_k^+; \mu)$$

$$= \mathbb{E}\left[-(1+\alpha)\tilde{\alpha}D_f(x_{k+1}, x^\star) - \frac{\mu(1+\alpha)\tilde{\alpha}}{2}\|v_{k+1} - x^\star\|^2 - \frac{\tilde{\alpha}\mu}{2}\|v_k - x_{k+1}\|^2\right.$$

$$\left. - D_f(x_k^+, x_{k+1}; \mu) + (\frac{\tilde{\alpha}\alpha(1+\sigma^2)}{2\mu} - \frac{\tilde{\alpha}\beta}{2})\|\nabla f(x_{k+1})\|^2\right] \quad \text{(C.8)}$$

$$= \mathbb{E}\left[-(1+\alpha)\tilde{\alpha}(f(x_{k+1}) - f(x^\star)) - \frac{\mu(1+\alpha)\tilde{\alpha}}{2}\|v_{k+1} - x^\star\|^2 - \frac{\tilde{\alpha}\mu}{2}\|v_k - x_{k+1}\|^2\right.$$

$$\left. - D_f(x_k^+, x_{k+1}; \mu) + (\frac{\tilde{\alpha}\alpha(1+\sigma^2)}{2\mu} - \frac{\tilde{\alpha}\beta}{2})\|\nabla f(x_{k+1})\|^2\right]$$

Using Lemma B.2 to obtain

$$\mathbb{E}\left[\mathcal{E}(z_{k+1}^+;\mu)\right] - \mathcal{E}(z_k^+;\mu)$$

$$\leq \mathbb{E}\left[-(1+\alpha)\tilde{\alpha}(f(x_{k+1}^+) - f(x^\star)) - \tilde{\alpha}\frac{(1+\alpha)\mu}{2}\|v_{k+1} - x^\star\|^2 - \frac{\tilde{\alpha}\mu}{2}\|v_k - x_{k+1}\|^2\right.$$

$$\left. - D_f(x_k^+, x_{k+1};\mu) + \big(\frac{\tilde{\alpha}\alpha(1+\sigma^2)}{2\mu} - \frac{\tilde{\alpha}\beta}{2}(1+(1+\alpha)\tilde{\alpha})\big)\|\nabla f(x_{k+1})\|^2\right] \qquad (C.9)$$

$$= \mathbb{E}\left[-(1+m\alpha)\tilde{\alpha}\mathcal{E}(z_{k+1}^+;\mu) - (1-m)\alpha\tilde{\alpha}(f(x_{k+1}^+) - f(x^\star)) - \frac{\tilde{\alpha}\mu}{2}\|v_k - x_{k+1}\|^2\right.$$

$$\left. - D_f(x_k^+, x_{k+1};\mu) + \big(\frac{\tilde{\alpha}\alpha(1+\sigma^2)}{2\mu} - \frac{\tilde{\alpha}\beta}{2}(1+(1+\alpha)\tilde{\alpha})\big)\|\nabla f(x_{k+1})\|^2\right]$$

By moving $\mathbb{E}\left[\mathcal{E}(z_{k+1}^+;\mu)\right]$ to the left side of the inequality to obtain the desired result. $\qquad\square$

Now we begin to prove Theorem C.1.

*Proof.* Under the parameter choices $0 < \tilde{\alpha} \leq \frac{1}{1+\sigma^2}\sqrt{\frac{\mu}{L}}$ and $\beta = \frac{\tilde{\alpha}(1+\sigma^2)}{\mu}$, we have $\tilde{\alpha}\beta \leq \frac{1}{(1+\sigma^2)L}$. According to Lemma C.1, in order to obtain the decay of $\mathbb{E}\left[\mathcal{E}(z_{k+1}^+;\mu)\right]$, we need the last term $\mathbb{E}\left[\big(\frac{\tilde{\alpha}\alpha(1+\sigma^2)}{2\mu} - \frac{\tilde{\alpha}\beta}{2}(1+(1+\alpha)\tilde{\alpha})\big)\|\nabla f(x_{k+1})\|^2\right]$ to be non-positive.

Using $\alpha = \frac{\tilde{\alpha}}{1-m\tilde{\alpha}}$, we have

$$\frac{\tilde{\alpha}\alpha(1+\sigma^2)}{2\mu} = \frac{\tilde{\alpha}^2(1+\sigma^2)}{2\mu(1-m\tilde{\alpha})} = \frac{\tilde{\alpha}\beta}{2(1-m\tilde{\alpha})}$$

and

$$\frac{1}{1-m\tilde{\alpha}} - 1 - (1+\alpha)\tilde{\alpha} = \frac{1}{1-m\tilde{\alpha}} - 1 - (1+\frac{\tilde{\alpha}}{1-m\tilde{\alpha}})\tilde{\alpha} = \frac{(m-1)\tilde{\alpha}(1+\tilde{\alpha})}{1-m\tilde{\alpha}} \leq 0$$

holds when $0 \leq m \leq 1$.

Therefore, we have

$$(1+(1+m\alpha)\tilde{\alpha})\mathbb{E}\left[\mathcal{E}(z_{k+1}^+;\mu)\right] - \mathcal{E}(z_k^+;\mu)$$

$$\leq \mathbb{E}\left[-(1-m)\alpha\tilde{\alpha}(f(x_{k+1}^+) - f(x^\star)) - \frac{\tilde{\alpha}\mu}{2}\|v_k - x_{k+1}\|^2 - D_f(x_k^+, x_{k+1};\mu)\right] \qquad (C.10)$$

which implies that

$$\mathbb{E}\left[\mathcal{E}(z_{k+1}^+;\mu)\right] \leq (1+(1+m\alpha)\tilde{\alpha})^{-1}\mathbb{E}\left[\mathcal{E}(z_k^+;\mu)\right] \leq (1+(1+m\alpha)\tilde{\alpha})^{-k-1}\mathcal{E}(z_0;\mu)$$

$\qquad\square$

**Corollary C.1.** *Under the setting of Theorem C.1, choose* $\tilde{\alpha} = \frac{1}{1+\sigma^2}\sqrt{\frac{\mu}{L}}$, $\beta = \frac{(1+\sigma^2)\tilde{\alpha}}{\mu}$, *and* $\alpha = \frac{\tilde{\alpha}}{1-m\tilde{\alpha}}$ *with* $0 \leq m \leq 1$, *SHANG++ guarantees an* $\varepsilon$-*precision solution within the following number of iterations:*

$$k \geq (1+\sigma^2)\sqrt{\frac{L}{\mu}}\log\left(\frac{f(x_0) - f(x^\star) + \frac{\mu}{2}\|v_0 - x^\star\|^2}{\varepsilon}\right)$$

## C.2    PROOF OF THEOREM 2.2

To facilitate analysis, we define an auxiliary time-scaling factor $\tilde{\gamma}_k = \frac{\gamma_k}{1+m\alpha_k}$. For any $m \geq 0$, setting $\alpha_k = \frac{2}{k+1}$, $\tilde{\alpha}_k = \frac{\alpha_k}{1+m\alpha_k} = \frac{2}{k+1+2m}$ and $\gamma_k = \alpha_k \tilde{\alpha}_k (1+\sigma^2)^2 L$, we have

$$
\begin{aligned}
\frac{\tilde{\gamma}_{k+1} - \tilde{\gamma}_k}{\tilde{\alpha}_k} &= \frac{1+m\alpha_k}{\alpha_k}\Big(\frac{\alpha_{k+1}^2 (1+\sigma^2)^2 L}{(1+m\alpha_{k+1})^2} - \frac{\alpha_k^2 (1+\sigma^2)^2 L}{(1+m\alpha_k)^2}\Big) \\
&= \frac{k+1+2m}{2}\Big(\frac{4(1+\sigma^2)^2 L}{(k+2+2m)^2} - \frac{4(1+\sigma^2)^2 L}{(k+1+2m)^2}\Big) \\
&= \frac{k+1+2m}{2}\Big(1 - \frac{(k+2+2m)^2}{(k+1+2m)^2}\Big)\tilde{\gamma}_{k+1} \\
&= -\Big(1 + \frac{1}{2(k+1+2m)}\Big)\tilde{\gamma}_{k+1} \\
&\leq -\tilde{\gamma}_{k+1}
\end{aligned}
\tag{C.11}
$$

Define $x_k^+ = x_k - \tilde{\alpha}_k \beta_k g(x_k)$, we can obtain the following equivalent form of SHANG++ for convex problems:

$$
\begin{aligned}
\frac{x_{k+1} - x_k^+}{\tilde{\alpha}_k} &= v_k - x_{k+1} \\
\frac{v_{k+1} - v_k}{\alpha_k} &= -\frac{1}{\gamma_k} g(x_{k+1}) \\
\frac{\tilde{\gamma}_{k+1} - \tilde{\gamma}_k}{\tilde{\alpha}_k} &\leq -\tilde{\gamma}_{k+1}
\end{aligned}
\tag{C.12}
$$

Denote the discrete Lyapunov function by

$$
\mathcal{E}(z_k^+; \tilde{\gamma}_k) = f(x_k^+) - f(x^\star) + \frac{\tilde{\gamma}_k}{2}\|v_k - x^\star\|^2
\tag{C.13}
$$

The following Lemma establishes a decay bound for $\mathbb{E}\left[\mathcal{E}(z_k^+; \tilde{\gamma}_k)\right]$.

**Lemma C.2.** *Let $f \in \mathcal{S}_{0,L}$, Lyapunov function $\mathcal{E}$ is defined by (C.13). Given $(x_k, v_k, x_k^+)$, $(x_{k+1}, v_{k+1})$ are generated by (C.12) and $x_{k+1}^+ = x_{k+1} - \tilde{\alpha}_k \beta_k g(x_{k+1})$. Assume $0 < \tilde{\alpha}_k \beta_k = \tilde{\alpha}_{k+1}\beta_{k+1} \leq \frac{1}{L(1+\sigma^2)}$, we have*

$$
(1+\tilde{\alpha}_k)\mathbb{E}\left[\mathcal{E}(z_{k+1}^+; \tilde{\gamma}_{k+1})\right]
$$
$$
\leq \mathcal{E}(z_k^+; \tilde{\gamma}_k) + \mathbb{E}\left[-\tilde{\alpha}_k D_f(x^\star, x_{k+1}) - D_f(x_k^+, x_{k+1}) + \frac{1}{2}\Big(\frac{\tilde{\alpha}_k^2(1+\sigma^2)}{\tilde{\gamma}_k} - (1+\tilde{\alpha}_k)\tilde{\alpha}_k \beta_k\Big)\|\nabla f(x_{k+1})\|^2\right]
$$

*proof of Lemma C.2.* By Lemma B.2, if $0 < \tilde{\alpha}_k \beta_k = \tilde{\alpha}_{k+1}\beta_{k+1} \leq \frac{1}{L(1+\sigma^2)}$, we obtain the one-step decrease

$$
\begin{aligned}
&\mathbb{E}\left[\mathcal{E}(z_{k+1}^+; \tilde{\gamma}_{k+1})\right] - \mathcal{E}(z_k^+; \tilde{\gamma}_k) \\
&\leq \mathbb{E}\left[\mathcal{E}(z_{k+1}; \tilde{\gamma}_{k+1}) - \mathcal{E}(z_k^+; \tilde{\gamma}_k) - \frac{\tilde{\alpha}_k \beta_k}{2}\|\nabla f(x_{k+1})\|^2\right] \\
&= \mathbb{E}\left[\mathcal{E}(z_{k+1}; \tilde{\gamma}_k) - \mathcal{E}(z_k^+; \tilde{\gamma}_k) + \frac{\tilde{\gamma}_{k+1} - \tilde{\gamma}_k}{2}\|v_{k+1} - x^\star\|^2 - \frac{\tilde{\alpha}_k \beta_k}{2}\|\nabla f(x_{k+1})\|^2\right]
\end{aligned}
\tag{C.14}
$$

Expand the above equation and use the update to obtain

$$\mathbb{E}\left[\mathcal{E}(z_{k+1}^+; \tilde{\gamma}_{k+1})\right] - \mathcal{E}(z_k^+; \tilde{\gamma}_k)$$

$$\leq \mathbb{E}\left[\langle \nabla \mathcal{E}(z_{k+1}; \tilde{\gamma}_k), z_{k+1} - z_k^+ \rangle - D_{\mathcal{E}}(z_k^+, z_{k+1}; \tilde{\gamma}_k) + \frac{\tilde{\gamma}_{k+1} - \tilde{\gamma}_k}{2}\|v_{k+1} - x^\star\|^2 - \frac{\tilde{\alpha}_k \beta_k}{2}\|\nabla f(x_{k+1})\|^2\right]$$

$$\leq \mathbb{E}\left[\langle \nabla f(x_{k+1}) - \nabla f(x^\star), x_{k+1} - x_k^+ \rangle + \tilde{\gamma}_k \langle v_{k+1} - x^\star, v_{k+1} - v_k \rangle - D_{\mathcal{E}}(z_k^+, z_{k+1}; \tilde{\gamma}_k)\right.$$

$$\left. - \frac{\tilde{\alpha}_k \tilde{\gamma}_{k+1}}{2}\|v_{k+1} - x^\star\|^2 - \frac{\tilde{\alpha}_k \beta_k}{2}\|\nabla f(x_{k+1})\|^2\right]$$

$$= \mathbb{E}\left[-\tilde{\alpha}_k \langle \nabla f(x_{k+1}) - \nabla f(x^\star), x_{k+1} - x^\star \rangle + \tilde{\alpha}_k \langle \nabla f(x_{k+1}), v_k - x^\star \rangle - \frac{\alpha_k \tilde{\gamma}_k}{\gamma_k}\langle g(x_{k+1}), v_{k+1} - x^\star \rangle\right.$$

$$\left. - \frac{\tilde{\alpha}_k \tilde{\gamma}_{k+1}}{2}\|v_{k+1} - x^\star\|^2 - D_{\mathcal{E}}(z_k^+, z_{k+1}; \tilde{\gamma}_k) - \frac{\tilde{\alpha}_k \beta_k}{2}\|\nabla f(x_{k+1})\|^2\right]$$

(C.15)

Using Young Inequality, Cauchy-Schwarz Inequality and $\frac{\alpha_k \tilde{\gamma}_k}{\gamma_k} = \tilde{\alpha}_k$ to obtain

$$\mathbb{E}\left[\tilde{\alpha}_k \langle \nabla f(x_{k+1}), v_k - x^\star \rangle - \frac{\alpha_k \tilde{\gamma}_k}{\gamma_k}\langle g(x_{k+1}), v_{k+1} - x^\star \rangle\right]$$

$$= \mathbb{E}\left[\tilde{\alpha}_k \langle g(x_{k+1}), v_k - v_{k+1} \rangle\right]$$

$$\leq \mathbb{E}\left[\frac{\tilde{\alpha}_k^2}{2\tilde{\gamma}_k}\|g(x_{k+1})\|^2 + \frac{\tilde{\gamma}_k}{2}\|v_k - v_{k+1}\|^2\right]$$

$$\leq \mathbb{E}\left[\frac{\tilde{\alpha}_k^2(1 + \sigma^2)}{2\tilde{\gamma}_k}\|\nabla f(x_{k+1})\|^2 + \frac{\tilde{\gamma}_k}{2}\|v_k - v_{k+1}\|^2\right]$$

(C.16)

Substituting (B.11) and (C.16) back into (C.15), we can obtain

$$\mathbb{E}\left[\mathcal{E}(z_{k+1}^+; \tilde{\gamma}_{k+1})\right] - \mathcal{E}(z_k^+; \tilde{\gamma}_k)$$

$$\leq \mathbb{E}\left[-\tilde{\alpha}_k D_f(x_{k+1}, x^\star) - \tilde{\alpha}_k D_f(x^\star, x_{k+1}) + \frac{1}{2}\left(\frac{\tilde{\alpha}_k^2(1 + \sigma^2)}{\tilde{\gamma}_k} - \tilde{\alpha}_k \beta_k\right)\|\nabla f(x_{k+1})\|^2\right.$$

$$\left. - \frac{\tilde{\alpha}_k \tilde{\gamma}_{k+1}}{2}\|v_{k+1} - x^\star\|^2 - D_f(x_k^+, x_{k+1})\right]$$

$$\leq \mathbb{E}\left[-\tilde{\alpha}_k \mathcal{E}(z_{k+1}^+; \tilde{\gamma}_{k+1}) - \tilde{\alpha}_k D_f(x^\star, x_{k+1}) + \frac{1}{2}\left(\frac{\tilde{\alpha}_k^2(1 + \sigma^2)}{\tilde{\gamma}_k} - (1 + \tilde{\alpha}_k)\tilde{\alpha}_k \beta_k\right)\|\nabla f(x_{k+1})\|^2\right.$$

$$\left. - D_f(x_k^+, x_{k+1})\right]$$

(C.17)

By moving $\mathbb{E}\left[\mathcal{E}(z_{k+1}^+; \tilde{\gamma}_{k+1})\right]$ to the left side of the inequality to obtain the desired result. $\qquad\square$

Now we prove the theorem 2.2.

*Proof.* Assume $\alpha_k = \frac{2}{k+1}$, $\gamma_k = \alpha_k \tilde{\alpha}_k (1 + \sigma^2)^2 L$ and $\beta_k = \frac{(1+\sigma^2)\alpha_k}{\gamma_k}$. Then

$$\tilde{\alpha}_k \beta_k = \frac{(1 + \sigma^2)\tilde{\alpha}_k \alpha_k}{\gamma_k} = \frac{(1 + \sigma^2)\tilde{\alpha}_k^2}{\tilde{\gamma}_k}$$

(C.18)

Using Lemma C.2 to obtain

$$\mathbb{E}\left[\mathcal{E}(z_{k+1}^+; \tilde{\gamma}_{k+1})\right] \leq (1 + \tilde{\alpha}_k)^{-1}\mathcal{E}(z_k^+; \tilde{\gamma}_k) \leq \Pi_{i=0}^k (1 + \tilde{\alpha}_i)^{-1}\mathcal{E}(z_0^+; \tilde{\gamma}_0)$$

(C.19)

Since $\tilde{\alpha}_k = \frac{2}{k+1+2m}$, then

$$\Pi_{i=0}^k (1 + \tilde{\alpha}_i)^{-1} = \Pi_{i=0}^k \frac{i + 1 + 2m}{i + 3 + 2m} = \frac{(1 + 2m)(2 + 2m)}{(k + 3 + 2m)(k + 2 + 2m)}$$

$\qquad\square$

**Corollary C.2.** *Under the setting of Theorem 2.2, choose $m \geq 0$, $\alpha_k = \frac{2}{k+1}$, $\tilde{\alpha}_k = \frac{\alpha_k}{1+m\alpha_k}$, $\gamma_k = \alpha_k \tilde{\alpha}_k (1+\sigma^2)^2 L$ and $\beta_k = \frac{(1+\sigma^2)\alpha_k}{\gamma_k}$, SHANG++ guarantees to reach an $\varepsilon$-precision at the following interations:*

$$k \geq \sqrt{(1+2m)(2+2m)(f(x_0) - f(x^\star) + \frac{2(1+\sigma^2)^2 L}{(1+2m)^2}\|x_0 - x^\star\|^2)/\varepsilon}$$

**Corollary C.3.** *Under the setting of Theorem 2.2 and C.1, $f(x_k^+) \xrightarrow{a.s.} f(x^\star)$.*

The proof is fully analogous to that of Corollary B.2, with the only difference being the decay-rate parameter $q$ in the final step.

# D VARIANCE DECAY ANALYSIS

We study the variance decay of the Lyapunov energy (B.2)

$$\mathcal{E}_k := \mathcal{E}(z_k^+; \tilde{\gamma}_k) = f(x_k^+) - f(x^\star) + \frac{\tilde{\gamma}_k}{2}\|v_k - x^\star\|^2$$

under the unified stochastic model of SHANG and SHANG++. Throughout we work on a probability space $(\Omega, \mathcal{F}, \mathbb{P})$ with the *post-update* filtration $\mathcal{F}_k := \sigma(x_0, v_0, \zeta_0, \ldots, \zeta_k)$, where each $\zeta_k$ collects the randomness used to form the stochastic gradient at step $k$. We write $g_k := g(x_k, \zeta_k)$ and $g_{k+1} := g(x_{k+1}, \zeta_{k+1})$.

**Assumptions.** We make the following standard assumptions.

- A1. **Smooth convexity.** $f \in \mathcal{S}_{\mu,L}$ with $0 \leq \mu < L < \infty$.
- A2. **Unbiasedness at the query point.** $\mathbb{E}[g_{k+1} \mid \mathcal{F}_k] = \nabla f(x_{k+1})$. Equivalently, with $\xi_{k+1} := g_{k+1} - \nabla f(x_{k+1})$, $\mathbb{E}[\xi_{k+1} \mid \mathcal{F}_k] = 0$.
- A3. **Multiplicative noise scaling (MNS).** $\mathbb{E}[\|\xi_{k+1}\|^2 \mid \mathcal{F}_k] \leq \sigma^2 \|\nabla f(x_{k+1})\|^2$.
- A4. **Bounded conditional kurtosis.** There exists $\chi \geq 1$ such that $\mathbb{E}[\|\xi_{k+1}\|^4 \mid \mathcal{F}_k] \leq \chi(\mathbb{E}[\|\xi_{k+1}\|^2 \mid \mathcal{F}_k])^2$ (e.g., $\chi = 3$ for Gaussian noise).

**Unified stochastic model.** The updates for SHANG/SHANG++ can be written as

$$x_k^+ = x_k - \tilde{\alpha}_k \beta_k\, g_k$$
$$\frac{x_{k+1} - x_k^+}{\tilde{\alpha}_k} = v_k - x_{k+1}$$
$$\frac{v_{k+1} - v_k}{\alpha_k} = \frac{\mu}{\gamma_k}(x_{k+1} - v_{k+1}) - \frac{1}{\gamma_k}g_{k+1} \qquad \text{(D.1)}$$
$$\frac{\gamma_{k+1} - \gamma_k}{\alpha_k} = \mu - \gamma_{k+1}.$$

where $\alpha_k > 0$, $\gamma_k > 0$, and we introduce $\tilde{\alpha}_k = \frac{\alpha_k}{1+m\alpha_k}$ and $\tilde{\gamma}_k = \frac{\gamma_k}{1+m\alpha_k}$ with $m \geq 0$. Equivalently (and crucial for variance analysis), $(x_{k+1}^+, v_{k+1})$ are *affine in the fresh gradient* $g_{k+1}$ while $x_{k+1}$ depends only on past randomness:

$$x_{k+1}^+ = \frac{1}{1+\tilde{\alpha}_k}x_k^+ + \frac{\tilde{\alpha}_k}{1+\tilde{\alpha}_k}v_k - \tilde{\alpha}_{k+1}\beta_{k+1}g_{k+1} = x_{k+1} - \tilde{\alpha}_{k+1}\beta_{k+1}g_{k+1},$$
$$v_{k+1} = \frac{\alpha_k \mu}{(\gamma_k + \alpha_k \mu)(1+\tilde{\alpha}_k)}x_k^+ + \left(\frac{\gamma_k}{\gamma_k + \alpha_k \mu} + \frac{\tilde{\alpha}_k \alpha_k \mu}{(\gamma_k + \alpha_k \mu)(1+\tilde{\alpha}_k)}\right)v_k - \frac{\alpha_k}{\gamma_k + \alpha_k \mu}g_{k+1}$$
$$\gamma_{k+1} = \frac{\alpha_k}{1+\alpha_k}\mu + \frac{1}{1+\alpha_k}\gamma_k$$

$$\text{(D.2)}$$

By the filtration choice, $x_{k+1}$ is $\mathcal{F}_k$-measurable and $g_{k+1}$ uses fresh randomness $\zeta_{k+1}$; hence with $\xi_{k+1} := g_{k+1} - \nabla f(x_{k+1})$ we have $\mathbb{E}[\xi_{k+1} \mid \mathcal{F}_k] = 0$. This linear structure will allow us to bound the one-step fluctuation $\mathcal{E}_{k+1} - \mathbb{E}[\mathcal{E}_{k+1} \mid \mathcal{F}_k]$ and to propagate variance.

**Lemma D.1** (One-step fluctuation). *There exist explicit constants $A_k, B_k, C_k \geq 0$ (functions of $\alpha_k, \tilde{\alpha}_k, \tilde{\gamma}_k, \mu, L$) such that, with $\xi_{k+1}$,*

$$\left|\mathcal{E}_{k+1} - \mathbb{E}[\mathcal{E}_{k+1} \mid \mathcal{F}_k]\right| \leq A_k\sqrt{\mathcal{E}_k}\|\xi_{k+1}\| + B_k\|\xi_{k+1}\|^2 + C_k\mathcal{E}_k$$

*and*

$$A_k = \left(B_x(1 + B_xL)\sqrt{2Lc_1} + B_v\tilde{\gamma}_{k+1}(c_2 + B_v\sqrt{2Lc_1(\tilde{\alpha}_k, \tilde{\gamma}_k, L)})\right)$$

$$B_k = \frac{LB_x^2 + \tilde{\gamma}_{k+1}B_v^2}{2}$$

$$C_k = (LB_x^2 + \tilde{\gamma}_{k+1}B_v^2)Lc_1(\tilde{\alpha}_k, \tilde{\gamma}_k, L)\sigma^2$$

*where $c_1 = \max\{\frac{1}{1+\tilde{\alpha}_k}, \frac{\tilde{\alpha}_k}{1+\tilde{\alpha}_k}\frac{L}{\tilde{\gamma}_k}\}$, $c_2 = \max\{\frac{\alpha_k\sqrt{2\mu}}{(\gamma_k+\alpha_k\mu)(1+\tilde{\alpha}_k)}, \left(\frac{\gamma_k}{\gamma_k+\alpha_k\mu} + \frac{\tilde{\alpha}_k\alpha_k\mu}{(\gamma_k+\alpha_k\mu)(1+\tilde{\alpha}_k)}\right)\sqrt{\frac{2}{\tilde{\gamma}_k}}\}$ when $\mu > 0$ and $c_2 = \sqrt{\frac{2}{\tilde{\gamma}_k}}$ when $\mu = 0$. $B_x = \tilde{\alpha}_{k+1}\beta_{k+1}$ and $B_v = \frac{\alpha_k}{\gamma_k+\alpha_k\mu}$.*

*proof of Lemma D.1.* Using $\xi_{k+1} := g_{k+1} - \nabla f(x_{k+1})$, we can rewrite the updates of $x_{k+1}^+$ and $v_{k+1}$ as

$$x_{k+1}^+ = U_k - \tilde{\alpha}_{k+1}\beta_{k+1}\nabla f(x_{k+1}) - \tilde{\alpha}_{k+1}\beta_{k+1}\xi_{k+1} = \hat{U}_k - B_x\xi_{k+1}$$

$$v_{k+1} = V_k - \frac{\alpha_k}{\gamma_k + \alpha_k\mu}\nabla f(x_{k+1}) - \frac{\alpha_k}{\gamma_k + \alpha_k\mu}\xi_{k+1} = \hat{V}_k - B_v\xi_{k+1} \qquad \text{(D.3)}$$

where $U_k = \frac{1}{1+\tilde{\alpha}_k}x_k^+ + \frac{\tilde{\alpha}_k}{1+\tilde{\alpha}_k}v_k$, $V_k = \frac{\alpha_k\mu}{(\gamma_k+\alpha_k\mu)(1+\tilde{\alpha}_k)}x_k^+ + \left(\frac{\gamma_k}{\gamma_k+\alpha_k\mu} + \frac{\tilde{\alpha}_k\alpha_k\mu}{(\gamma_k+\alpha_k\mu)(1+\tilde{\alpha}_k)}\right)v_k$, $\hat{U}_k = U_k - B_x\nabla f(x_{k+1})$ and $\hat{V}_k = V_k - B_v\nabla f(x_{k+1})$. $B_x = \tilde{\alpha}_{k+1}\beta_{k+1}$ and $B_v = \frac{\alpha_k}{\gamma_k+\alpha_k\mu}$ are positive constants. It should be noted that $U_k, \hat{U}_k, V_k$ and $\hat{V}_k$ are measurable with respect to $\mathcal{F}_k$.

Let's first focus on the left part of $\mathcal{E}_{k+1}$. Expanding $f(x_{k+1}^+) = f(\hat{U}_k - B_x\xi_{k+1})$ at point $\hat{U}_k$ using Taylor series gives

$$f(\hat{U}_k - B_x\xi_{k+1}) = f(\hat{U}_k) - \langle\nabla f(\hat{U}_k), B_x\xi_{k+1}\rangle + r(\hat{U}_k, \xi_{k+1}) \qquad \text{(D.4)}$$

where

$$| r(\hat{U}_k, \xi_{k+1}) | = \left|\int_0^1 \langle\nabla f(\hat{U}_k - tB_x\xi_{k+1}) - \nabla f(\hat{U}_k), -B_x\xi_{k+1}\rangle dt\right| \leq \frac{L}{2}\|B_x\xi_{k+1}\|^2 = \frac{LB_x^2}{2}\|\xi_{k+1}\|^2 \qquad \text{(D.5)}$$

Then

$$| f(x_{k+1}^+) - f(x^\star) - \mathbb{E}\left[f(x_{k+1}^+) - f(x^\star) \mid \mathcal{F}_k\right] |$$

$$= | f(\hat{U}_k - B_x\xi_{k+1}) - f(x^\star) - \mathbb{E}\left[f(\hat{U}_k - B_x\xi_{k+1}) - f(x^\star) \mid \mathcal{F}_k\right] |$$

$$= | -\langle\nabla f(\hat{U}_k), B_x\xi_{k+1}\rangle + r(\hat{U}_k, \xi_{k+1}) - \mathbb{E}\left[r(\hat{U}_k, \xi_{k+1}) \mid \mathcal{F}_k\right] | \qquad \text{(D.6)}$$

$$\leq B_x\|\nabla f(\hat{U}_k)\| \cdot \|\xi_{k+1}\| + \frac{LB_x^2}{2}\|\xi_{k+1}\|^2 + \frac{LB_x^2}{2}\mathbb{E}\left[\|\xi_{k+1}\|^2 \mid \mathcal{F}_k\right]$$

where the last step uses Cauchy-Schwarz inequality and (D.5).

Since $\hat{U}_k = U_k - B_x\nabla f(x_{k+1}) = x_{k+1} - B_x\nabla f(x_{k+1})$ and $x_{k+1} = \frac{1}{1+\tilde{\alpha}_k}x_k^+ + \frac{\tilde{\alpha}_k}{1+\tilde{\alpha}_k}v_k$, by triangle inequality and smooth convexity of $f$, we have

$$\|\nabla f(\hat{U}_k)\| \leq \|\nabla f(\hat{U}_k) - \nabla f(x_{k+1})\| + \|\nabla f(x_{k+1})\|$$

$$\leq L\|\hat{U}_k - x_{k+1}\| + \|\nabla f(x_{k+1})\|$$

$$= (1 + B_xL)\|\nabla f(x_{k+1})\|$$

$$\leq (1 + B_xL)\sqrt{2L}\sqrt{f(x_{k+1}) - f(x^\star)}$$

$$\leq (1 + B_xL)\sqrt{2L}\sqrt{\frac{1}{1+\tilde{\alpha}_k}(f(x_k^+) - f(x^\star)) + \frac{\tilde{\alpha}_k}{1+\tilde{\alpha}_k}(f(v_k) - f(x^\star))} \qquad \text{(D.7)}$$

$$\leq (1 + B_xL)\sqrt{2L}\sqrt{\frac{1}{1+\tilde{\alpha}_k}(f(x_k^+) - f(x^\star)) + \frac{\tilde{\alpha}_k}{1+\tilde{\alpha}_k}\frac{L}{2}\|v_k - x^\star\|^2}$$

$$\leq (1 + B_xL)\sqrt{2Lc_1(\tilde{\alpha}_k, \tilde{\gamma}_k, L)}\sqrt{\mathcal{E}_k}$$

where $c_1(\tilde{\alpha}_k, \tilde{\gamma}_k, L) = \max\{\frac{1}{1+\tilde{\alpha}_k}, \frac{\tilde{\alpha}_k}{1+\tilde{\alpha}_k}\frac{L}{\tilde{\gamma}_k}\}$.

On the other hand,

$$\frac{LB_x^2}{2}\mathbb{E}\left[\|\xi_{k+1}\|^2 \mid \mathcal{F}_k\right] \leq \frac{LB_x^2\sigma^2}{2}\|\nabla f(x_{k+1})\|^2 \leq L^2 B_x^2 \sigma^2 c_1(\tilde{\alpha}_k, \tilde{\gamma}_k, L)\mathcal{E}_k \tag{D.8}$$

Substituting (D.7) and (D.8) back into (D.6), we have

$$| f(x_{k+1}^+) - f(x^\star) - \mathbb{E}\left[f(x_{k+1}^+) - f(x^\star) \mid \mathcal{F}_k\right] |$$

$$\leq B_x(1 + B_x L)\sqrt{2Lc_1(\tilde{\alpha}_k, \tilde{\gamma}_k, L)}\sqrt{\mathcal{E}_k}\|\xi_{k+1}\| + \frac{LB_x^2}{2}\|\xi_{k+1}\|^2 + L^2 B_x^2 \sigma^2 c_1(\tilde{\alpha}_k, \tilde{\gamma}_k, L))\mathcal{E}_k \tag{D.9}$$

For the middle part of $\mathcal{E}_{k+1}$, since

$$\frac{\tilde{\gamma}_{k+1}}{2}\|v_{k+1} - x^\star\|^2 = \frac{\tilde{\gamma}_{k+1}}{2}\|\hat{V}_k - x^\star\|^2 + \frac{\tilde{\gamma}_{k+1}B_v^2}{2}\|\xi_{k+1}\|^2 - \tilde{\gamma}_{k+1}\langle\hat{V}_k - x^\star, B_v\xi_{k+1}\rangle, \tag{D.10}$$

we have

$$| \frac{\tilde{\gamma}_{k+1}}{2}\|v_{k+1} - x^\star\|^2 - \mathbb{E}\left[\frac{\tilde{\gamma}_{k+1}}{2}\|v_{k+1} - x^\star\|^2 \mid \mathcal{F}_k\right] |$$

$$=| -\tilde{\gamma}_{k+1}\langle\hat{V}_k - x^\star, B_v\xi_{k+1}\rangle + \frac{\tilde{\gamma}_{k+1}B_v^2}{2}\left(\|\xi_{k+1}\|^2 - \mathbb{E}\left[\|\xi_{k+1}\|^2 \mid \mathcal{F}_k\right]\right) | \tag{D.11}$$

$$\leq B_v\tilde{\gamma}_{k+1}\|\hat{V}_k - x^\star\| \cdot \|\xi_{k+1}\| + \frac{\tilde{\gamma}_{k+1}B_v^2}{2}\|\xi_{k+1}\|^2 + \frac{\tilde{\gamma}_{k+1}B_v^2}{2}\mathbb{E}\left[\|\xi_{k+1}\|^2 \mid \mathcal{F}_k\right]$$

Using triangle inequality and convexity of $\|\cdot\|$, we have

$$\|\hat{V}_k - x^\star\|$$
$$= \|V_k - x^\star - B_v\nabla f(x_{k+1})\|$$
$$\leq \|\frac{\alpha_k\mu}{(\gamma_k + \alpha_k\mu)(1 + \tilde{\alpha}_k)}x_k^+ + \left(\frac{\gamma_k}{\gamma_k + \alpha_k\mu} + \frac{\tilde{\alpha}_k\alpha_k\mu}{(\gamma_k + \alpha_k\mu)(1 + \tilde{\alpha}_k)}\right)v_k - x^\star\| + B_v\|\nabla f(x_{k+1})\|$$
$$\leq \frac{\alpha_k\mu}{(\gamma_k + \alpha_k\mu)(1 + \tilde{\alpha}_k)}\|x_k^+ - x^\star\| + \left(\frac{\gamma_k}{\gamma_k + \alpha_k\mu} + \frac{\tilde{\alpha}_k\alpha_k\mu}{(\gamma_k + \alpha_k\mu)(1 + \tilde{\alpha}_k)}\right)\|v_k - x^\star\| + B_v\|\nabla f(x_{k+1})\|$$
$$\leq \frac{\alpha_k\mu}{(\gamma_k + \alpha_k\mu)(1 + \tilde{\alpha}_k)}\|x_k^+ - x^\star\| + \left(\frac{\gamma_k}{\gamma_k + \alpha_k\mu} + \frac{\tilde{\alpha}_k\alpha_k\mu}{(\gamma_k + \alpha_k\mu)(1 + \tilde{\alpha}_k)}\right)\|v_k - x^\star\|$$
$$+ B_v\sqrt{2Lc_1(\tilde{\alpha}_k, \tilde{\gamma}_k, L)}\sqrt{\mathcal{E}_k} \tag{D.12}$$

Next, we will consider two cases.

**Case 1:** $\mu > 0$. Using the strong convexity of $f$, we have

$$\|\hat{V}_k - x^\star\|$$
$$\leq \frac{\alpha_k\sqrt{2\mu}}{(\gamma_k + \alpha_k\mu)(1 + \tilde{\alpha}_k)}\sqrt{f(x_k^+) - f(x^\star)} + \left(\frac{\gamma_k}{\gamma_k + \alpha_k\mu} + \frac{\tilde{\alpha}_k\alpha_k\mu}{(\gamma_k + \alpha_k\mu)(1 + \tilde{\alpha}_k)}\right)\sqrt{\frac{2}{\tilde{\gamma}_k}}\sqrt{\frac{\tilde{\gamma}_k}{2}}\|v_k - x^\star\|$$
$$+ B_v\sqrt{2Lc_1(\tilde{\alpha}_k, \tilde{\gamma}_k, L)}\sqrt{\mathcal{E}_k}$$
$$\leq \left(c_2(\tilde{\alpha}, \mu, \gamma_k) + B_v\sqrt{2Lc_1(\tilde{\alpha}_k, \tilde{\gamma}_k, L)}\right)\sqrt{\mathcal{E}_k} \tag{D.13}$$

where $c_2(\tilde{\alpha}, \mu, \gamma_k) = \max\{\frac{\alpha_k\sqrt{2\mu}}{(\gamma_k + \alpha_k\mu)(1+\tilde{\alpha}_k)}, \left(\frac{\gamma_k}{\gamma_k+\alpha_k\mu} + \frac{\tilde{\alpha}_k\alpha_k\mu}{(\gamma_k+\alpha_k\mu)(1+\tilde{\alpha}_k)}\right)\sqrt{\frac{2}{\tilde{\gamma}_k}}\}$. Thus,

$$| \frac{\tilde{\gamma}_{k+1}}{2}\|v_{k+1} - x^\star\|^2 - \mathbb{E}\left[\frac{\tilde{\gamma}_{k+1}}{2}\|v_{k+1} - x^\star\|^2 \mid \mathcal{F}_k\right] |$$

$$\leq B_v\tilde{\gamma}_{k+1}\left(c_2(\tilde{\alpha}, \mu, \gamma_k) + B_v\sqrt{2Lc_1(\tilde{\alpha}, \mu, L)}\right)\sqrt{\mathcal{E}_k}\|\xi_{k+1}\| + \frac{\tilde{\gamma}_{k+1}B_v^2}{2}\|\xi_{k+1}\|^2 + \tilde{\gamma}_{k+1}B_v^2 L\sigma^2 c_1(\tilde{\alpha}_k, \tilde{\gamma}_k, L)\mathcal{E}_k \tag{D.14}$$

Combining (D.9) and (D.14), we have

$$| \mathcal{E}_{k+1} - \mathbb{E}\left[\mathcal{E}_{k+1} \mid \mathcal{F}_k\right] |$$

$$\leq \left(B_x(1 + B_x L)\sqrt{2Lc_1(\tilde{\alpha}_k, \tilde{\gamma}_k, L)} + B_v \tilde{\gamma}_{k+1}(c_2(\tilde{\alpha}, \mu, \gamma_k) + B_v \sqrt{2Lc_1(\tilde{\alpha}_k, \tilde{\gamma}_k, L)})\right) \sqrt{\mathcal{E}_k}\|\xi_{k+1}\|$$

$$+ \frac{LB_x^2 + \tilde{\gamma}_{k+1} B_v^2}{2}\|\xi_{k+1}\|^2 + (LB_x^2 + \tilde{\gamma}_{k+1} B_v^2)Lc_1(\tilde{\alpha}_k, \tilde{\gamma}_k, L)\sigma^2 \mathcal{E}_k \tag{D.15}$$

**Case 2:** $\mu = 0$.

$$\|\hat{V}_k - x^\star\| \leq \|v_k - x^\star\| + B_v \sqrt{2Lc_1(\tilde{\alpha}_k, \tilde{\gamma}_k, L)}\sqrt{\mathcal{E}_k}$$

$$\leq (\sqrt{\frac{2}{\tilde{\gamma}_k}} + B_v \sqrt{2Lc_1(\tilde{\alpha}_k, \tilde{\gamma}_k, L)})\sqrt{\mathcal{E}_k} \tag{D.16}$$

Thus,

$$| \frac{\tilde{\gamma}_{k+1}}{2}\|v_{k+1} - x^\star\|^2 - \mathbb{E}\left[\frac{\tilde{\gamma}_{k+1}}{2}\|v_{k+1} - x^\star\|^2 \mid \mathcal{F}_k\right] |$$

$$\leq B_v \tilde{\gamma}_{k+1}(\sqrt{\frac{2}{\tilde{\gamma}_k}} + B_v \sqrt{2Lc_1(\tilde{\alpha}_k, \tilde{\gamma}_k, L)})\sqrt{\mathcal{E}_k}\|\xi_{k+1}\| + \frac{\tilde{\gamma}_{k+1}B_v^2}{2}\|\xi_{k+1}\|^2 + \tilde{\gamma}_{k+1}B_v^2 L\sigma^2 c_1(\tilde{\alpha}_k, \tilde{\gamma}_k, L)\mathcal{E}_k \tag{D.17}$$

Combining (D.9) and (D.17), we have

$$| \mathcal{E}_{k+1} - \mathbb{E}\left[\mathcal{E}_{k+1} \mid \mathcal{F}_k\right] |$$

$$\leq \left(B_x(1 + B_x L)\sqrt{2Lc_1(\tilde{\alpha}_k, \tilde{\gamma}_k, L)} + B_v \tilde{\gamma}_{k+1}(\sqrt{\frac{2}{\tilde{\gamma}_k}} + B_v \sqrt{2Lc_1(\tilde{\alpha}_k, \tilde{\gamma}_k, L)})\right) \sqrt{\mathcal{E}_k}\|\xi_{k+1}\|$$

$$+ \frac{LB_x^2 + \tilde{\gamma}_{k+1} B_v^2}{2}\|\xi_{k+1}\|^2 + (LB_x^2 + \tilde{\gamma}_{k+1} B_v^2)Lc_1(\tilde{\alpha}_k, \tilde{\gamma}_k, L)\sigma^2 \mathcal{E}_k \tag{D.18}$$

$\square$

**Proposition D.1** (Conditional variance bound). *Let $S_k := 2L\sigma^2 c_1(\tilde{\alpha}_k, \tilde{\gamma}_k, L)$ with $c_1(\tilde{\alpha}_k, \tilde{\gamma}_k, L) = \max\{\frac{1}{1+\tilde{\alpha}_k}, \frac{\tilde{\alpha}_k}{1+\tilde{\alpha}_k}\frac{L}{\tilde{\gamma}_k}\}$. Under assumptions (A2)–(A4) and the setting of Lemma D.1 (In particular, stepsizes and hence $A_k, B_k, C_k, S_k$ are $\mathcal{F}_k$-measurable),*

$$\mathrm{Var}(\mathcal{E}_{k+1} \mid \mathcal{F}_k) \leq K_{2,k}\mathcal{E}_k^2, \qquad K_{2,k} = 3\left(A_k^2 S_k + \chi B_k^2 S_k^2 + C_k^2\right)$$

*proof of Proposition D.1.* By the definition of conditional variance,

$$\mathrm{Var}(\mathcal{E}_{k+1} \mid \mathcal{F}_k) = \mathbb{E}\left[\left(\mathcal{E}_{k+1} - \mathbb{E}\left[\mathcal{E}_{k+1} \mid \mathcal{F}_k\right]\right)^2 \mid \mathcal{F}_k\right] \tag{D.19}$$

From Lemma D.1 and inequality $(x + y + z)^2 \leq 3(x^2 + y^2 + z^2)$,

$$\left(\mathcal{E}_{k+1} - \mathbb{E}\left[\mathcal{E}_{k+1} \mid \mathcal{F}_k\right]\right)^2 \leq 3\left(A_k^2 \mathcal{E}_k\|\xi_{k+1}\|^2 + B_k^2\|\xi_{k+1}\|^4 + C_k^2 \mathcal{E}_k^2\right) \tag{D.20}$$

Since $A_k, B_k, C_k$ and $\mathcal{E}_k$ are all measurable with respect to the $\sigma$-algebra $\mathcal{F}_k$. Using assumptions (A2-A4) yields

$$\mathbb{E}\left[\|\xi_{k+1}\|^2 \mid \mathcal{F}_k\right] \leq \sigma^2\|\nabla f(x_{k+1})\|^2 \leq 2L\sigma^2 c_1 \mathcal{E}_k = S_k \mathcal{E}_k \tag{D.21}$$

and

$$\mathbb{E}\left[\|\xi_{k+1}\|^4 \mid \mathcal{F}_k\right] \leq \chi\left(\mathbb{E}\left[\|\xi_{k+1}\|^2 \mid \mathcal{F}_k\right]\right)^2 \leq \chi S_k^2 \mathcal{E}_k^2 \tag{D.22}$$

Taking $\mathbb{E}\left[\cdot \mid \mathcal{F}_k\right]$ in the previous inequality gives

$$\mathrm{Var}(\mathcal{E}_{k+1} \mid \mathcal{F}_k) \leq 3(A_k^2 S_k + \chi B_k^2 S_k^2 + C_k^2)\mathcal{E}_k^2 \tag{D.23}$$

$\square$

**Theorem D.1** (Geometric variance decay). *Assume the drift inequality (from the expectation analysis)*

$$\mathbb{E}[\mathcal{E}_{k+1} \mid \mathcal{F}_k] \leq q\mathcal{E}_k \qquad \text{for some } q \in (0, 1), \tag{D.24}$$

*and assumptions (A2)–(A4) hold. Let $K_{2,k}$ be given in Proposition D.1 and suppose $K_2 := \sup_k K_{2,k} < 1 - q^2$ satisfied. Then with $\theta := q^2 + K_2 \in (0, 1)$, for all $k \geq 0$, given initial $\mathcal{E}_0$,*

$$\mathrm{Var}(\mathcal{E}_{k+1}) \leq \mathcal{E}_0^2 \theta^{k+1}$$

*Proof.* By the law of total variance and Proposition D.1,

$$\text{Var}(\mathcal{E}_{k+1}) = \mathbb{E}\big[Var(\mathcal{E}_{k+1} \mid \mathcal{F}_k)\big] + \text{Var}\big(\mathbb{E}[\mathcal{E}_{k+1} \mid \mathcal{F}_k]\big) \leq K_2 \mathbb{E}[\mathcal{E}_k^2] + q^2 \text{Var}(\mathcal{E}_k). \quad \text{(D.25)}$$

Since $\mathbb{E}[\mathcal{E}_k^2] = \text{Var}(\mathcal{E}_k) + \big(\mathbb{E}[\mathcal{E}_k]\big)^2$ and (Eq.(D.24)), we get

$$\text{Var}(\mathcal{E}_{k+1}) \leq (K_2 + q^2)\,\text{Var}(\mathcal{E}_k) + K_2\big(\mathbb{E}[\mathcal{E}_k]\big)^2 \leq (K_2 + q^2)\,\text{Var}(\mathcal{E}_k) + K_2(\mathbb{E}[\mathcal{E}_0])^2 q^{2k} \quad \text{(D.26)}$$

Solving this linear recursion yields

$$\text{Var}(\mathcal{E}_{k+1}) \leq (K_2+q^2)^{k+1}\,\text{Var}(\mathcal{E}_0) + K_2(\mathbb{E}[\mathcal{E}_0])^2 \sum_{j=0}^{k}(K_2+q^2)^{k-j}q^{2j} \leq (K_2+q^2)^{k+1}(\text{Var}(\mathcal{E}_0)+(\mathbb{E}[\mathcal{E}_0])^2)$$

$$\text{(D.27)}$$

Since $\mathcal{E}_0$ is given by the initial point $x_0 = v_0$, it is a constant ,then $\text{Var}(\mathcal{E}_0) = 0$ and $\mathbb{E}[\mathcal{E}_0] = \mathcal{E}_0$. □

**Corollary D.1** (Upper bound of $K_{2,k}$ in strongly convex setting). *Define $\kappa = \frac{L}{\mu}$ is the condition number of $f$. Under the setting of Theorem B.1-2.1 and Assumptions (A1)-(A4), with $K_{2,k} = 3(A_k^2 S_k + \chi B_k^2 S_k^2 + C_k^2)$ defined above, we have the explicit upper bound*

*(1) For SHANG,*

$$K_2 \leq \begin{cases} 12a_0^2\sigma^2((3+\sigma^2)a_0+1)^2 + 12(\chi+1)a_0^4\sigma^4 & \alpha\kappa \leq 1 \\ 12a_0^3\sigma^2\sqrt{\kappa}(1+(3+\sigma^2)a_0^{\frac{3}{2}}\kappa^{\frac{1}{4}})^2 + 12(\chi+1)\,a_0^6\,\sigma^4\,\kappa & \alpha\kappa \geq 1 \end{cases}$$

*(2) For SHANG++,*

$$K_2 \leq \begin{cases} 12a_0^2\sigma^2\big((3+\sigma^2)a_0+1\big)^2 + 12(\chi+1)a_0^4\sigma^4, & \tilde{\alpha}\kappa \leq 1 \\ 12a_0^3\sigma^2\sqrt{\kappa}\big(1+(3+\sigma^2)a_0^{3/2}\kappa^{1/4}\big)^2 + 12(\chi+1)a_0^6\sigma^4\kappa, & \tilde{\alpha}\kappa \geq 1 \end{cases}$$

*Proof.* **Case 1: SHANG.** When $m = 0$, scheme (D.1) is algorithm SHANG. From Theorem B.1, when $\gamma = \mu$, $\alpha = \frac{1}{(1+\sigma^2)\sqrt{\kappa}}$ and $\beta = \frac{(1+\sigma^2)\alpha}{\mu}$, we have

$$\mathbb{E}[\mathcal{E}_{k+1} \mid \mathcal{F}_k] \leq (1+\alpha)^{-1}\mathcal{E}_k = q\mathcal{E}_k \quad \text{(D.28)}$$

and

$$A = A_k = \frac{\alpha^2}{\mu}\big(1+\sigma^2+(1+\sigma^2)^2\alpha^2\kappa + \frac{1}{(1+\alpha)^2}\big)\sqrt{2Lc_1} + \frac{\alpha}{1+\alpha}c_2$$

$$B = B_k = \frac{\alpha^2}{2\mu}((1+\sigma^2)^2\alpha^2\kappa + \frac{1}{(1+\alpha)^2})$$

$$C = C_k = \frac{\alpha^2}{\mu}((1+\sigma^2)^2\alpha^2\kappa + \frac{1}{(1+\alpha)^2})L\sigma^2 c_1$$

$$S = S_k = 2L\sigma^2 c_1$$

where $c_1 = \max\{\frac{1}{1+\alpha}, \frac{\alpha}{1+\alpha}\kappa\}$ and $c_2 = \frac{1+\alpha+\alpha^2}{(1+\alpha)^2}\sqrt{\frac{2}{\mu}}$.

**(1): Assume $\alpha\kappa \leq 1$, i.e., $\kappa \leq (1+\sigma^2)^2$, so that $c_1 = \frac{1}{1+\alpha}$.**

Since $c_1 = \frac{1}{1+\alpha}$ and $\alpha^2\kappa = \frac{1}{(1+\sigma^2)^2} \leq 1$, we bound each term in $K_2$.

For the $B^2S^2$ term, using $B = \frac{\alpha^2}{2\mu}((1+\sigma^2)^2\alpha^2\kappa + \frac{1}{(1+\alpha)^2})$,

$$B^2S^2 = \Big[\frac{\alpha^2}{2\mu}\big((1+\sigma^2)^2\alpha^2\kappa + \frac{1}{(1+\alpha)^2}\big)\Big]^2 \cdot (2\mu\kappa\sigma^2 c_1)^2$$

$$= \alpha^4\kappa^2\sigma^4 c_1^2\Big((1+\sigma^2)^2\alpha^2\kappa + \frac{1}{(1+\alpha)^2}\Big)^2 \leq 4\,a_0^4\,\sigma^4, \quad \text{(D.29)}$$

where we used $c_1 \leq 1$ and $\alpha^4\kappa^2 = \frac{1}{(1+\sigma^2)^4}$. We denote $a_0 = \frac{1}{1+\sigma^2}$. Hence $3\chi B^2S^2 \leq 12\chi a_0^4\sigma^4$.

For the $C^2$ term, note $C = 2BLc_1\sigma^2$ implies $C^2 = B^2S^2$. Hence

$$C^2 \leq 4\,a_0^4\,\sigma^4, \tag{D.30}$$

so $3C^2 \leq 12\,a_0^4\,\sigma^4$.

For the $A^2S$ term, splitting $A = A_1 + A_2$ with

$$A_1 := \frac{\alpha^2}{\mu}\left((1+\sigma^2) + (1+\sigma^2)^2\alpha^2\kappa + \frac{1}{(1+\alpha)^2}\right)\sqrt{2Lc_1}, \qquad A_2 := \frac{\alpha}{1+\alpha}\,c_2,$$

For $A_2$, since $c_2 = \frac{1+\alpha+\alpha^2}{(1+\alpha)^2}\sqrt{2/\mu} \leq \sqrt{2/\mu}$,

$$A_2^2S = \frac{\alpha^2}{(1+\alpha)^2}\,c_2^2 \cdot 2\mu\kappa\sigma^2 c_1 \leq 4\,\kappa\sigma^2 c_1 \cdot \frac{\alpha^2}{(1+\alpha)^2} = 4\,\sigma^2 \cdot \frac{\alpha^2\kappa}{(1+\alpha)^3} \leq 4a_0^2\sigma^2 \tag{D.31}$$

For $A_1$, using $c_1 = \frac{1}{1+\alpha}$ and $\alpha^2\kappa = a_0^2 \leq 1$,

$$A_1^2S = \left[\frac{\alpha^2}{\mu}\sqrt{2Lc_1}\right]^2\left((1+\sigma^2) + (1+\sigma^2)^2\alpha^2\kappa + \frac{1}{(1+\alpha)^2}\right)^2 \cdot 2L\sigma^2 c_1$$

$$= 4\,\alpha^4\kappa^2 c_1^2\,\sigma^2\left((1+\sigma^2)^2 + 1 + \frac{1}{(1+\alpha)^2}\right)^2 \leq 4\,a_0^4\,\sigma^2 \cdot (3+\sigma^2)^2 = 4(3+\sigma^2)^2 a_0^4\sigma^2 \tag{D.32}$$

Therefore, using $(x+y)^2 \leq (1+\tau)x^2 + (1+1/\tau)y^2$ with $\tau = \sqrt{A_2^2S/A_1^2S}$:

$$3A^2S \leq 3(\sqrt{A_1^2S} + \sqrt{A_2^2S})^2 \leq 3(2(3+\sigma^2)a_0^2\sigma + 2a_0\sigma)^2 = 12a_0^2\sigma^2((3+\sigma^2)a_0 + 1)^2 \tag{D.33}$$

Combining (D.29)-(D.33), we have

$$K_2 \leq 12a_0^2\sigma^2((3+\sigma^2)a_0 + 1)^2 + 12(\chi+1)a_0^4\sigma^4 \tag{D.34}$$

**(2): Assume** $\alpha\kappa \geq 1$**, i.e.,** $\kappa \geq (1+\sigma^2)^2$**, so that** $c_1 = \frac{\alpha}{1+\alpha}\kappa$**.**

For the $B^2S^2$ and $C^2$ terms. We have

$$B^2S^2 = \frac{\alpha^4\kappa^2\sigma^4}{(1+\alpha)^2}\left((1+\sigma^2)^2\alpha^2\kappa + \frac{1}{(1+\alpha)^2}\right)^2 \leq \frac{\alpha^6\kappa^4\sigma^4}{(1+\alpha)^2} \cdot 4 \leq 4\,a_0^6\,\sigma^4\,\kappa \tag{D.35}$$

Hence

$$3(\chi B^2S^2 + C^2) \leq 12(\chi+1)\,a_0^6\,\sigma^4\,\kappa \tag{D.36}$$

For the $A^2S$ term,

$$A_2^2S = \frac{\alpha^2}{(1+\alpha)^2}c_2^2 \cdot 2L\sigma^2 c_1 \leq \frac{\alpha^2}{(1+\alpha)^2} \cdot \frac{2}{\mu} \cdot 2\mu\kappa\sigma^2 c_1 = 4\,\sigma^2\frac{\alpha^3}{(1+\alpha)^3}\kappa^2 \leq 4\,a_0^3\,\sigma^2\sqrt{\kappa} \tag{D.37}$$

Moreover,

$$A_1^2S = \frac{4\,\alpha^6\sigma^2\,\kappa^4}{(1+\alpha)^2}\left((1+\sigma^2) + (1+\sigma^2)^2\alpha^2\kappa + \frac{1}{(1+\alpha)^2}\right)^2 \leq \frac{4\,\alpha^6\sigma^2\,\kappa^4}{(1+\alpha)^2} \cdot (3+\sigma^2)^2 \leq 4(3+\sigma^2)^2\,a_0^6\,\sigma^2\,\kappa. \tag{D.38}$$

Combining (D.37) and (D.38),

$$3A^2S \leq 3(\sqrt{A_1^2S} + \sqrt{A_2^2S})^2 \leq 12a_0^3\sigma^2\sqrt{\kappa}(1 + (3+\sigma^2)a_0^{\frac{3}{2}}\kappa^{\frac{1}{4}})^2 \tag{D.39}$$

Adding (D.35) - (D.39) yields

$$K_2 \leq 12a_0^2\sigma^2\sqrt{\kappa}(1 + (3+\sigma^2)a_0^{\frac{3}{2}}\kappa^{\frac{1}{4}})^2 + 12(\chi+1)\,a_0^6\,\sigma^4\,\kappa \tag{D.40}$$

**Case 2: SHANG++.** When $m = 1$, scheme (D.1) is algorithm SHANG++. From Theorem 2.1, when $\gamma = \mu$, $\tilde{\alpha} = \frac{1}{(1+\sigma^2)\sqrt{\kappa}}$, $\alpha = \frac{\tilde{\alpha}}{1-\tilde{\alpha}}$ and $\beta = \frac{(1+\sigma^2)\tilde{\alpha}}{\mu}$, we have

$$\mathbb{E}[\mathcal{E}_{k+1} \mid \mathcal{F}_k] \leq (1+\alpha)^{-1}\mathcal{E}_k = q\mathcal{E}_k \tag{D.41}$$

and

$$A = A_k = \frac{\tilde{\alpha}^2}{\mu}\left(1 + \sigma^2 + (1+\sigma^2)^2\tilde{\alpha}^2\kappa + 1\right)\sqrt{2Lc_1} + \tilde{\alpha}^2 c_2$$

$$B = B_k = \frac{\tilde{\alpha}^2}{2\mu}((1+\sigma^2)^2\tilde{\alpha}^2\kappa + 1)$$

$$C = C_k = \tilde{\alpha}c_1\sigma^2\kappa(\tilde{\alpha}^2(1+\sigma^2)^2\kappa + 1)$$

$$S = S_k = 2L\sigma^2 c_1$$

where $c_1 = \max\{\frac{1}{1+\tilde{\alpha}}, \frac{\tilde{\alpha}}{1+\tilde{\alpha}}\kappa\}$ and $c_2 = \max\{\frac{\tilde{\alpha}}{1+\tilde{\alpha}}\sqrt{\frac{2}{\mu}}, (\frac{1}{1+\alpha} + \frac{\tilde{\alpha}}{1+\tilde{\alpha}}\frac{\alpha}{1+\alpha})\sqrt{\frac{2}{\mu}}\}$.

Similar to the derivation of SHANG, we have

**(1): Case $\tilde{\alpha}\kappa \le 1$.** In this case $\kappa \le (1+\sigma^2)^2$ and hence $c_1 = \frac{1}{1+\tilde{\alpha}} \le 1$.

$$K_2 \le 12a_0^2\sigma^2\left((3+\sigma^2)a_0 + 1\right)^2 + 12(\chi+1)a_0^4\sigma^4 \tag{D.42}$$

**(2): Case $\tilde{\alpha}\kappa \ge 1$.** In this case $\kappa \ge (1+\sigma^2)^2$ and $c_1 = \frac{\tilde{\alpha}}{1+\tilde{\alpha}}\kappa$.

$$K_2 \le 12a_0^3\sigma^2\sqrt{\kappa}\left(1 + (3+\sigma^2)a_0^{3/2}\kappa^{1/4}\right)^2 + 12(\chi+1)a_0^6\sigma^4\kappa \tag{D.43}$$

$\square$

**When does variance decay hold?** By Theorem D.1, geometric variance decay

$$\text{Var}(\mathcal{E}_k) \le \mathcal{E}_0^2(q^2 + K_2)^k$$

holds whenever $K_2 < 1 - q^2$, where $q = (1+\alpha)^{-1}$. The bounds in Corollary D.1 make this condition directly checkable as a function of the condition number $\kappa = L/\mu$, the noise level $\sigma^2$ via $a_0 = (1+\sigma^2)^{-1}$, and the stepsize $\alpha$:

- In the *low-condition* regime (the branch with smaller $c_1$), $K_2$ scales like

$$K_2 = \mathcal{O}(a_0^2\sigma^2) + \mathcal{O}(a_0^4\sigma^4)$$

  for both SHANG and SHANG++, whereas $1 - q^2 = \Theta(\alpha) = \Theta(a_0/\sqrt{\kappa})$.

- In the *high-condition* regime (the branch with larger $c_1$), the leading term is

$$K_2 = \mathcal{O}(a_0^3\sigma^2\sqrt{\kappa}) + \mathcal{O}(a_0^6\sigma^4\kappa),$$

  while we still have $1 - q^2 = \Theta(a_0/\sqrt{\kappa})$. The same scaling holds for both SHANG and SHANG++; only the constant factors differ mildly.

Thus, for fixed $\kappa$, smaller noise (larger $a_0$) and moderate stepsizes make $K_2 < 1 - q^2$ easier to satisfy; for large $\kappa$, the $\mathcal{O}(\sqrt{\kappa})$ factor in the leading term of $K_2$ becomes the main bottleneck.

**How to enforce the condition in practice.** Two standard knobs guarantee $K_2 < 1 - q^2$ without fine tuning:

1. *Stepsize damping.* Replace $\alpha$ by $\beta\alpha$ with $\beta \in (0, 1]$. Then the leading term in $K_2$ scales like $\mathcal{O}(\beta^3)$, whereas $1 - q^2$ scales like $\mathcal{O}(\beta)$ (for both SHANG and SHANG++); hence there exists $\beta_0 = \beta_0(\kappa, \sigma^2, \chi) \in (0, 1]$ such that $K_2 < 1 - q^2$ for all $\beta \le \beta_0$.

2. *Mini-batching or averaging multiple independent estimates.* Replacing $\sigma^2$ by $\sigma^2/M$ reduces the leading term in $K_2$ by a factor $1/M$ while leaving $1 - q^2$ essentially unchanged; the explicit constants in the corollary yield simple batch-size thresholds (e.g., $M \gtrsim \sigma^2\sqrt{\kappa}$ up to the displayed constants). Averaging $M$ independent estimates incurs almost no extra computational cost compared with performing $M$ successive iterations using a single estimate.

## E    SNAG as a Discretization of the HNAG Flow

Under the multiplicative noise assumption, one of the most recent first-order stochastic methods designed to overcome the divergence of NAG and accelerate SGD is the Stochastic Nesterov Accelerated Gradient (SNAG) method (Nesterov, 2012) (Hermant et al., 2025). Its iteration reads:

$$
\begin{aligned}
x_{k+1} &= \hat{\alpha}_{k+1} x_k + (1 - \hat{\alpha}_{k+1}) v_{k+1} - \hat{\alpha}_{k+1} s\, g(x_k), \\
v_{k+1} &= \hat{\beta} v_k + (1 - \hat{\beta}) x_k - \eta_k g(x_k),
\end{aligned}
\tag{E.1}
$$

where $g(x_k)$ is a stochastic gradient estimator, and $\hat{\alpha}_{k+1}$, $s$, $\hat{\beta}$, and $\eta_k$ are parameters.

By reparameterizing as

$$
\hat{\alpha}_{k+1} = \frac{1}{1 + \alpha_{k+1}}, \quad s = \alpha_{k+1} \beta_{k+1}, \quad \hat{\beta} = \frac{1}{1 + \frac{\alpha_{k+1}\mu}{\gamma_{k+1}}}, \quad \eta_k = \frac{1}{1 + \frac{\alpha_{k+1}\mu}{\gamma_{k+1}}} \frac{\alpha_{k+1}}{\gamma_{k+1}}, \tag{E.2}
$$

the SNAG scheme (E.1) becomes equivalent to the following update:

$$
\begin{aligned}
\frac{x_{k+1} - x_k}{\alpha_{k+1}} &= v_{k+1} - x_{k+1} - \beta_{k+1} g(x_k), \\
\frac{v_{k+1} - v_k}{\alpha_{k+1}} &= \frac{\mu}{\gamma_{k+1}} (x_k - v_{k+1}) - \frac{1}{\gamma_{k+1}} g(x_k), \\
\frac{\gamma_{k+1} - \gamma_k}{\alpha_{k+1}} &\le \mu - \gamma_{k+1}.
\end{aligned}
\tag{E.3}
$$

Hence, SNAG can be interpreted as a new discretization of the HNAG flow (2.3).

**Parameter choices.**    For convex objectives $f \in \mathcal{S}_{0,L}^{1,1}$, Hermant et al. (2025) shows that the optimal parameters are

$$
s = \frac{1}{L(1 + \sigma^2)}, \quad \eta_k = \frac{k+1}{2L(1 + \sigma^2)^2}, \quad \hat{\beta} = 1, \quad \hat{\alpha}_k = \frac{\frac{k^2}{k+1}}{2 + \frac{k^2}{k+1}}.
$$

This leads to

$$
\alpha_{k+1} = \frac{2}{k + 1 - \frac{k+1}{k+2}}, \quad \alpha_{k+1}\beta_{k+1} = \frac{1}{L(1 + \sigma^2)}, \quad \gamma_{k+1} = \alpha_{k+1} \frac{2}{k+1} (1 + \sigma^2)^2 L.
$$

For strongly convex objectives $f \in \mathcal{S}_{\mu,L}$, the optimal parameters become

$$
s = \frac{1}{L(1 + \sigma^2)}, \quad \eta_k = \eta = \frac{1}{(1 + \sigma^2)\sqrt{\mu L}}, \quad \hat{\beta} = 1 - \frac{1}{1 + \sigma^2}\sqrt{\frac{\mu}{L}}, \quad \hat{\alpha}_k = \hat{\alpha} = \frac{1}{1 + \frac{1}{1+\sigma^2}\sqrt{\frac{\mu}{L}}}.
$$

Consequently,

$$
\alpha = \frac{1}{1 + \sigma^2}\sqrt{\frac{\mu}{L}}, \quad \alpha\beta = \frac{1}{L(1 + \sigma^2)}, \quad \gamma = \mu(1 - \alpha).
$$

The condition $\gamma = \mu(1 - \alpha)$ indicates that, in the strongly convex case, the update for $v$ is more accurately viewed as applying a rescaled step size $\tilde{\alpha} = \frac{\alpha}{1-\alpha}$ to the $v$–dynamics of the HNAG flow:

$$
\frac{v_{k+1} - v_k}{\tilde{\alpha}} = x_k - v_{k+1} - \frac{1}{\mu} g(x_k).
$$

In summary, the above parameter rearrangements confirm that the optimal choices in SNAG are consistent with those obtained from various discretization schemes of the HNAG flow, see Chen & Luo (2021) for details.

## F  RELATED WORK

Accelerated variants of SGD have been extensively studied. A natural idea is to combine SGD with first-order momentum methods, such as the Heavy-Ball (HB) and Nesterov's Accelerated Gradient (NAG) algorithms, in order to achieve faster convergence through momentum. However, in stochastic settings, gradient noise often weakens or even destroys the acceleration effect. Kidambi et al. (2018); Sutskever et al. (2013); Yuan et al. (2016); Nemirovski et al. (2009); Ghadimi & Lan (2012) have shown that both HB and NAG fail to accelerate SGD in expectation under gradient noise. In practice, the apparent superiority of momentum methods largely stems from large mini-batching, which reduces the variance of stochastic gradients and brings the stochastic dynamics closer to the deterministic regime.

To address this, many efforts have been devoted to developing truly accelerated first order stochastic momentum methods. Starting from (Jain et al., 2018), a series of accelerated stochastic algorithms have been proposed (Liu & Belkin, 2020; Vaswani et al., 2019; Even et al., 2021; Bollapragada et al., 2022; Laborde & Oberman, 2020; Gupta et al., 2024; Hermant et al., 2025), aiming to preserve acceleration while maintaining robustness under stochastic noise. These methods introduce various variance-control mechanisms, adaptive damping, or noise-aware correction terms to balance efficiency and stability.

Noise modeling is essential for understanding and improving stochastic optimization. While early studies assume additive noise with bounded variance, empirical studies show that SGD noise is anisotropic (concentrated in a low-rank subspace) and state-dependent in deep neural networks (Wu et al., 2022a), and often exhibits heavy-tailed non-Gaussian fluctuations (Zhao et al., 2024; Hodgkinson & Mahoney, 2021; Zhou et al., 2020). Additive noise with bounded variance often fails in deep learning, where gradient noise may scale with the signal norm or exhibit low-rank and heavy-tailed characteristics (Wojtowytsch, 2023; Wu et al., 2022b). In particular, the noise variance scales with the loss or gradient norm while covariance spectra are highly skewed, with only a few large eigen-directions. These non-classical properties observed in practice as multiplicative (Wu et al., 2019; Gupta et al., 2024; Hodgkinson & Mahoney, 2021), low-rank/degenerate(Damian et al., 2021; Li et al., 2022; Bassily et al., 2018; Wojtowytsch, 2021; 2023), and heavy-tailed gradient noise. These insights motivate the design of optimizers that are resilient to complex, non-Gaussian noise structures.

