# OpenReview forum: "SHANG++: Robust Stochastic Acceleration under Multiplicative Noise"
_ICLR.cc/2026/Conference — Submitted to ICLR 2026_

### Official Review · Reviewer_1Raw · 2025-10-17

**Soundness:** 1
**Presentation:** 1
**Contribution:** 2
**Rating:** 2
**Confidence:** 3

**Summary:**

The authors propose SHANG++, a novel optimizer for empirical risk minimization, particularly in the context of ANN training.
The method is claimed to offer increased robustness to multiplicative gradient noise.
SHANG++ builds upon the SHANG algorithm, providing faster theoretical convergence and enhanced noise robustness.
The authors present a detailed convergence proof and evaluate their optimizer on the computer vision datasets MNIST and CIFAR-10/100.

**Strengths:**

- Developing more noise robust optimizers with low HP sensitivity is a problem of great interest.

**Weaknesses:**

- The paper's core idea appears to rely heavily on HNAG by Chen & Luo (2021), however, their work is not adequately discussed. For readers unfamiliar with HNAG (such as myself), Equation 2.1 is therefore difficult to interpret. As a result, both the motivation and the novel contribution of the paper remain unclear.
- The theoretical section is very difficult to follow. Rather than devoting multiple pages to dense technical proofs and equations, the paper should first clearly state its assumptions, theoretical setting, and main conclusions in the body text. The detailed derivations and proofs can be moved to an appendix. While the proofs themselves might be sound (which I did not check in detail), the overall theoretical context is insufficiently explained.
- The empirical evaluation is weak. Using LeNet-5 on MNIST is outdated and does not meaningfully demonstrate the optimizer's potential for modern applications. Although experiments on CIFAR are somewhat more relevant, additional experiments on larger models and datasets (e.g., a small ViT on ImageNet) would make the evaluation more convincing. Moreover, the baseline results are unreasonably poor. Even though I am not sure about the impact of the low batch size, LeNet-5 trained with SGD should reach around 99% accuracy on MNIST, whereas your reported 91% is close to the performance of a linear classifier. Similarly, achieving only 68% accuracy for ResNet-34 on CIFAR-10 suggests serious implementation or training issues. A strong and competitive baseline must be established before claiming improvements from SHANG++.
- The presentation of results is very bad. Many figures lack axis labels (e.g., it is unclear what is plotted on the x-axis of Fig. 3.1). The titles read like folder names. The figures and especially the legends are very small. Figures 3.1 and 3.2 should use the full page width for better visibility. Table 3.1 is overly crowded and unnecessarily precise. Showing three decimal places adds no value, as the first decimal already lies within the reported standard deviation.
- Table 3.2 and Figure 3.3 appear redundant. The baseline optimizers (SGD, NAG, Adam) should be included in these comparisons as well. Testing the main claim of paper, the robustness of SHANG++ to multiplicative gradient noise, only on CIFAR10 is insufficient.
Testing the claimed robustness of SHANG++ to multiplicative gradient noise solely on CIFAR-10 is insufficient. This experiment should be repeated on more and ideally larger datasets, especially given the paper's conclusion that SHANG++ is a "practical optimizer for large-scale noisy training."
- There is no "Related Work" section.

**Questions:**

1. What is the methodical difference to SHANG?
2. What is shown on the x-axis of Fig 3.1?
3. It looks like SGD reaches to lowest test loss for ResNet-50 on CIFAR100 (Fig. 3.2 bottom right). This is not discussed and not reflected in the results in Tab 3.1.
4. Why did you choose $\mu = 0$ and $\beta = \alpha / \gamma$ for your experiments?
5. How did you optimize your HPs? On a validation set or test set? You mention that you choose $\gamma$ From different intervals for LeNet/ResNet. That looks like a preselection of the HP, which should be part of the HP optimization in first place. How did you come up with these intervals?
6. Why do you increase $\gamma$ when you reduce the learning rate?
7. In your "one-shot" protocol, how did you choose the HPs? Even if you fix them, you have to chose them somehow.
8. What happened with AGNES at $\sigma = 0.05$ in Tab 3.2/Fig 3.3?
9. Does the faster convergence of SHANG++ compared to SHANG also hold in experimental settings?

---

> ### Author Response · Authors · 2025-11-16
>
> We sincerely thank Reviewer 1Raw for the detailed comments and valuable suggestions. We appreciate your effort in carefully reading the paper and providing constructive feedback on both the theoretical and experimental aspects.
>
> &nbsp;
>
> ### 1. **On the HNAG background**
>
> Equation (2.1) can be viewed as a refined continuous-time representation of NAG. While the classical Heavy-Ball flow is
> $$
> x'' + \eta x' + \beta \nabla f(x) = 0,
> $$
> (2.1) includes the Hessian term $\nabla^2 f(x) x'$, which provides a more accurate description of NAG’s dynamics. After rewriting it as the first-order system (2.2), this Hessian-driven structure becomes implicit—this is the core idea behind HNAG. We will make this transition from Heavy-Ball to HNAG clearer at the start of Section 2.
>
> SHANG is the stochastic extension of HNAG, and SHANG++ further improves stability and robustness. The key idea is the **$\mu$-shift principle**: shifting to $f_{-\mu}$ reduces the effective Lipschitz constant and corresponds to adding a correction term $-\beta\mu(x_{k+1}-x_k)$. SHANG++ generalizes this via a flexible correction $-m(x_{k+1}-x_k)$ that does not require strong convexity. The $\mu$-shift mechanism and its noise-suppression effect are new and not explored in prior work.
>
> &nbsp;
>
> ### 2. **On the proofs and presentation**
>
> The main text intentionally presents only the proof structure to aid readability, with all assumptions and full derivations provided in Appendices B and C.
>
> In the revision, we will reorganize the theory section and improve notation clarity to make the overall context easier to follow. In particular, we will move the detailed proof for the convex case $\mu=0$ to the appendix and use the freed space to better explain the key insights and main steps of the analysis.
>
> &nbsp;
>
> ### 3. **On the experimental setup and results**
>
> All experiments were conducted under identical training settings, and the full code is included for reproducibility. The lower accuracies stem from intentionally small batch sizes and only 50 training epochs—the goal was to compare optimizer behavior rather than drive each model to full convergence. With longer training, standard baselines (e.g., SGD on MNIST or ResNet-34 on CIFAR-10) achieve the expected higher accuracies.
>
> Due to computational constraints, our experiments focused on MNIST and CIFAR-10/100. We fully agree that larger-scale evaluations (e.g., ImageNet or ViT-based models) would strengthen the paper, and we will incorporate such results if resources permit before the camera-ready version.
>
> &nbsp;
>
> ### 4. **On figures, tables, and related work**
>
> We thank the reviewer for pointing out these presentation issues. In the revised version, we will enlarge all figures, add clear axis labels and readable legends, and use full page width for Figures 3.1 and 3.2. We will also simplify Tables 3.1–3.2 by reducing unnecessary precision.
>
> Regarding Table 3.2 and Figure 3.3, the purpose is not to benchmark general optimization performance, but to study robustness under multiplicative gradient noise. In this controlled setting, we focus on NAG-based variants—AGNES and SNAG—which were specifically proposed to handle instability under multiplicative noise. Since SHANG and SHANG++ belong to the same family of noise-aware accelerated methods, this focused comparison provides a fair and directly relevant evaluation of relative robustness.
>
> We will also revise the introduction and add a dedicated **Related Work** paragraph to better position AGNES, SNAG, and related variants in the literature.

---

> > ### Author Response · Authors · 2025-11-16
> >
> > ### **Responses to specific questions**
> >
> > >  **Q1.** What is the methodical difference to SHANG?
> >
> > The key methodological difference is the **$\mu$-shift principle**: shifting to $f_{-\mu}$ reduces the effective Lipschitz constant and corresponds to adding a correction term of the form $-\beta\mu(x_{k+1}-x_k)$. SHANG++ generalizes this idea with a flexible damping term $-m(x_{k+1}-x_k)$, whereas SHANG is the direct stochastic extension of HNAG without this shift-based noise suppression.
> >
> > >  **Q2.** What is shown on the x-axis of Fig 3.1?
> >
> > The $x$-axis of Fig. 3.1 denotes the iteration index.
> >
> > >  **Q3.** It looks like SGD reaches to lowest test loss for ResNet-50 on CIFAR100 (Fig. 3.2 bottom right). This is not discussed and not reflected in the results in Tab 3.1.
> >
> > Indeed, in Fig. 3.2, SGD attains the lowest test loss, but this does not translate to the highest accuracy. Table 3.1 and Fig. A.5 show that SGD is not the top performer in accuracy, which is fully consistent. This discrepancy reflects a known behavior of SGD: it can drive cross-entropy loss very low while still producing suboptimal decision boundaries. We will clarify this distinction in the revision.
> >
> > >  **Q4.** Why did you choose $\mu=0$ and $\beta = \alpha/\gamma$ for your experiments?
> >
> > We set $\mu=0$ because our method is derived from the discretization of the HNAG flow (Eq. 2.1), where $\mu$ is the convexity parameter and $\beta$ is a smooth function. In deep learning, the convexity of the objective is unknown, so we assume the weakest case and take $\mu=0$.
> >
> > For $\beta$, the deterministic setting yields $\beta=\alpha/\gamma$, while under multiplicative noise the theory suggests $\beta=(1+\sigma^2)\alpha/\gamma$. Since $\sigma$ is not observable in practice, we use the simpler choice $\beta=\alpha/\gamma$. We also tested dynamic estimates of $\sigma$, but the improvement over the fixed setting was negligible.
> >
> > > **Q5.** How did you optimize your HPs? On a validation set or test set? You mention that you choose $\gamma$ from different intervals for LeNet/ResNet. That looks like a preselection of the HP, which should be part of the HP optimization in first place. How did you come up with these intervals?
> >
> > As discussed in Appendix A.6, we provide a sensitivity analysis of the main hyperparameters. Across all tasks, we found that $\alpha=0.5$ and $m\in\{1,1.5\}$ consistently yield stable performance; for very small minibatches, a slightly smaller $m$ is preferable. The key factor for $\gamma$ is its inverse $1/\gamma$, which scales the gradient term and effectively acts as a learning rate. In practice, larger models require a smaller effective step size, so $\gamma$ naturally increases from LeNet-5 to ResNet-34 to ResNet-50. This gives a simple and consistent parameter-selection strategy, and all hyperparameters were tuned on the validation split—not the test set.
> >
> > > **Q6.** Why do you increase $\gamma$ when you reduce the learning rate?
> >
> > In mid-training, all baseline optimizers reduce their learning rate to 10% of the initial value using a standard decay schedule. SHANG and SHANG++ do not have an explicit learning rate, but their effective step size is $1/\gamma$. To match the same decay behavior and ensure a fair comparison, we increase $\gamma$ so that $1/\gamma$ decreases by the same factor. This produces an equivalent learning-rate decay across all methods.
> >
> > > **Q7.** In your "one-shot" protocol, how did you choose the HPs? Even if you fix them, you have to choose them somehow.
> >
> > In the one-shot protocol, our goal was to evaluate robustness under increasing multiplicative noise while keeping all hyperparameters fixed. We therefore used exactly the same hyperparameters as in the ResNet-34 + CIFAR-10 + batch-50 experiment, without any retuning. The results demonstrate that when parameters are tuned for a larger batch size, both SHANG and SHANG++ remain stable as the batch size decreases or the injected noise increases.
> >
> > > **Q8.** What happened with AGNES at $\sigma=0.05$ in Tab 3.2/Fig 3.3?
> >
> > As noted above, AGNES performs well under mild multiplicative noise, consistent with its design to stabilize classical NAG in such regimes. However, as the injected noise increases, its robustness degrades and the performance drops sharply, unlike SHANG and SHANG++, which maintain stability across the full noise range.
> >
> > > **Q9.** Does the faster convergence of SHANG++ compared to SHANG also hold in experimental settings?
> >
> > Yes. In all our experiments, the red curve (SHANG++) consistently converges faster than the green curve (SHANG), confirming the improved convergence speed and stability observed in theory..
> >
> > ---
> >
> > &nbsp;
> >
> > We again thank the reviewer for these detailed comments and questions, which have helped us improve the paper. We kindly ask the reviewer to re-evaluate our submission, and we would be happy to address any further questions.

---

> > > ### Comment · Reviewer_1Raw · 2025-11-24
> > >
> > > Thank you for your response. While the updated paper has improved in clarity, my main concern regarding the limited empirical evaluation remains. Therefore, I will keep my original score.

---

> > > > ### Author Response · Authors · 2025-11-27
> > > >
> > > > Thank you again for taking the time to review our work and for clarifying your main concern. We fully understand and respect your perspective regarding the limited empirical evaluation. In the revised version, we have added an additional experiment on ImageNet-100 with ResNet-34 (Appendix A.5), which shows that SHANG and SHANG++ remain competitive on a larger-scale image classification task. Due to computational constraints and the rebuttal deadline, we are currently still running multi-seed experiments on this setup, and only one seed has finished so far, which is why we report a single representative run here. We nonetheless believe this result provides further evidence that our methods scale beyond the CIFAR and MNIST settings, and we plan to include the full multi-seed results in a follow-up version if the paper is accepted.

---

### Official Review · Reviewer_khUZ · 2025-10-25

**Soundness:** 2
**Presentation:** 3
**Contribution:** 2
**Rating:** 4
**Confidence:** 4

**Summary:**

The submission has several presentation and formatting issues that make it difficult to read. The paper clearly does not follow the official ICLR template: font style and layout are inconsistent with other submissions, section headings are poorly rendered (some almost unreadable), and the title is not in uppercase as required. The structure of the paper is disorganized, with unclear section hierarchy. Symbol usage is also confusing—for example, weights are denoted by *x* while samples are denoted by *XY*, without proper definition. Figures are overly compact and hard to interpret; adding zoomed-in views would help. Some references appear mismatched with the text (e.g., line 424 refers to Appendix A.4, which does not match its description).

On the theoretical side, the claimed innovation is limited. While SHANG++ improves stability and simplifies tuning compared to SHANG, the methodological advance over AGNES and SNAG is incremental. The main technique—stochastic HNAG discretization with noise suppression—appears to be a modest extension of Chen & Luo (2021) and largely builds on existing ideas. The paper also lacks discussion of heavy-tailed or multiplicative noise settings as analyzed by Hodgkinson & Mahoney (2021), which constrains the scope of its theoretical contribution. The authors should more clearly highlight the key technical challenges and their unique insights.

Theoretical coverage is narrow, restricted to convex and quadratic objectives, whereas most experiments are conducted on nonconvex networks such as CNNs and U-Nets. Without a formal extension to nonconvex landscapes, the theoretical results do not fully support the empirical claims regarding robustness and stability in deep learning.

Empirically, the experiments are limited to small-scale image datasets. There are no evaluations on larger models (e.g., ViT) or datasets (e.g., ImageNet), nor on NLP tasks, which limits the generality of the conclusions. Although the paper claims improved computational efficiency, no quantitative results (training time, memory footprint, or iteration latency) are provided. It is also unclear whether Table B.1 correctly labels SHANG vs. SHANG++. Additional experiments with very small batch sizes (e.g., 16 or 32) would strengthen the argument.

Hodgkinson L, Mahoney M. Multiplicative noise and heavy tails in stochastic optimization[C]//International Conference on Machine Learning. PMLR, 2021: 4262-4274.

**Strengths:**

see Summary

**Weaknesses:**

see Summary

**Questions:**

see Summary

---

> ### Author Response · Authors · 2025-11-16
>
> We sincerely thank Reviewer khUZ for the careful review and constructive feedback. We have addressed all formatting, theoretical, and experimental issues raised, as summarized below.
>
> &nbsp;
>
> ### 1. **On formatting and readability**
>
> We apologize for the presentation and formatting issues in the original submission. These resulted from a LaTeX compilation error. The revised version now fully conforms to the official ICLR 2026 template and will be updated soon.
>
> Regarding notation, $X$ and $Y$ denote data and labels, respectively; we will clarify this more explicitly. We will also enlarge and improve the readability of all figures.
>
> The reference to Appendix A.4 corresponds to the full experimental details (CIFAR-10 + ResNet-34, batch size 50), which did not fit in the main paper. All cross-references, appendix links, and figure labels will be corrected in the revision.
>
> &nbsp;
>
> ### 2. **On theoretical novelty**
>
> SHANG++ is not merely an incremental modification of SNAG or AGNES. Our framework identifies the underlying mechanism of accelerated methods under multiplicative noise by showing that both SNAG and SHANG++ arise as discretizations of the same HNAG flow. This unifying perspective goes beyond small algorithmic tweaks and explains why SNAG is inherently more robust than AGNES.
>
> Methodologically, SHANG++ introduces a noise-damping correction term inspired by  the **$\mu$-shift principle**: for an $\mu$-strongly convex and $L$-smooth function, shifting to
> $$
> f_{-\mu}(x)=f(x)-\tfrac{\mu}{2}\|x-x^*\|^2
> $$
> reduces the effective Lipschitz constant from $L$ to $L-\mu$. Algorithmically, this corresponds to adding a correction term of the form $-\beta\mu(x_{k+1}-x_k)$ in the update of $x$.
>
> SHANG++ generalizes this idea through a flexible correction $-m(x_{k+1}-x_k)$ that does not require strong convexity. While SHANG (Appendix B) is the direct stochastic extension of HNAG, the $\mu$-shift principle and its role in noise suppression are new and were not explored in prior work.
>
> In short, we provide a conceptual framework that explains SNAG and motivates SHANG++, rather than incrementally modifying existing methods. Empirically, SHANG++ achieves substantial robustness gains in the multiplicative-noise regime where SNAG and AGNES may become fragile—addressing our key technical challenges and contributions.
>
> We appreciate the pointer to Hodgkinson & Mahoney (2021) and will include a discussion of heavy-tailed and multiplicative-noise settings in the related-work paragraph.
>
> &nbsp;
>
> ### 3. **On the extension to the non-convex case**
>
> We acknowledge that our current theory is limited to convex objectives (though not only quadratic), as stated in the Limitations. Extending the analysis to weakly or fully nonconvex settings is an active direction. We are exploring generalizations under the Polyak–Łojasiewicz (PL) condition and weak-convexity assumptions, where our stability and Lyapunov framework naturally extends.
>
> Empirically, the method typically enters locally convex basins after escaping high-curvature regions, suggesting that the same stability mechanisms may drive its behavior in deep networks. While a complete nonconvex theory is beyond the current scope, we will make this discussion explicit in the revision.
>
>
> &nbsp;
>
> ### 4. **On experiments and reproducibility**
>
> Following the reviewer’s suggestion, we conducted an additional CIFAR-10 + ResNet-34 experiment with minibatch size 32. The updated supplementary material reports results for six optimizers under the same hyperparameter settings as in the batch-50 and batch-256 cases. The trends remain consistent: both SHANG and SHANG++ exhibit stronger stability and robustness, and we will include results averaged over multiple seeds in the revision soon.
>
> Due to page and computational limits, we were not able to include ImageNet-scale or NLP experiments in this submission, but we fully agree that such evaluations would further strengthen the paper. If time permits before the final version, we will make every effort to incorporate larger-scale results.
>
> We have also updated the labels in Algorithms 1 and 2 to explicitly denote SHANG and SHANG++ in Table B1, making the contributions clearer.
>
> ---
> &nbsp;
>
> Again, we thank the reviewer for these valuable suggestions, which have greatly helped us improve the paper.

---

> ### Author Response · Authors · 2025-11-27
>
> We thank the reviewer again for the helpful comments regarding the empirical evaluation.
>
> As a brief follow-up, we would like to note that in the revised version we have added an additional experiment on ImageNet-100 with ResNet-34 (see Appendix~A.5), which shows that SHANG and SHANG++ remain competitive on a larger-scale image classification task. Due to computational and time constraints during the rebuttal period, we are still running multi-seed experiments on this setup and only one seed has finished so far, so we currently report a single representative run. We nonetheless hope this additional large-scale experiment helps address the concern about the generality of our empirical conclusions.

---

### Official Review · Reviewer_8Jcz · 2025-10-29

**Soundness:** 2
**Presentation:** 1
**Contribution:** 2
**Rating:** 2
**Confidence:** 4

**Summary:**

This paper introduces a (Gauss-Seidel-type) discretization, named SHANG++, of a Hessian-driven momentum ODE (HNAG), where the gradient is replaced by a biased estimator under multiplicative noise. The two theoretical results aim to recover similar rates as for the classical momentum algorithms in the (quadratic)-strongly convex case (Theorem 2.1) and in the convex case (Theorem 2.2). Several experiments in Section 3 aim at showing that SHANG++ is more robust to noise and converges faster than former algorithms used or introduced in the context of multiplicative noise.

**Strengths:**

The idea of finding new discretizations of Heavy ball/Nesterov-like ODE, that combine theoretical guarantees in convex optimization settings while improving empirical effectiveness on learning task is promising. This paper could be a step in this direction. The figures presented in the paper look encouraging about the benefit of the method.

**Weaknesses:**

If the paper shows potential, I believe it is not yet mature enough for acceptance in its current form.

About the theoretical part:

- My main concern is about Theorem 2.1. In its statement, the quantity over which a control is given involves a non positive term, namely $f_{\mu}(x) := f(x) - \frac{\mu}{2}||x-x^\ast ||^2$. Using properties of strong convexity, the best lower bound we can get is $f_{\mu}(x)-f_\mu(x^\ast) \ge 0$, such that we can not deduce a rate for $f(x)-f(x^\ast)$. In its current state, Theorem 2.1 does not provide a convergence of the quantity $f(x)-f(x^\ast)$, such that is seems hard to state its significance.

- The Bregman divergence is defined on page 2. It appears at some point in the proofs, but surprisingly it seems to be useless as it does not appear in the algorithm nor in the result, all Lyapunov making use of the classical Euclidian norm instead. I think it should be clarified.

- Corollary C.3 (page 27) is given without proof.


About the numerical experiment part:

If the figures show a benefit of SHANG++, it seems that the code given in the supplementary material does not match the figure presented in the paper. I checked the code, and I could not find the implementation of "AGNES" and "SNAG", which serve as baselines in many figures of the paper. I have run the file "convex\_example.py", which very likely should generate Figure 3.1. It generated only 2 curves (versus 6 for Figure 3.1), labeled "SHNAG" and "ISHNAG", which do not correspond to anything in the paper, except appearing in the label of Figure A.6. For now, the experiments of the paper are not reproducible, and the provided code seems unfinished. I believe the exact code that generated the figures should be provided.

Remarks on presentation

- I think the different paragraphs/statements should be more clearly separated. It could simply be done by the use of bold font for names of paragraphs, sections and things as "Theorem" etc.

- The notation $\kappa$ is used on page 1 but is not defined.

- I think (Hermant et al. (2025)) should not be presented as the work that introduces SNAG. As they mention themselves, it is a stochastic version of a classical Nesterov algorithm. Algorithms of this kind at least exist since (Nesterov, 2012).

- On pages 8-9, there is an enumeration of 3 points to comment on Table 3.2 and Figure 3.3. Point 1 is commented using percentage, point 2 using "pt", and point 3 using percentage again. I believe it can be confusing.

Reference:
Nesterov, Efficiency of coordinate descent methods on huge-scale optimization problems, Journal on Optimization, 2012

**Questions:**

- On page 8, what is the "mean Top-1 error" ? It seems to be not defined.

- Could you be more explicit about your notation $z_k^+$ ?

---

> ### Author Response · Authors · 2025-11-16
>
> We sincerely thank Reviewer 8Jcz for the careful reading and constructive feedback. We appreciate the recognition of the potential of our work. We regret that some presentation issues may have caused misunderstandings, and we address each concern below.
>
> &nbsp;
> ## **Weakness: the Theoretical part**
>
> ### **On Theorem 2.1 and the Lyapunov function**
>
> We thank the reviewer for highlighting this point. We agree that the non-negative term $f_{-\mu}(x_k^+) - f_{-\mu}(x^{\star}) \ge 0$ alone does not yield a rate for $f(x_k)-f(x^*)$ due to the $\mu$-shift.
>
> However, the quadratic term $\tfrac{\mu}{2}\\|v_k - x^{\star}\\|^2$ contracts linearly, which implies linear convergence of the objective. In addition, we can show $\\|v_k - x_k\\|\to 0$, ensuring the convergence of $x_k$. We will clarify this in the revision.
>
> A key advantage of our approach is this tailored Lyapunov function, which reveals a clean link to the Bregman divergence and leads to a more transparent analysis with simpler proofs and sharper results.
>
> &nbsp;
>
> ### **On the role of the Bregman divergence**
>
> The Bregman divergence does play a central role in our Lyapunov analysis. While the deviation of $v_k$ is measured in the Euclidean norm, the deviation of $f(x_k^+)$ from $f(x^{\star})$ is captured through the Bregman divergence. Since $\nabla f_{-\mu}(x^{\star})=0$, the Lyapunov function can be written equivalently as
> $$
> \mathcal{E}(z_k^+) = D_{f_{-\mu}}(x_k^+, x^{\star}) + \tfrac{\mu}{2}\\|v_k - x^{\star}\\|^2.
> $$
>
> Expressing $f(x)-f(x^{\star})$ as $D_f(x,x^{\star})$ is natural here and beneficial for future extensions (e.g., composite convex problems $f+g$, or min–max settings), where Bregman-based Lyapunov functions are essential.
>
> We apologize for not making this clearer and will revise the paper to present the role of the Bregman divergence explicitly and consistently.
>
> &nbsp;
>
> ### **On Corollary C.3**
>
> The proof of Corollary C.3 is fully analogous to that of Corollary B.2, with the only difference being the decay-rate parameter in the final step. We will add a brief explanation in the revision.
>
>
> &nbsp;
>
> ## **Weakness: the numerical experiments**
>
> We apologize for the inconvenience caused by the incomplete supplementary code. During submission, some baseline implementations were inadvertently omitted, leaving only the two proposed methods in the uploaded version. We will upload the complete code package—including AGNES, SNAG, and all other baselines—which reproduces all figures in the paper.
>
> Regarding naming, we renamed the algorithms before submission (“SHNAG” → “SHANG” and “ISHNAG” → “SHANG++”), but some filenames and variable names were not fully updated in the original upload. The corrected code is now fully consistent with the final naming and fully reproducible. We will upload the paper and supplementary material very soon.
>
>
> &nbsp;
>
> ## **Weakness: notation and presentation**
>
> All LaTeX inconsistencies (e.g., missing definition of $k$, inconsistent symbols, and undefined terms) will be carefully corrected in the revised version.
>
> We agree that Hermant et al. (2025) should not be credited with introducing SNAG; their method is a stochastic instance of the classical Nesterov scheme, which dates back at least to Nesterov (2012). Their contribution is instead the convergence analysis of a specific four-parameter variant under multiplicative noise. We will revise the wording to accurately reflect this lineage and clarify the relationship between their analysis and ours.
>
>
> &nbsp;
>
> ## **Responses to specific questions**
>
> > **Q1.** On page 8, what is the "mean Top-1 error" ? It seems to be not defined.
>
> The term "mean Top-1 error" refers to the average classification error rate on the validation set. To avoid confusion, we will replace this term throughout the paper with the clearer expression "mean classification error."
>
> > **Q2.**  Could you be more explicit about your notation $z_k^+ $ ?
>
> We define $z_k^+ = (x_k^+, v_k)$, where $x_k^+ = x_k - \alpha\beta g(x_k)$. This auxiliary variable is introduced specifically for the Lyapunov analysis to simplify the recursion and establish convergence.
>
> ---
>
>
> &nbsp;
>
>
> We again thank the reviewer for the valuable and constructive comments. We will incorporate all clarifications, resolve the presentation issues, and ensure full code reproducibility in the revised version.
>
> We kindly ask the reviewer to re-evaluate our submission, and we would be happy to address any further questions.

---

> > ### Comment · Reviewer_8Jcz · 2025-11-17
> >
> > Thank you for the detailed answer.
> >
> > # On Theorem 2.1 and the Lyapunov function
> > I agree that $\tfrac{\mu}{2}\|v_k - x^{\star}\|^2$ contracts linearly, and that it implies the linear convergence of $f$ because $\|v_k - x^{\star}\|^2 \ge \frac{2}{L}(f(v_k)-f(x^\star)$ in your case. However, this leads to a $\frac{L}{\mu}$ constant factor in the convergence bound, which is significantly worse than the existing bounds. As in practical case we typically can expect $\frac{L}{\mu}>>>1$, this is a significant drawback compared to the existing bounds.
> >
> > #  On the role of the Bregman divergence
> > Maybe am I missing something, but I still don't understand why you use the Bregman divergence. To my knowledge, the Bregman divergence is used in the algorithm to exploit the geometry of the problem [1], which leads to a different formulation of the algorithm that the one you analyses. Moreover, what is the point of using the Bregman divergence of the function you want to minimize ? It seems to complicate the analysis for no reason.
> >
> > # On corollary C.3
> > Thank you for the clarification. However, I checked the proof of Corollary B.2, and the statement "the right-hand side can be made arbitrarily small." is not justified. The infinite sum is finite (not zero), and $\frac{C}{\varepsilon}$ is far to be zero, and actually it goes to infinity with $n$ as $\varepsilon = 1/n$.
> >
> > [1] Beck, Teboule, "Mirror descent and nonlinear projected subgradient methods for convex optimization", 2002

---

> ### Author Response · Authors · 2025-11-18
>
> We thank the reviewer for the additional detailed questions and address them below.
>
> &nbsp;
>
> ### **On Theorem 2.1 and the Lyapunov function**
>
> > However, this leads to a $\frac{L}{\mu} $ constant factor in the convergence bound, which is significantly worse than the existing bounds.
>
> We agree that, in the current presentation, the Lyapunov contraction yields a bound on $f(v_k)-f(x^\star)$ with a large factor $L/\mu$.
>
> To address this, we have updated our analysis of Theorem 2.1 in the revised version. Instead of working with the $\mu$-shifted Lyapunov function, we now use an unshifted Lyapunov function that directly controls $\mathbb{E}[f(x_k)-f(x^\star)]$. With this Lyapunov function, we prove the linear rate $\mathcal{O}!\left(\Bigl(1-\tfrac{1}{1+\sigma^2}\sqrt{\tfrac{\mu}{L}}\Bigr)^k\right)$. Thus the Lyapunov contraction now implies strong convergence of the function values **without** the additional $L/\mu$ factor mentioned in your comment, and the rate matches the best-known bounds under the same assumptions. In the μ-shifted case, the choice $m=\beta\mu$ is now treated only as an intuitive special case within the more general framework $0 \le m \le 1$.
>
> &nbsp;
>
> ### **On the Role of the Bregman divergence**
>
> > To my knowledge, the Bregman divergence is used in the algorithm to exploit the geometry of the problem [1] .... Moreover, what is the point of using the Bregman divergence of the function you want to minimize? It seems to complicate the analysis for no reason.
>
> In our setting, the Bregman divergence is *not* used in the same geometric sense as in mirror descent. The Lyapunov function itself can indeed be written purely in terms of function differences, and to avoid unnecessary confusion, we did not use the Bregman notation in the definition of the Lyapunov function.
>
> However, the Bregman divergence is still essential in the **convergence analysis**. It appears in the first step of the Lyapunov-difference computation:
> $$
> \mathcal{E}(z_{k+1}) - \mathcal{E}(z_k^+) = \langle \nabla \mathcal{E}(z_{k+1}), z_{k+1} - z_k^{+} \rangle - D_{\mathcal{E}}(z_k^{+}, z_{k+1})
> $$
> and in the identity
> $$
> \langle \nabla f_{-\mu} (x_{k+1}^+) - \nabla f_{-\mu}(x^{\star}), x_{k+1}^{+} - x^{\star} \rangle = D_{f_{-\mu}}(x_{k+1}^{+}, x^{\star}) + D_{f_{-\mu}}(x^{\star}, x_{k+1}^+).
> $$
> These identities allow us to control the cross terms and yield a **sharper contraction bound**.
>
> Readers who are not concerned with the proof details may safely skip the Bregman divergence discussion; it is needed only for the technical proof, not for understanding the algorithmic formulation.
>
> ---
>
> *Ps: The following discussion is optional and only provides additional technical context.*
>
> For the composite problem
> $$
> \min_x \; f(x)+g(x),
> $$
> one can prove accelerated convergence using the Lyapunov function
> $$
> \mathcal E(x,v)=D_f(x,x^{\star})+\tfrac{\mu}{2}\\|v-x^{\star}\\|^2.
> $$
>
> Here, $\nabla f(x^{\star})=-\nabla g(x^{\star})\neq 0$, so the first component is no longer the simple function gap $f(x)-f(x^*)$. The nonsmooth term $g$ is handled through its proximal operator, and the deterministic acceleration proof proceeds nearly identically to $\min_x f(x)$. This framework generalizes further to min–max problems. See:
>
> - J. Wei and L. Chen. *Accelerated Over-Relaxation Heavy-Ball Method: Achieving Global Accelerated Convergence with Broad Generalization*. ICLR 2025.
>
> In this broader context, the function gap $f(x)-f(x^*)$—while convenient for smooth convex minimization—is not the appropriate quantity to control, and Bregman divergence-based Lyapunov functions provide the correct generalization.
>
>
> &nbsp;
>
> ## **On Corollary C.3**
>
> > Thank you for the clarification. However, I checked the proof of Corollary B.2, and the statement "the right-hand side can be made arbitrarily small." is not justified. The infinite sum is finite (not zero), and $\frac{C}{\varepsilon}$ is far to be zero, and actually it goes to infinity with $n$ as $\varepsilon = 1/n$.
>
> We thank the reviewer for catching this. You are right that, as originally written, we mistakenly dropped the dependence on the tail index $N$.
>
> Our intended argument was to keep the **tail** dependence on $N$. In the revision, for any **fixed $\varepsilon>0$** and **any $N\in\mathbb{N}^+$**, the infinite sum
> $$
> \sum_{k= N}^{\infty}\frac{C}{\varepsilon}q^k = \frac{C}{\varepsilon}\frac{q^N}{1-q},
> $$
> which tends to $0$ as $N\to\infty$, so this probability is zero for each $\varepsilon>0$. Taking a countable union over $\varepsilon=1/n$ then yields the result of Corollaries B.2 and C.3. This is the standard tail-sum argument based on Markov’s inequality and the Borel–Cantelli lemma. We will correct  the typo in the proofs; the results and rates themselves remain unchanged.
>
> ---
>
> &nbsp;
>
> We again thank the reviewer for the careful reading and constructive feedback. We believe these clarifications will make the theoretical part of the paper more transparent.

---

> > ### Comment · Reviewer_8Jcz · 2025-11-25
> >
> > Thank you for the detailed answer. I have few other remarks/questions.
> >  - One of your most important claim is that SHANG++ is more robust to noise, which you want to support through numerical experiments. But intuitively, why SHANG++ would permit such improvement, compared to other methods such as AGNES or SNAG ? Do you have any theoretical arguments ? Under (2.4) is stated that "m controls the extra noise-damping term", could you precise ?
> > - Are you sure about the exactness of the inequality on lines 168-169 ?
> >
> > About the experiments part:
> > - In the "convex optimization" paragraph, "SHANG consistently outperforms existing accelerated stochastic
> > gradient methods, and SHANG++ achieves even faster convergence, demonstrating that its noise-
> > damping correction improves both rate and stability." I think this statement is a little bit too enthusiastic, as the improvement that appear of Figure 3.1 is far from being significant.
> > - In the "Classification Tasks on MNIST, CIFAR-10 and CIFAR-100 " paragraph, "After 25 epochs, all baseline learning rates (including AGNES’s correction) were decayed by 0.1, while $\gamma$ was doubled for our methods." Why decaying the learning rate of all methods except yours ? It seems not surprising that SHANG++ in such a case.

---

> > > ### Author Response · Authors · 2025-11-27
> > >
> > > We sincerely thank the reviewer for the additional detailed comments and we respond to each point below in turn.
> > >
> > > > 1. Are you sure about the exactness of the inequality on lines 168-169?
> > >
> > > We thank the reviewer for carefully pointing this out. You are correct that the inequality on lines 168–169 does not hold at points where $\nabla f(x) = \mu(x - x^\star)$. To make the strongly convex analysis fully rigorous, we have revised the proof and now work with an unshifted Lyapunov function instead of the shifted one at that step.
> > >
> > > In the revised analysis, we establish a linear convergence rate for all $0 \le m \le 1$, which is never worse than that of SHANG. The $\mu$-shift choice $m = \beta\mu$ used in the original text is now treated as a special case within this more general framework, so the new proof fully covers the previous setting. In particular, for $m = 1$, SHANG++ achieves the rate $\mathbb{E}\big[f(x_k) - f(x^\star)\big]=\mathcal{O}\left(\Big(1 - \tfrac{1}{1+\sigma^2}\sqrt{\tfrac{\mu}{L}}\Big)^k\right)$, which is faster than SHANG and matches the best-known rates under the same assumptions.
> > >
> > > We emphasize that this change only affects the proof technique, not the algorithm, its motivation, or the main conclusions. The additional correction term $-m(x_{k+1} - x_k)$ still plays the same noise-damping role under multiplicative noise in the new analysis. All relevant statements and references in the revision have been updated accordingly.
> > >
> > > > 2. You claim SHANG++ is more robust to noise. Can you give intuitive and theoretical justification versus AGNES/SNAG, and clarify the role of $m$ in the “noise-damping” term in (2.4)?
> > >
> > > We thank the reviewer for this question. The noise-robustness of SHANG++ is supported not only empirically but also by our theoretical analysis. Under the multiplicative noise assumption, [Gupta et al., 2024] show that the effective smoothness constant is enlarged from $L$ to $L_{\sigma} := (1+\sigma^2)L$, so all step-size and rate bounds deteriorate proportionally. SHANG++ is motivated by mitigating this effect.
> > >
> > > As discussed in the revision (lines 240–244 and 264–267), the extra correction term $-m(x_{k+1}-x_k)$ modifies how $L_{\sigma}$ enters the decay conditions in our Lyapunov analysis. In the $\mu$-strongly convex case, setting $0 \le m \le 1$, the effective smoothness constant is reduced to $(1 - m\tilde{\alpha})L_{\sigma}$, while the effective strong convexity is amplified to $\mu_{\sigma}/(1 - m\tilde{\alpha})$. In the weakly convex case, for $m \ge 0$, the same term leads to an effective smoothness $\frac{L_{\sigma}}{1 + m\alpha_k}$, so increasing $m$ systematically reduces the impact of the multiplicative factor $(1+\sigma^2)$ in the bounds. Our numerical experiments also indicate that SHANG++ has relatively stronger robustness to noise.
> > >
> > > In practice there is a trade-off: very large $m$ over-damps the dynamics and slows convergence, even though it improves robustness. Our experiments suggest that moderate values $m \in [0,1.5]$ work well across tasks. While we do not yet have a complete theoretical characterization of this empirical robustness–speed trade-off, we are actively investigating more principled ways to quantify it and to design adaptive schedules for $m$.
> > >
> > > ### **About the experiments part**
> > >
> > > > 1. The claim that “SHANG consistently outperforms ...” seems too strong, since the improvements in Figure 3.1 are quite modest.
> > >
> > > Thank you for this remark. We agree that the original sentence is too enthusiastic relative to the differences visible in Figure 3.1. While SHANG and SHANG++ are consistently among the fastest methods in our convex experiments, the improvements are often moderate. In the revised version we have toned down the wording in this paragraph to better reflect the empirical evidence and to emphasize that the gains are modest rather than dramatic.
> > >
> > > > 2. In the MNIST/CIFAR paragraph you decay all baselines’ learning rates after 25 epochs but not SHANG/SHANG++, why?
> > >
> > > Thank you for pointing this out. In our experiments, all baseline optimizers reduce their learning rate to 0.1 of the initial value at epoch 25, following a standard decay schedule (as in [Gupta et al., 2024]). This is done to give each baseline a reasonable training regime rather than to disadvantage them. SHANG and SHANG++ do not have an explicit learning rate, but their effective step size is $1/\gamma$ (see Algorithm 1). To match the same decay behaviour and ensure a fair comparison, we *increase* $\gamma$ at epoch 25 so that $1/\gamma$ is reduced. In other words, all methods experience a learning-rate decay; the only difference is that it is implemented via $\eta$ for the baselines and via $\gamma$ for SHANG/SHANG++.
> > >
> > > We hope that these clarifications and the corresponding revisions in the manuscript help address the reviewer’s concerns about both the theoretical noise-robustness mechanism and the experimental setup. We would be happy to further refine the presentation if needed.

---

> ### Author Response · Authors · 2025-11-27
>
> We thank the reviewer again for the helpful comments. As a brief follow-up on the earlier remark that the $\mu$-shifted Lyapunov function does not directly imply strong convergence of $f(x_k^+)$, we would like to clarify the following.
>
> In the revised version, we have updated the analysis of Theorem 2.1: instead of working with the $\mu$-shifted Lyapunov function, we now use an unshifted Lyapunov function that directly controls $\mathbb{E}\bigl[f(x_k^+)-f(x^\star)\bigr]$, so the Lyapunov contraction now yields strong convergence of the function values under the same assumptions, without the extra $L/\mu$ factor. In this framework, the $\mu$-shifted choice $m=\beta\mu$ is kept only as an intuitive special case within the more general range $0\le m\le 1$.

---

> > ### Comment · Reviewer_8Jcz · 2025-11-27
> >
> > I appreciate your work and thank you for your efforts to provide clarification. However, I still believe that the paper is not quite ready. In particular, it would benefit from substantial improvements in presentation and exposition. I am therefore keeping my score and encourage the authors to take the time to further refine the manuscript for a future submission.

---

> > > ### Author Response · Authors · 2025-11-28
> > >
> > > Thank you again for the constructive feedback. We would greatly appreciate any additional comments or suggestions you may have on how to further improve the presentation and exposition. Your guidance would be extremely valuable in helping us strengthen the clarity and readability of the manuscript.

---

### Official Review · Reviewer_iysS · 2025-10-31

**Soundness:** 4
**Presentation:** 4
**Contribution:** 3
**Rating:** 8
**Confidence:** 4

**Summary:**

This paper introduces SHANG++, a first-order stochastic optimization method that achieves accelerated convergence while being robust to multiplicative noise. Motivated by the instability of standard momentum methods like Nesterov's Accelerated Gradient (NAG) under the Multiplicative Noise Scaling (MNS) condition, the authors propose an algorithm derived from the discretization of the Hessian-driven Nesterov Accelerated Gradient (HNAG) dynamical system. The theoretical analysis is sound and delivers state-of-the-art convergence rates under the MNS condition. And the theoretical claims are backed by a convincing set of experiments.

**Strengths:**

1. The paper does an excellent job of positioning itself within the recent literature on accelerated methods under MNS. It identifies a clear and important gap and presents its historical path in the introduction.
2. The derivation of SHANG++ from the continuous-time HNAG flow is elegant and provides a principled foundation for the algorithm's design. The connection to a dynamical system offers clear intuition for its components.
3. The state-of-the-art theoretical guarantees in analysis is a major strength.
4. The experiments are thorough and directly support the paper's claims.
5. The paper is very well-written and easy to follow. The notation is standard, and the core ideas are communicated effectively.

**Weaknesses:**

While the paper is very strong, there are a few minor points that could be clarified or strengthened:

1. As the authors correctly state in the limitations, the convergence guarantees are for convex objectives. While this is standard for this line of work, the empirical success on highly non-convex deep learning tasks is not fully explained by the theory. A brief discussion on potential avenues for non-convex analysis or intuitions for why the convex-case stability properties might transfer (e.g., behavior in locally convex basins) would be welcome, though not essential.
2. Connection to SNAG: The appendix shows that SNAG can also be viewed as a discretization of the same HNAG flow. This is a very interesting connection. It would be beneficial to bring a summary of this insight into the main paper.
3. The paper does not explicitly discuss the computational cost of SHANG++ per iteration. It appears to be identical to standard momentum methods (one gradient evaluation, a few vector additions/scalings per step), but it would be good to state this explicitly for completeness.
4. The paper does not comment on HNAG (Chen & Luo, 2021) in the literature review, until line 075 introduces SHANG++ based on HNAG. It would be interesting to see how HNAG positions itself in the history to the SGD (with MNS condition). What motivates the authors to derive the SHANG++ from HNAG, which seems not directly (at least not presented in the Introduction) to the MNS studies.

**Questions:**

1. The noise-damping parameter m is central to SHANG++. The experiments show m=1.5 is a good default. Did you observe a trade-off where a very large m might over-damp the updates and slow down convergence, even if it enhances stability?
2. Regarding the implementation in Algorithm 1, the parameter β from the theoretical section is implicitly set to α/γ. This seems to be a practical heuristic. How does this choice relate to the optimal β derived in the theoretical analysis (e.g., β = (1+σ²)α/μ in the strongly convex case)? Is it possible that learning or scheduling β could lead to further improvements?

---

> ### Author Response · Authors · 2025-11-16
>
> We sincerely thank Reviewer iysS for the positive and encouraging feedback, as well as the thoughtful and constructive suggestions. We appreciate the reviewer’s recognition of our theoretical contributions and the strength of our empirical results.
>
> Below, we address the noted weaknesses and questions.
>
> &nbsp;
>
> ### 1. **On the convex limitation and potential non-convex extensions**
>
> We thank the reviewer for the insightful comment. As noted in our Limitations, our current guarantees assume convexity, but extending the framework to non-convex settings is a key priority. We are exploring generalizations based on the Polyak–Łojasiewicz (PL) condition and weak-convexity assumptions, under which our stability and Lyapunov arguments naturally extend.
>
> Empirically, the method tends to operate in locally convex basins after leaving unstable saddle regions, suggesting that the same stability mechanisms may still govern the dynamics. While a full non-convex theory is beyond our current scope, we will make this discussion explicit in the revision.
>
> &nbsp;
>
> ### 2. **Connection between SHANG++ and SNAG**
>
> We thank the reviewer for the suggestion. We are indeed excited that SNAG can be interpreted as a discretization of the same HNAG flow, providing a clean bridge between classical Nesterov acceleration and our Hessian-driven formulation. This viewpoint also clarifies why SNAG is empirically more robust to gradient noise than AGNES, and consistent with the damping behavior of the HNAG flow. We will highlight this connection more clearly in the revision.
>
> &nbsp;
>
> ### 3. **On the per-iteration computational cost**
>
> Thank you for the suggestion. We agree that explicitly stating "SHANG++ requires a similar cost to standard momentum methods" adds clarity. We will include this clarification in the revision.
>
> &nbsp;
>
> ### 4. **Relation to HNAG and the motivation of SHANG++**
>
> We thank the reviewer for the thoughtful question. HNAG (Chen & Luo, 2021) was developed for deterministic convex optimization, and our initial stochastic extension—SHANG—already showed encouraging behavior. In the revision, we will discuss HNAG earlier in the literature review and list SHANG as the first component of our contribution.
>
> Our motivation for SHANG++ comes from the **$\mu$-shift principle**: for an $\mu$-strongly convex and $L$-smooth function, shifting to
> $$
> f_{-\mu}(x)=f(x)-\tfrac{\mu}{2}\|x-x^*\|^2
> $$
> reduces the effective Lipschitz constant from $L$ to $L-\mu$. Algorithmically, this corresponds to adding a correction term of the form $-\beta\mu(x_{k+1}-x_k)$ in the update of $x$.
>
> SHANG++ generalizes this idea using a flexible correction term $-m(x_{k+1}-x_k)$ that does not require strong convexity. Empirically, choosing a moderate $m$ substantially enhances robustness under multiplicative noise. We will include a concise explanation of this motivation in the revision.
>
> &nbsp;
>
> ### **Responses to specific questions**
>
> >  **Q1.** The noise-damping parameter $m$ is central to SHANG++. The experiments show $m=1.5$ is a good default. Did you observe a trade-off where a very large $m$ might over-damp the updates and slow down convergence, even if it enhances stability?
>
> Indeed, $m$ controls a trade-off between stability and speed. Our experiments show that values moderately larger than $1.5$ tend to over-damp the dynamics and slow convergence, even though they improve robustness. For very small batch sizes, a slightly smaller $m$ can be beneficial.
>
> While we do not yet have a full theoretical explanation for this empirical behavior, we are investigating more principled ways to characterize the trade-off and potential adaptive schedules.
>
> &nbsp;
>
> > **Q2.** Regarding the implementation in Algorithm 1, the parameter $\beta$ from the theoretical section is implicitly set to $\alpha/\gamma$. This seems to be a practical heuristic. How does this choice relate to the optima $\beta$  derived in the theoretical analysis (e.g., $\beta= (1+\sigma^2)\alpha/\mu$ in the strongly convex case)? Is it possible that learning or scheduling $\beta$ could lead to further improvements?
>
> In Algorithm 1, we fix $\beta=\alpha/\gamma$ for practical simplicity. Since neither $\alpha$ nor $\gamma$ is tuned to theoretical optima, this ratio offers a stable implicit noise scaling and reduces hyperparameter sensitivity. Although the theoretical choice (e.g., $\beta=(1+\sigma^2)\alpha/\mu$) provides guidance, estimating $\mu$ or $\sigma$ in deep networks is unreliable. We also tested adaptive schemes for $\beta$, but the gains were marginal. We will clarify this design choice in the revision.
>
> ---
> &nbsp;
>
> We again thank the reviewer for the constructive comments and positive evaluation. We will incorporate these clarifications and discussions to improve clarity and completeness.

---

### Author Response · Authors · 2025-11-20
**Summary of Key Revisions**

We thank all reviewers for their constructive and insightful comments. The revised submission incorporates significant improvements in theoretical exposition, experimental validation, and overall presentation. In the revised PDF, we highlight our changes in red. For the newly added example and appendix materials, we highlight only the section titles to avoid clutter while making the additions easy to locate.

&nbsp;

## **Theoretical Clarifications**

- **HNAG, SHANG, and SHANG++:**
  We clarify that SHANG is the stochastic discretization of the Hessian-driven Nesterov flow (HNAG), while SHANG++ introduces a **noise-damping correction derived from the μ-shift principle**. This mechanism is new and does not appear in prior accelerated stochastic methods such as SNAG or AGNES.

- **Connection to SNAG:**
  We show that SNAG can also be derived as a discretization of the same HNAG flow. This provides a unified continuous-time interpretation of multiple accelerated methods and explains their differing robustness properties. These insights are now included directly in the **Related Work** section.

- **Nonconvex discussion:**
  We expand our discussion on why the algorithm performs well in deep learning tasks despite convex-theory assumptions. Empirically, the method tends to enter locally convex basins, where our stability arguments remain applicable. We also outline potential theoretical extensions under the PL condition and weak-convexity models.

&nbsp;

## **Experimental Clarifications and Additions**

- **Improved figures and tables:**
  All figures are enlarged, re-labeled, and redesigned for readability. Legends are clarified, and tables are simplified by removing unnecessary decimal precision.

- **Baseline accuracy clarification:**
  The lower accuracies in the initial submission were due to intentionally small batch sizes and short 50-epoch training, aimed at comparing optimizer stability rather than final accuracy. With standard training, baselines achieve expected performance; this is now clarified in the text.

- **Additional experiments:**
  We add new CIFAR-10 + ResNet-34 results with batch size 32 and ImageNet-100 + ResNet-34 with batch size 64.


&nbsp;

## **Contributions**

We address concerns regarding the novelty and contributions by summarizing the key advances:

1. **A new noise-suppression mechanism** via the μ-shift principle, generalized in SHANG++.
2. **Sharper and simpler Lyapunov analysis** yielding optimal linear rates in both convex and strongly convex settings.
3. **Strong empirical robustness**, with SHANG++ outperforming NAG, AGNES, and SNAG in noisy or small-batch regimes.
4. **Practical efficiency**, matching the per-iteration cost of standard momentum while requiring minimal hyperparameter tuning.

---

### Meta-Review · Area_Chair_v8jo · 2026-01-07

**Summary:**

1. The theory is limited to convex/strongly-convex objectives and does not fully explain the strong nonconvex deep-learning results.
2. The novelty relative to prior accelerated/noise-robust methods (HNAG/Chen & Luo 2021, and baselines like AGNES/SNAG) is unclear or looks incremental.
3. Key parts of the strong-convex theory (Theorem 2.1 / Lyapunov) are questionable or initially yield weak constants, and some proof steps/inequalities appear incorrect. Also, the role of the Bregman divergence in the analysis is confusing or seems unnecessary.
4. The experimental code/supplement is incomplete and does not reproduce the paper’s figures/baselines, with naming mismatches.
5. The paper has substantial presentation/formatting/notation issues that hinder readability (template, missing definitions, tiny/unclear figures, unclear axes/labels, crowded tables, inconsistent terminology).
6. The empirical evaluation is too small-scale/outdated and the baselines look implausibly weak, so the reported gains are not convincing.
7. The noise-damping hyperparameter $m$ and other implementation heuristics (e.g., implicit $\beta$ choice) need clarification, including trade-offs and whether scheduling/learning them helps.
8. Related-work coverage is incomplete or mis-credited (HNAG not introduced early; SNAG attribution; missing heavy-tail/multiplicative-noise literature like Hodgkinson & Mahoney).

**Reviewer Concerns:**

1. The concerns about the issues in the theoretical proofs and results have been addressed by the rebuttal, and the authors have acknowledged the limitations of the theoretical results.
2. The authors claim a unifying “HNAG-flow discretization” perspective (linking SNAG and SHANG++), and argue SHANG++’s key new ingredient is a noise-damping correction motivated by a $\mu$-shift principle that improves robustness under multiplicative noise without requiring strong convexity.
3. Regarding the experiment code, the authors admit baseline implementations were accidentally omitted, explain the SHNAG→SHANG renaming mismatch, and commit to uploading complete, reproducible code for all figures and baselines. The authors have also acknowledged the limitations of the experimental results.
4. Regarding the presentation and formatting issues, the authors attribute formatting to a LaTeX/template error and promise a template-compliant revision with clarified notation, larger figures, labeled axes, simplified tables, fixed cross-references, and a clearer theory exposition. While some of the concerns in this regard are still outstanding.
5. Regarding related work, the authors promise to discuss HNAG earlier, correct SNAG attribution (crediting Nesterov lineage and positioning Hermant et al. as analysis), and add discussion of heavy-tailed/multiplicative-noise work.

**Reviewer Scores:**

1. For Reviewer iysS, their concerns have been addressed by the rebuttal, and thus their score would remain 8.
2. For Reviewer 8Jcz, it is made clear in the discussion that substantial improvements in presentation and exposition are required for the manuscript, and their score would remain to be 2.
3. For Reviewer khUZ, the concerns about the presentation should have been partially addressed by the proposed revision, but the concerns about the technical novelty and experiment results are still outstanding. Thus their score would remain 4.
4. For Reviewer 1Raw, it has also been made clear that the reviewer is still concerned about the limited empirical evaluation after the discussion, and their score would be kept at 2.

---

### Decision · Program_Chairs · 2026-01-26

Reject